# Z-REX uncovers a bifurcation in function of Keap1 paralogs

**Alexandra Van Hall-Beauvais[1†], Jesse R Poganik[1,2†], Kuan-Ting Huang[1†], Saba Parvez[3], Yi Zhao[1,4], Hong-Yu Lin[5], Xuyu Liu[1,6,7], Marcus John Curtis Long[8]\*, Yimon Aye[1]\***

[1]Swiss Federal Institute of Technology Lausanne, Lausanne, Switzerland; [2]Division of Genetics, Department of Medicine, Brigham and Women's Hospital, Harvard Medical School, Boston, United States; [3]Department of Pharmacology and Toxicology, College of Pharmacy, University of Utah, Salt Lake City, United States; [4]BayRay Innovation Center, Shenzhen Bay Laboratory, Shenzhen, China; [5]Department of Chemical Biology, College of Chemistry and Chemical Engineering, Xiamen University, Xiamen, China; [6]School of Chemistry, The University of Sydney, Sydney, Australia; [7]The Heart Research Institute, Newtown, Newtown, Australia; [8]Department of Biochemistry, Faculty of Biology and Medicine, University of Lausanne, Lausanne, Switzerland

**\*For correspondence:**
marcusjohncurtis.long@unil.ch (MJCL);
yimon.aye@epfl.ch (YA)

†These authors contributed equally to this work

**Competing interest:** The authors declare that no competing interests exist.

**Abstract** Studying electrophile signaling is marred by difficulties in parsing changes in pathway flux attributable to on-target, vis-à-vis off-target, modifications. By combining bolus dosing, knockdown, and Z-REX—a tool investigating on-target/on-pathway electrophile signaling, we document that electrophile labeling of one zebrafish-Keap1-paralog (zKeap1b) stimulates Nrf2- driven antioxidant response (AR) signaling (like the human-ortholog). Conversely, zKeap1a is a dominant-negative regulator of electrophile-promoted Nrf2-signaling, and itself is nonpermissive for electrophile-induced Nrf2-upregulation. This behavior is recapitulated in human cells: (1) zKeap1b-expressing cells are permissive for augmented AR-signaling through reduced zKeap1b–Nrf2 binding following whole-cell electrophile treatment; (2) zKeap1a-expressing cells are non-permissive for AR-upregulation, as zKeap1a–Nrf2 binding capacity remains unaltered upon whole-cell electrophile exposure; (3) 1:1 ZKeap1a:zKeap1b-co-expressing cells show no Nrf2-release from the Keap1-complex following whole-cell electrophile administration, rendering these cells unable to upregulate AR. We identified a zKeap1a-specific point-mutation (C273I) responsible for zKeap1a's behavior during electrophilic stress. Human-Keap1(C273I), of known diminished Nrf2-regulatory capacity, dominantly muted electrophile-induced Nrf2-signaling. These studies highlight divergent and interdependent *electrophile signaling* behaviors, despite conserved *electrophile sensing*.

## Editor's evaluation

This is an elegant, solid, carefully performed, and substantial study investigating the divergent functions of two zebrafish paralogs of Keap1, which, in mammals, is the main negative regulator of transcription factor Nrf2, which controls cell responses to antioxidants. Curiously, one zebrafish paralog augments and the other opposes Nrf2 signaling. Creative use is made of photocaged lipid-derived electrophiles to activate one Keap1 paralog at a time without stimulating other electrophile sensors. The results will be of interest to redox biologists and those interested in the regulation of stress responses through Keap1 and Nrf2.

## Introduction

Many proteins are now implicated in sensing native reactive metabolites such as lipid-derived electrophiles (LDEs; *Schopfer et al., 2011*). The resulting modified states impinge on cell physiology and behavior (*Parvez et al., 2018*; *Jacobs and Marnett, 2010*). Electrophile-responsive proteins span a large range in reactivity—from sensors engaging sluggishly with LDEs to those reacting faster than expected based on the inherent reactivity of cysteine (*Parvez et al., 2018*). Many state-of-the-art target-ID methods agree that the number of highly LDE-reactive proteins in the proteome is relatively small (*Wang et al., 2014*). Thus reactive-electrophile sensors, or privileged first responders (PFRs) *Liu et al., 2019*, have interesting properties warranting further investigation.

Our laboratory established T-REX—the only platform that interrogates with high spatiotemporal resolution individual PFR-specific LDE-modification in live cells and organisms (*Parvez et al., 2016*; *Long et al., 2017c*; *Figure 1—figure supplement 1A*). T-REX can probe consequences of on-target PFR-modification, at an accurately-determined ligand/target-binding stoichiometry, ligand chemotype, and spatiotemporal context of target engagement. These features contrast with traditional bolus electrophile-dosing methods, that simultaneously modify many proteins, and typically give readouts that cannot be clearly linked to labeling of specific protein(s), and are limited in assignment of individual ligand occupancy. T-REX has shown that electrophile signaling on PFRs typically functions through phenotypically-dominant pathways, such as gain-of-function or dominant-negative signaling *Parvez et al., 2018*. Several PFRs discovered were not enzymes, opening a new dimension into how PFRs can function to orchestrate signal propagation. Furthermore, sensor residues within PFR-enzymes are surprisingly not essential for enzyme-activity, or even close to the active site. We further demonstrated recently that understanding precision electrophile signaling is a new means to uncover novel pathway players *Poganik et al., 2021* and intersections *Long et al., 2017b*, protein-targets *Long et al., 2017c*; *Zhao et al., 2018*; *Surya et al., 2018*; *Zhao et al., 2022*; *Poganik et al., 2019* and chemically actionable sites (*Liu et al., 2020*) not traditionally considered drug discovery fare (*Long et al., 2019*).

How such sensing and dominant-signaling behaviors come about remain poorly understood. These gaps in our understanding can be traced to the fact that good model systems for reactive electrophile signaling are just coming into focus and the tools to study 'on-target' electrophile signaling have only existed for a few years. It thus remains critical to interrogate known and established signaling pathways more thoroughly using a range of different methods and comparing the outputs and conclusions that can be drawn.

One of the most venerable LDE-signal-responsive pathways is the Keap1/Nrf2-antioxidant response (AR) axis (*Hayes and Dinkova-Kostova, 2014*), wherein dimeric-Keap1 is *both* a cytosolic anchor *and* an essential component of the E3-ligase complex responsible for the degradation of the key AR-promoting transcription factor, Nrf2. *Both* functions of Keap1 serve to inhibit Nrf2. When Keap1 is LDE-modified, Keap1 function is compromised and active Nrf2 accumulates, promoting AR. Two main models have been proposed for Nrf2/AR-axis: inhibition of Nrf2 degradation through the formation of an abortive Nrf2-Keap1 complex *Zhang and Chapman, 2020*; and release of Nrf2 from Keap1 *Kensler et al., 2007*, causing inhibition of Keap1 and a boost in free Nrf2 concentrations. The Keap1/Nrf2 pathway is a primary target of some drugs in clinic, for example, the blockbuster multiple sclerosis drug, dimethyl fumarate (Tecfidera) (*Poganik et al., 2021*; *Poganik and Aye, 2020*; *Cuadrado et al., 2019*; *Figure 1—figure supplement 1B*) compounds causing release of Nrf2 from Keap1 are also under investigation *Raghunath et al., 2019*. However, consistent with the pleiotropic nature of electrophiles, it was not until recently that it was unambiguously shown that substoichiometric modification of Keap1-alone is sufficient to promote gain-of-function Nrf2/AR- signaling (*Parvez et al., 2016*; *Poganik et al., 2021*; *Long et al., 2017b*; *Parvez et al., 2015*; *Lin et al., 2015*). Thus, despite the importance of the Keap1/Nrf2-pathway, we are far from unraveling all its mechanistic mysteries.

One major roadblock is that empirical systems to study Keap1/Nrf2 signaling are limited. Experiments are typically conducted in cancer cells, which have undergone rewiring of their AR and other pathways that feedback with AR. For instance, aberrant Wnt-signaling proteins, often present in cancer cells, rewire the interaction between AR and Wnt signaling (*Long et al., 2017b*). Furthermore, mutants that exert genetically-predictable effects on AR, in terms of dominant-suppression of AR-signaling upon electrophile exposure, would also be a useful addition to the armory with which to study AR, particularly in cancer cells. However, we have often found that mutation of several postulated

electrophile-sensor residues within human Keap1 (hKeap1), did not ablate electrophile sensing, and did not ablate pathway-activation upon hKeap1(mutant)-specific labeling (*Parvez et al., 2016*; *Parvez et al., 2015*; *Lin et al., 2015*). Nrf2 mutations are also hampered by complex roles that Nrf2 plays in development and other processes loosely linked to AR (*Mills et al., 2020*).

To avoid several of the above issues, we here demonstrate a generalizable means to study on-target electrophile signaling along the druggable Keap1/Nrf2/AR-axis in zebrafish (Z-REX) (*Figure 1A*). The Keap1/Nrf2/AR-axis and consequent AR-mediated gene upregulation have been previously studied in zebrafish models, but only under bolus electrophile dosing. Using Z-REX, we explore how different segments of the embryo respond to bolus LDE exposure vs. Z-REX-assisted on-target LDE-delivery in vivo. Intriguingly, under Z-REX and under bolus dosing, AR-response is mounted in the fish tail: the head is recalcitrant to AR-upregulation. Targeted knockdown of different zebrafish paralogs of Keap1 (zKeap1) identified that zKeap1a and zKeap1b are *both* inhibitors of AR in the basal (i.e. non-electrophile-stimulated) state, but have functionally diverged in their response to electrophiles: zKeap1b is permissive for electrophile-stimulated AR-upregulation similarly to hKeap1; zKeap1a is unresponsive to electrophile-stimulated AR-induction, even in the presence of equal amounts of Keap1b. Such contrasted electrophile sensing/signaling between these paralogs was recapitulated in cell culture. These data collectively illuminate intricate on-target-modification-specific nuances of electrophile signaling. They further highlight that for some PFRs electrophile, signaling modalities are partially uncoupled from electrophile sensing.

## Results

### Halo-hKeap1 is functionally active in zebrafish and expressed at similar levels to zKeap1

To enable delivery of the desired electrophile directly to Keap1, Z-REX requires ectopic expression of Halo-Keap1 in larval fish (*Figure 1A*, *Figure 1—figure supplement 1A*). Since Halo-hKeap1 has been proven to be amenable to on-target LDE-signaling studies primarily in cultured cells (*Parvez et al., 2016*; *Long et al., 2017b*; *Parvez et al., 2015*; *Lin et al., 2015*; *Fang et al., 2013*) and *C. elegans* (*Van Hall-Beauvais et al., 2018*; *Long et al., 2018*), we first demonstrated functionality of this fusion protein in fish. Following mRNA injection directly to the yolk sac at the 1–4 cell stage, the resulting expression of Halo-hKeap1 at 24 hr post fertilization (hpf) was: (*1*) ubiquitous, and (*2*) similar to the overall level of the endogenous zKeap1, since global Keap1-signal doubled in transgene-injected fish, relative to controls (*Figure 1—figure supplement 1C*). **Note:** our Keap1-antibody detects hKeap1 with similar or higher efficiency to zKeap1a/b-paralogs-combined (*Figure 1—figure supplement 1D*). We next confirmed using two independent readouts (live-imaging and mRNA-analysis) that Halo-hKeap1 was functionally active. The imaging readout used the well-established transgenic AR-reporter strain, *Tg(−3.5gstp1:GFP)/it416b National BioResource Project Zebrafish, 2020* [hereafter *Tg(gstp1:GFP)*] (*Figure 1—figure supplement 2A*). In this and all subsequent experiments with these reporter fish, adult *Tg(gstp1:GFP)* zebrafish were crossed with wild-type (wt) zebrafish to generate embryos for experimentation. This strategy ensures transgenic progeny maintain consistent zygosity (heterozygotes) and standardization of number of reporter alleles per transgenic fish (*Figure 1B*). Imaging data were further backed-up by qRT-PCR analysis of Nrf2/AR-driven endogenous downstream genes in Casper zebrafish (*Figure 1—figure supplement 2B*). In both instances, a drop in overall basal AR and AR-driven genes was observed selectively in zebrafish injected with Halo-hKeap1 mRNA, compared to controls injected with Halo-mRNA alone (*Figure 1—figure supplement 2A-B*).

### Halo-hKeap1 manifests intriguing effects on AR in different segments of the fish

We observed during these validations differential AR-responsivity between the head and tail of the fish. This intriguing disparity was first measured in *Tg(gstp1:GFP)* fish, wherein ubiquitous Halo-hKeap1-expression led to AR-downregulation selectively in the tail (*Figure 1—figure supplement 2A*). This spatially-selective response was also observed in Halo-hKeap1-mRNA-injected Casper embryos using qRT-PCR, where the tail showed a more prominent attenuation in the AR-driven endogenous zebrafish genes, than the head (*Figure 1—figure supplement 2B*). Thus, although the qRT-PCR and

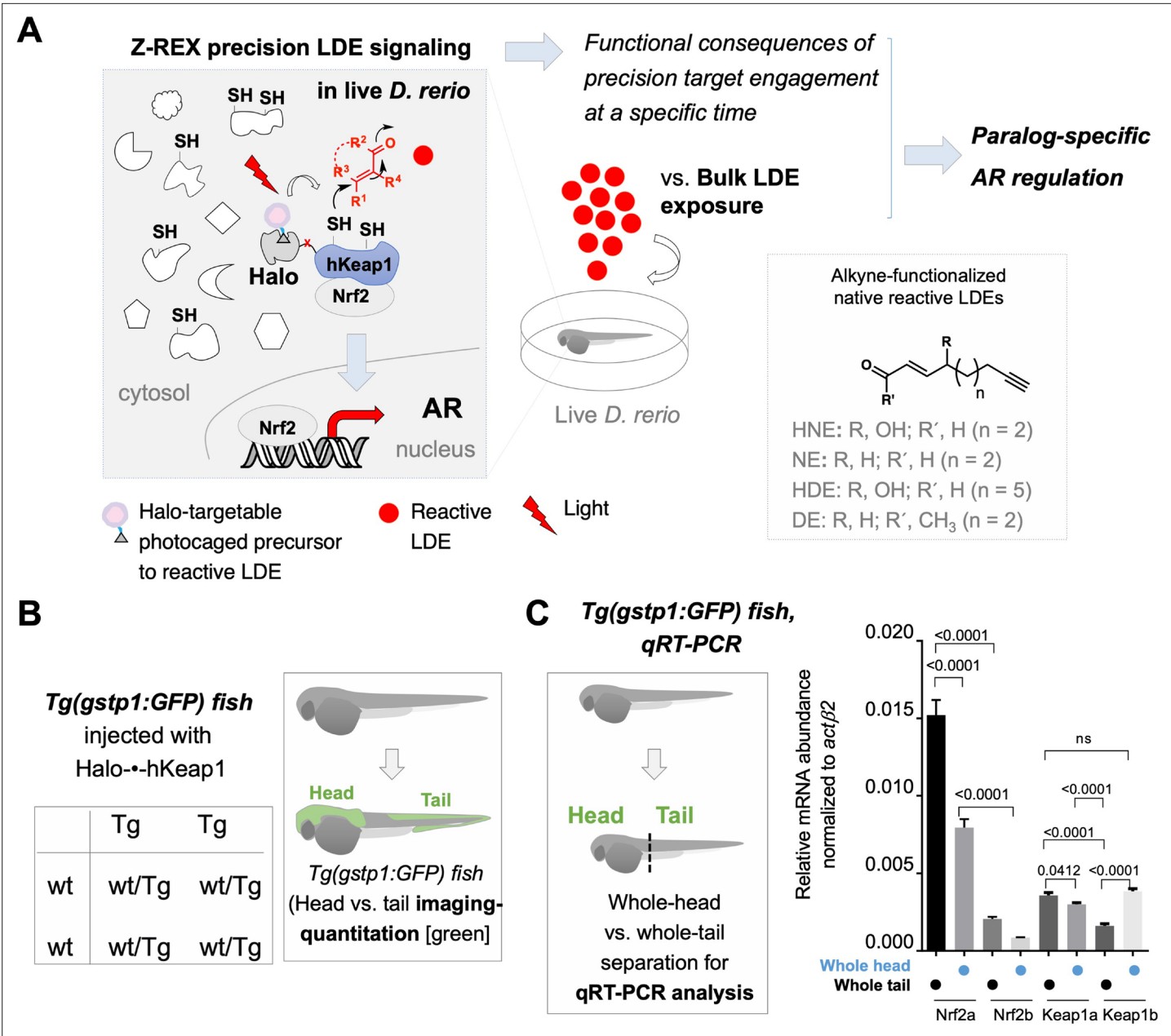

**Figure 1.** Z-REX directly evaluates the functional consequences of reactive electrophile–target engagement in live zebrafish embryos at a specific time. (**A**) This work investigates the biological impacts on druggable antioxidant response (AR) pathway at the organismal level following Z-REX-enabled hKeap1-specific electrophilic modification in live zebrafish embryos and compares these results to those obtained from bulk reactive electrophile exposure. In the process, novel paralog-specific regulation of AR was discovered. *Inset, lower right:* Structures of alkyne-functionalized lipid-derived electrophiles (LDEs). Unless otherwise specified, all LDEs deployed were alkyne-functionalized. See also ***Figure 1—figure supplement 1A***. (**B**) *Left:* Punnett square denoting how cross of wild-type zebrafish with a transgenic strain (homozygous, or heterozygous, not shown) ensures that all transgenic progeny are heterozygous for the AR-reporter GFP-gene. *Right:* Using whole-mount immunofluorescence (IF) imaging, *Tg(gstp1:GFP)* heterozygotes were quantified separately for AR levels in head and tail regions indicated in green. Note: GFP expression was detected using immunofluorescence (IF) in fixed fish, analyzed by red fluorescence. The IF protocol is used because auto-fluorescence in the green channel is high in fish and prevents accurate quantitation and this avoids concerns regarding effects of electrophile on GFP fluorescence. (For whole-head/whole-tail separation in qRT-PCR analysis, see Figure 1C, inset). (**C**) *Inset, left:* Illustration for head vs. tail qRT-PCR analysis, where the fish were mechanically separated as marked by the dashed line. *Right:* The relative levels of mRNA of each paralog were assessed using qRT-PCR following physical separation of head and tail (see inset on left). Number of embryos analyzed: Head, all paralogs (6); Tail zKeap1a and zKeap1b (10), zNrf2a (5), zNrf2b (6). Note: these segments contain tissue other than the areas that express the AR-reporter GFP-gene. All numerical data present mean ± sem. Numbers above the bars represent analysis by two-tailed *t*-tests.

*Figure 1 continued on next page*

*Figure 1 continued*

The online version of this article includes the following source data and figure supplement(s) for figure 1:

**Source data 1.** Quantification results.

**Figure supplement 1.** Halo-•-hKeap1-(2xHA) fusion protein deployed in Z-REX is ubiquitously expressed in fish to a similar level as endogenous zKeap1-paralogs.

**Figure supplement 1—source data 1.** Full view blot image.

**Figure supplement 1—source data 2.** Raw blot images.

**Figure supplement 1—source data 3.** Quantification results.

**Figure supplement 2.** Halo-•-hKeap1-(2xHA) fusion protein expressed in fish is a functional antagonist of antioxidant response (AR) pathway.

**Figure supplement 2—source data 1.** Quantification results.

**Figure supplement 3.** Validations of knockdown and MOs used by pathway response assessment and in vitro translation reporter analysis.

**Figure supplement 3—source data 1.** Quantification results.

**Figure supplement 4.** Validations of knockdown and MOs used by reverse transcription-PCR analysis.

**Figure supplement 4—source data 1.** Full view gel image.

**Figure supplement 4—source data 2.** Raw gel image.

**Figure supplement 4—source data 3.** Quantification results.

**Figure supplement 5.** Validations of knockdown and MOs used by protein expression level assessment.

**Figure supplement 5—source data 1.** quantification results.

the reporter assays likely observe different pools of cells, at least in some instances, data from the reporter and qRT-PCR are in agreement.

To further demonstrate that hKeap1 functions in zebrafish in a zNrf2-dependent manner, we made use of the zNrf2a/b-MOs previously described by several independent laboratories *Timme-Laragy et al., 2012*; *Kobayashi et al., 2002*; *Sant et al., 2017* and the resulting morphants in our hands also gave rise to the expected AR-suppression outcome in *Tg(gstp1:GFP)* fish tail (*Figure 1—figure supplement 2C*; left panel, 1st vs. 2nd bars; and *Figure 1—figure supplement 3A*). This result is consistent with the data from Keap1-overexpression above, analyzed by both imaging and qRT-PCR analysis (*Figure 1—figure supplement 2A-B*), further validating the MOs. Upon Halo-hKeap1 overexpression in these zNrf2-depleted morphants, no further AR-suppression in the tail was noted (*Figure 1—figure supplement 2C*; left panel: 1st-2nd vs. 4th-5th bars). This epistasis is strong evidence that hKeap1 and zNrf2 reside on the same axis, as required by the current model of Keap1/Nrf2/AR-signaling (*Figure 1A*). Furthermore, since hKeap1-expression had no effect on AR in the head in fish treated with control morpholinos (MOs) (*Figure 1—figure supplement 2C*; right panel: 1st vs. 4th bar), as well as in non-MO-treated reporter fish (*Figure 1—figure supplement 2A*: 3rd vs. 4th bar), there may be a dominant suppressor of AR in the head.

## Difference in head- vs. tail-AR-responsivity is principally due to zKeap1a expression

We examined zNrf2 and zKeap1 in the head and tail of the fish separately by qRT-PCR. (Consequences of expression levels of both paralogs—present in teleost fish due to a whole genome duplication event that occurred during their evolution—are also further examined below by paralog-specific-knockdown in fish and -overexpression in cell culture). **Note:** these data are, as illustrated in *Figure 1C* (inset), for the *whole* tail and *whole* head, and thus, do *not* likely reflect expression levels in specific tissues in the tail, e.g., where GFP, or other genes subsequently investigated for that matter, are expressed. The qRT-PCR analysis (*Figure 1C*) revealed that levels of both zNrf2a and zNrf2b were *overall* higher in the tail than the head. zNrf2b was a minor contributor to both segments of the fish and was almost undetectable in the head. Levels of zKeap1a were *overall* similar in the head and the tail, but zKeap1b was *overall* slightly depleted in the tail.

To investigate the function of zKeap1 paralogs and their effects on AR, we initially examined effects of their targeted-knockdown in *Tg(gstp1:GFP)* heterozygous embryos. The zKeap1a/b-MOs were first validated: (i) by in vitro translation reporter assay (for translation-blocking ATG-MOs) (*Figure 1—figure supplement 3B*), (ii) by RT-PCR (for splice-blocking SPL-MOs) (*Figure 1—figure supplement*

*4*), and (iii) by measuring depletion of signal by whole-mount immunofluorescence (for both types of MOs) (*Figure 1—figure supplement 5*). With these validations in hand, we naively expected zKeap1a-knockdown to increase AR. However, in the tail, only depletion of *both* zKeap1-paralogs increased AR (single knockdown was not effective at modulating AR) (*Figure 2A*). More strikingly still, loss of zKeap1b decreased AR in the head, but knockdown of both zKeap1-paralogs increased AR as expected. These data indicate an intriguing, antagonistic interplay between the two zKeap1 alleles.

Depletion of zNrf2a or zNrf2b (using the same MOs validated above *Timme-Laragy et al., 2012*; *Kobayashi et al., 2002*; *Sant et al., 2017*), decreased AR in the tail, confirming their expected AR-regulatory roles (*Figure 2A* left panel, *Figure 1—figure supplement 2C*, *Figure 1—figure supplement 3A*). There was little overall change in AR in the head when either Nrf2-paralog was knocked down (*Figure 2A* right panel, *Figure 1—figure supplement 3A*).

We thus investigated how AR-upregulation was affected in the different morphants by treatment with a representative bioactive LDE, 2-nonenal (NE) (*Figure 1A* inset). No effect was observed on the response in the head, upon depleting either zNrf2-paralog (*Figure 2B* right panel), but the head was unresponsive to AR regulation (*Figure 2A* right panel), and thus, this result is unsurprising. (Note: NE and other electrophiles can label proteins in the head and tail, as we describe below). Consistent with a previous report on Nrf2a-defective fish *Mills et al., 2020*, knockdown of zNrf2a suppressed the tail's ability to mount AR in response to an electrophile challenge, whereas zNrf2b-knockdown led to a hyper-elevated AR-response in the tail (*Figure 2B* left panel). From the data in the tail, it is likely that overall zNrf2b countermands electrophile-induced AR-upregulation, whereas zNrf2a functions in mounting AR in response to electrophile stimulation. Similar data for some AR-related genes were previously reported for the zNrf2-paralogs *Timme-Laragy et al., 2012*.

Since knockdown of zNrf2-paralogs did little to affect regulation of AR in the head, both in electrophile-stimulated and non-stimulated conditions (*Figure 2A–B*, right panels), we focused our subsequent investigations on how zKeap1a/b-knockdown affects LDE-induced AR. Previous overexpression studies have implied that zKeap1a (unresponsive) and zKeap1b (responsive) respond differently to different electrophiles, although these studies were performed via overexpression at early embryonic stages *Kobayashi et al., 2009*. We found that NE-treatment of zKeap1a-depleted fish led to elevated AR in both head and tail (*Figure 2B*). This effect was also observed in 8-hr old embryos. In this manifold, injection of zKeap1a-mRNA rescued effects of the MO (*Figure 2—figure supplement 1*). Unfortunately, zKeap1a-mRNA injection had minimal effect on older embryos, likely due to the stability of zKeap1a-mRNA, or possibly due to secondary effects incurred through ectopic expression of zKeap1a. Nonetheless, the fact that zKeap1a-knockdown was sufficient to rescue response to electrophile treatment in cells expressing GFP in the head (*Figure 2B*, right panel) strongly implies that the presence of zKeap1a is responsible for 'the head's' recalcitrance to mounting AR, following electrophile treatment. Similar, but less pronounced, effects were observed in the tail (*Figure 2B*, comparing the difference in the 1st and 4th bars in the left panel, vs. corresponding bars in the right panel), likely because intrinsic expression of zKeap1a within the GFP-positive cells in the tail is relatively lower than that in the head. The same posit also explains why in the absence of zKeap1a knockdown, NE induced higher fold AR-upregulation in the tail than in the head (*Figure 2B*, the 1st bars in the left vs. right panels). **Note**: the assessment of the mRNA-abundance of zKeap1a/b by qRT-PCR above (*Figure 1D*) represents global levels of allele expression in the two segments, but does not inform on levels in the GFP-positive cells used in the GFP-reporter assays. Thus, these data indicate that hypomorphism in NE-induced AR-upregulation in the tail is zKeap1a-dependent.

By contrast, in zKeap1b-knockdown fish, electrophile treatment elicited lower-fold AR-upregulation selectively in the tail (*Figure 2B* left panel). This effect could be rescued by injection of Keap1b mRNA, even though these procedures had little effect on the basal AR in this region (*Figure 2—figure supplement 2*). zKeap1b-knockdown elicited little or no effect on the head (*Figure 2B* right panel), compared to control fish. **Note:** in fish subjected to no knockdown, electrophile treatment resulted in little or no AR-upregulation (~1.1-fold) in the head, although the tail was responsive (~3.4-fold) (*Figure 2B*, 1st bars in both panels). Overall ~3-fold increase in the magnitude of AR following electrophile stimulation has been reported by us *Parvez et al., 2016*; *Parvez et al., 2015*; *Lin et al., 2015*; *Long et al., 2018* and other laboratories *Huang et al., 2012*; *Levonen et al., 2004*; *Chen et al., 2009*; *Zou et al., 2016*; *Kachadourian et al., 2011*; *Dev et al., 2015*; *Trott et al., 2008* in cultured cells and animals, using multiple orthogonal readouts including GFP- and luciferase-reporter assays

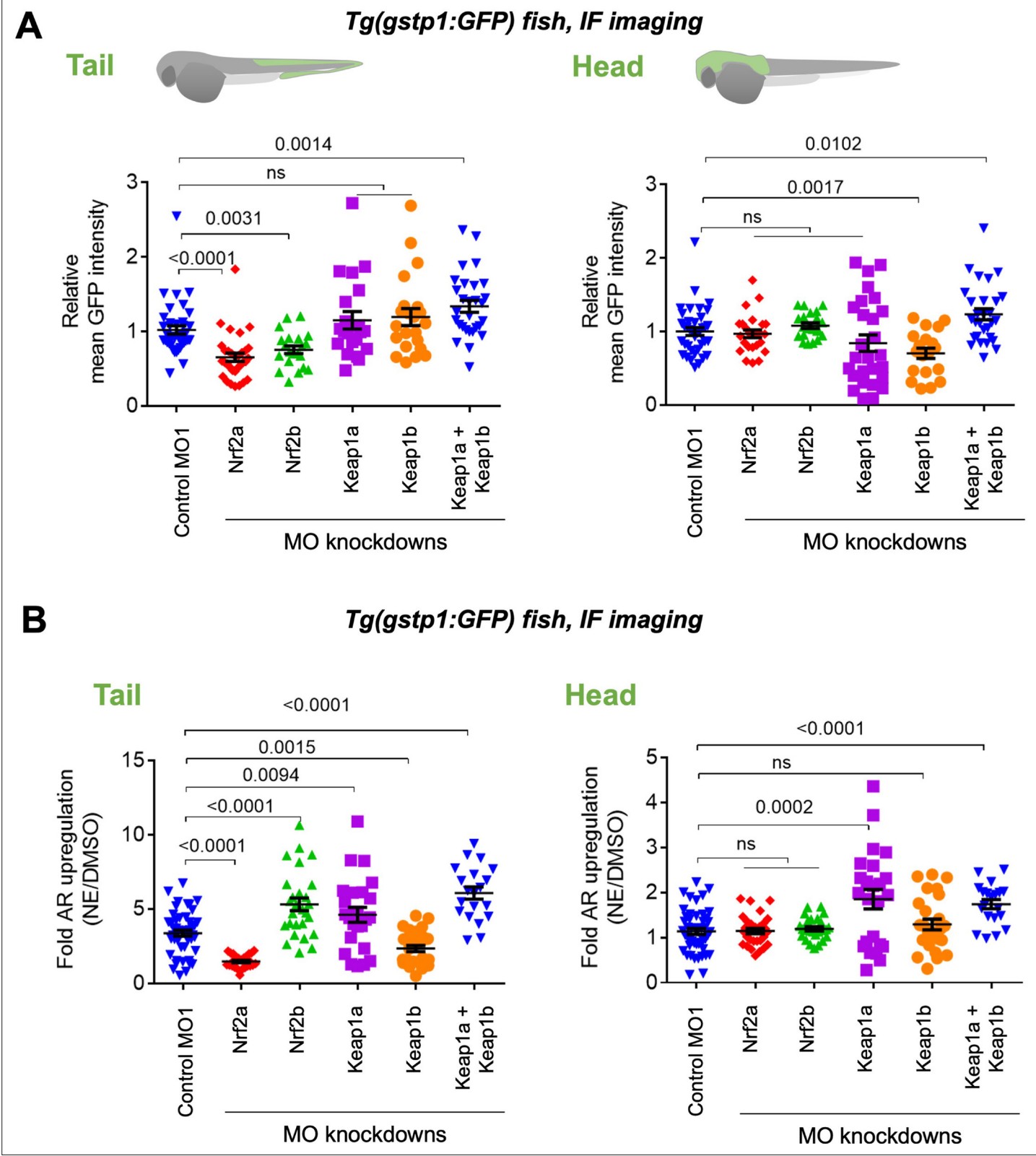

**Figure 2.** Assessments of AR-reporter in the fish tail vs head reveal differential roles of zKeap1a and zKeap1b. Homozygous *Tg(gstp1:GFP)* fish were crossed with wt fish. Resulting heterozygous embryos were injected with the stated morpholino (MO) at the 1- to four-cell stage. (See experimental Workflow in **Appendix 1-Scheme 1**). Image quantitation was performed on the head/tail-regions as illustrated. Note: GFP expression was detected using immunofluorescence (IF) in fixed fish, analyzed by red fluorescence. The IF protocol is used because auto-fluorescence in the green channel is

*Figure 2 continued on next page*

*Figure 2 continued*

high in fish and prevents accurate quantitation and IF avoids concerns regarding effects of electrophile on GFP fluorescence. ATG MOs used for single-MO injection, SPL MOs used for simultaneously knocking down zKeap1a and zKeap1b; see *Figure 1—figure supplements 3–5* and *Figure 2—figure supplements 1–2* for MO validations and Appendix for MO sequences. Also see *Figure 1—figure supplements 1–5*, *Figure 2—figure supplements 1–2*, *Figure 3—figure supplement 1*. (**A**) Quantitation of GFP expression (which indicates relative basal AR-levels) in the tail (left panel) and head (right) of *Tg(gstp1:GFP)* zebrafish larvae following MO-knockdown of the indicated zKeap1 and zNrf2 paralogs. No. embryos analyzed: Control MO (38), zNrf2a MO (32), zNrf2b MO (21), zKeap1a MO (21), zKeap1b MO (22), zKeap1a and zKeap1b MOs (29). (**B**) Quantitation of the relative *fold change* of AR level (GFP signal) in the tail (left panel) and head (right) following bulk electrophile (NE; see *Figure 1A* inset) exposure. No. embryos analyzed: Control MO (48), zNrf2a MO (27), zNrf2b MO (27), zKeap1a MO (24), zKeap1b MO (29), zKeap1a and zKeap1b MOs (20). All numerical data present mean ± sem. Numbers above the bars represent analysis by two-tailed *t*-tests.

The online version of this article includes the following source data and figure supplement(s) for figure 2:

**Source data 1.** Quantification results.

**Figure supplement 1.** Validations of zKeap1a-ATG-MO.

**Figure supplement 1—source data 1.** Quantification results.

**Figure supplement 2.** Validations of zKeap1b-ATG-MO.

**Figure supplement 2—source data 1.** Quantification results.

and by western blot and qRT-PCR analyses assessing endogenous AR-driven genes. Electrophile treatment of zKeap1a/1b-double-knockdown fish also upregulated AR in both head and tail (*Figure 2B*, both panels), an observation that may be due to re-routing of electrophile response in these fish, or possibly hinting at a dominant-negative effect being at play. Notably, zebrafish Keap1 paralogs have been shown to form heterodimers post lysis and these paralogs can occur in the same tissues, indicating that particularly in some tissues, a heterodimer of zKeap1 paralogs could be formed *Li et al., 2008*. Overall, our data and previous reports *Kobayashi et al., 2009* indicate that there is dominant-negative effect shown by one zKeap1-allele, but typical regulatory behavior shown by the other, at least under electrophile stimulation.

## Bulk LDE-exposure and hKeap1-specific LDE-modification upregulates AR similarly—analyzed by (i) AR-reporter Tg fish and (ii) qRT-PCR of endogenous AR-genes

One possible alternative explanation for the observed differences in AR in the head vs. the tail, is that bolus electrophile dosing to stimulate AR is dominated by divergent off-target effects between the head and tail (for instance, due to varied levels of detoxifying enzymes *Hayes and Dinkova-Kostova, 2014*, or different expression levels of electrophile-sensors *Parvez et al., 2018*). We thus used fish transiently expressing Halo-hKeap1 to examine how Keap1 'on-target' electrophile-modification affects AR signaling in the head and the tail.

Z-REX is built on proximity-directed delivery of an LDE in situ controlled by light (*Figure 1A* and *Figure 1—figure supplement 1A*). HaloTag–Ht-PreLDE complex serves as a latent source for rapid release of nascent LDE in an amount stoichiometric to Halo-POI *Parvez et al., 2016*; *Long et al., 2017c*. Because labeling occurs rapidly ($t_{1/2} < 1$ min) *Lin et al., 2015* and under what constitutes electrophile-limited ($k_{cat}/K_m$-type) conditions *Liu et al., 2019*, this scenario is in stark contrast to bolus LDE-administered from outside of animals, and does not significantly label proteins other than hKeap1 (see below for proof of this experimentally). This technique is not as subject to upstream factors extrinsic to hKeap1 (e.g. detoxification) as bolus dosing (*Figure 1A*), but is particularly effective at illuminating unexpected pathway intersections with Keap1-modification-specific changes in signaling *Long et al., 2017b*.

Immediately following mRNA-injection, Ht-PreLDE was introduced, and after 28.5 hr, excess Ht-PreLDE was removed by 30-min cycles of washing (x3) (see **Appendix-Scheme 1**). Our recent data establish that our Ht-PreHNE [LDE = 4-hydroxynonenal (HNE)] saturates Halo-binding site within Halo-POIs in fish embryos *Long et al., 2017c*. We found that Z-REX-targeted hKeap1-modification resulted in 1.5-fold AR-upregulation (~40% of what we observed with NE-bolus dosing) in the tail 34 hr-post-fertilization (hpf), following light-exposure 30 hpf (*Figure 3A*). The head was unresponsive.

We next compared these data with results from fish bolus-treated with HNE. The conditions for whole-animal LDE exposure (25 µM) were chosen to: (*1*) closely mirror Z-REX conditions, and (*2*)

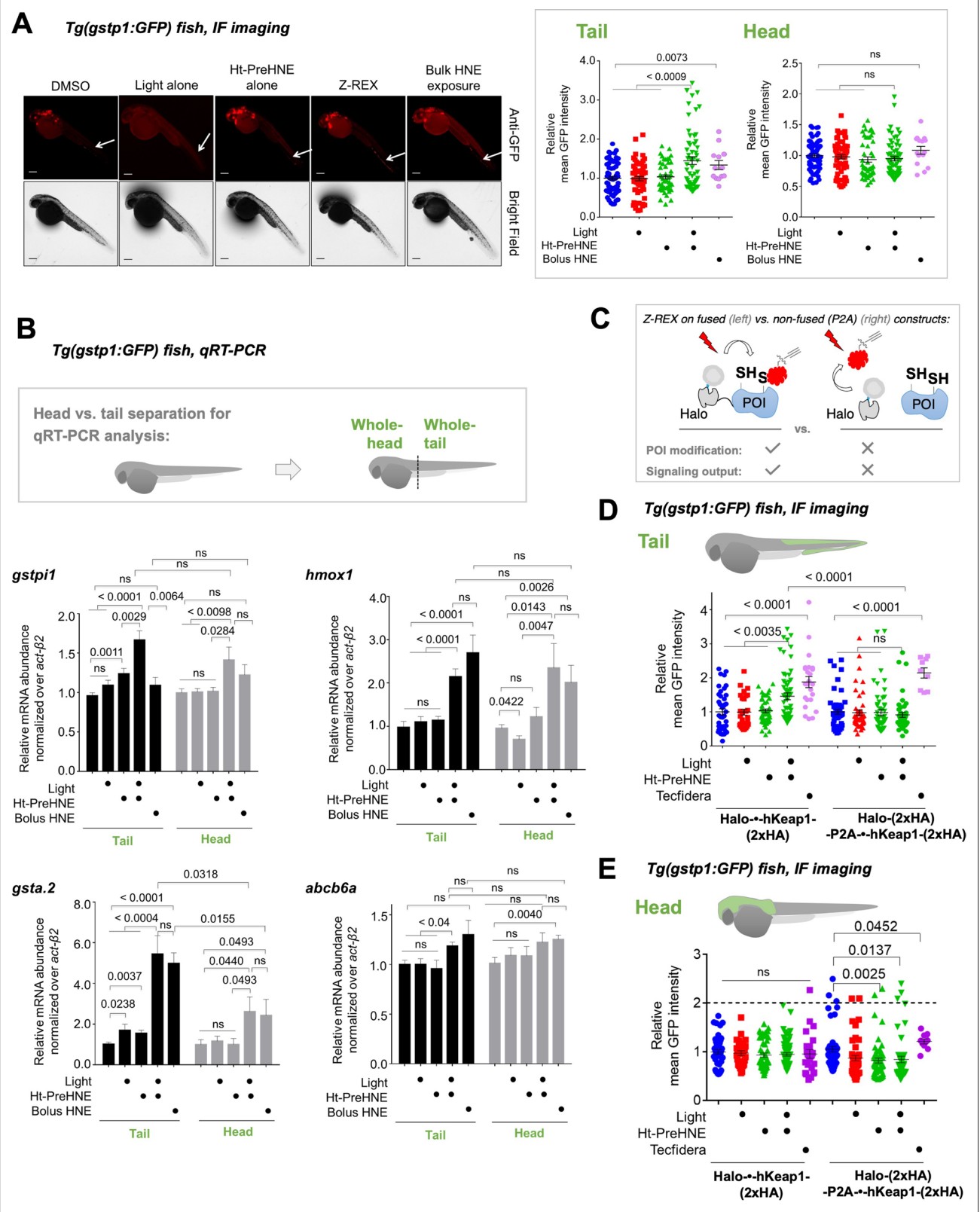

**Figure 3.** Fish expressing Halo-•-hKeap1-(2xHA) can mount AR following bolus-electrophile dosing or Z-REX-mediated hKeap1-specific electrophile labeling. (**A**) *Tg(gstp1:GFP)* heterozygotes injected with Halo-•-hKeap1-(2xHA) mRNA at one- to four-cell stage. (See *Appendix 1-Scheme 1* for workflow). *Left:* Representative IF-images of *Tg(gstp1:GFP)* fish 4 hr post Z-REX show an increase in GFP-signal intensity in the tail (arrows) subsequent to Z-REX-mediated Keap1-specific HNEylation. No AR activation was observed with all Z-REX controls (DMSO-treated, light alone, and Ht-PreHNE alone).

*Figure 3 continued on next page*

*Figure 3 continued*

[**Note:** GFP-expression was detected using red fluorescence because of high background signal in GFP (ex: 488 nm; em: 520–550 nm) channel]. *Right:* Image quantitation was performed on the head/tail-regions illustrated in *Figure 1B*. No. embryos analyzed: DMSO: No light (65), light (61); photocaged probe Ht-PreHNE: No light (47), with light (59); HNE (13). Also see *Figure 1—figure supplements 1C and 2A*, *Figure 3—figure supplement 1*. (**B**) Z-REX-targeted Keap1-specific HNEylation is sufficient to upregulate *endogenous* AR-genes represented by *gstpi1, gsta.2, hmox1, and abcb6a* (see **Appendix**). 2 hr post Z-REX or bolus HNE treatment, embryos were euthanized, RNA was isolated separately from head and tail and qRT-PCR analyses were performed as described in Methods. *Inset* above shows whole-head/-tail separation performed prior to RNA isolation. See, for workflow, Appendix 1-Scheme 1. n>4 independent biological replicates and 2 technical repeats for each sample. Also see *Figure 1—figure supplement 2B* (**C**) Illustration of a 'perfect' negative control for Z-REX using the non-fused construct that allows Halo and POI (protein of interest) to be expressed separately in vivo. See text for discussions. Replicating T-REX/Z-REX using the non-fused construct (here, P2A construct) results in ablation of POI modification by LDE as well as ablation of downstream signaling that are otherwise observed using the fused Halo-POI construct. Also see Figure 3D–E and *Figure 3—figure supplement 2*. (**D**) Z-REX-mediated AR-upregulation in the tail is observed only in Halo-•-hKeap1-(2xHA)-fusion-protein-expressing fish embryos, but *not* in the non-fused construct [i.e. Halo-(2xHA)-P2A-•-hKeap1-(2xHA)-mRNA]-injected embryos (see **Appendix** for mRNA sequence). See also Figure 3C and *Figure 3—figure supplement 2*. Bolus treatment of embryos expressing either construct with Tecfidera (*Figure 1—figure supplement 1B*) results in AR-upregulation in the tail. Image quantitation was performed on the tail-regions as illustrated. No. embryos analyzed: Halo-•-hKeap1-(2xHA): DMSO (43), Light alone (29), Ht-PreHNE alone (47), Z-REX (58), and Tecfidera (24); Halo-(2xHA)-P2A-•-hKeap1-(2xHA): DMSO (55), Light alone (49), Ht-PreHNE alone (52), Z-REX (47), and Tecfidera (9). See also *Figure 3—figure supplement 2B*. (**E**) Z-REX-mediated-AR-upregulation is not observed in the head. The dashed line indicates the average level of AR-upregulation in the tail following bulk exposure to Tecfidera (*Figure 1—figure supplement 1B*). Image quantitation was performed on the head as illustrated. No. embryos analyzed: Halo-•-hKeap1-(2xHA): DMSO (43), Light alone (29), Ht-PreHNE alone (49), Z-REX (65), and Tecfidera (24); Halo-(2xHA)-P2A-•-hKeap1-(2xHA): DMSO (55), Light alone (48), Ht-PreHNE alone (54), Z-REX (49), and Tecfidera (10). All numerical data present mean ± sem. Numbers above the bars represent analysis by two-tailed *t*-tests.

The online version of this article includes the following source data and figure supplement(s) for figure 3:

**Source data 1.** Quantification results.

**Figure supplement 1.** Z-REX execution maintains viability and proper development of embryos, whereas bolus LDE treated embryos show reduced viability following prolonged incubation.

**Figure supplement 1—source data 1.** Quantification results.

**Figure supplement 2.** Comparison of results from Z-REX using fused vs non-fused Halo-POI constructs (see *Figure 3C*) allows precise measurement of 'on-target/on-pathway' responses in vivo.

**Figure supplement 2—source data 1.** quantification results.

**Figure supplement 3.** Z-REX on-target electrophile modification strategy validates proximity-directed electrophile responsivity.

**Figure supplement 3—source data 1.** Full view blot image.

**Figure supplement 3—source data 2.** Raw blot image.

**Figure supplement 4.** Markedly contrasting to Z-REX (i.e., combined treatment with both light and Ht-PreHNE), wherein liberated LDE efficiently labels hKeap1 in situ, the extent of hKeap1 labeling upon bolus HNE-treatment of fish is minimal.

**Figure supplement 4—source data 1.** Full view blot image.

**Figure supplement 4—source data 2.** Raw blot image.

**Figure supplement 5.** Z-REX-mediated AR upregulation following hKeap1-specific low-occupancy LDE-modification, is transient.

**Figure supplement 5—source data 1.** Quantification results.

avoid prolonged bolus LDE exposure that was deleterious to embryonic development (*Figure 3—figure supplement 1*), whereas exposure to photocaged precursors of similar or higher dose/longer duration was easily tolerated in these larval fish: Ht-PreHNE showed little effect on development, even when Z-REX was performed at 30 hpf and embryos were left to develop for another 3 days (*Figure 3—figure supplement 1*). Whole-fish treatment with HNE (25 µM) at 30 hpf, for 4 hr, elicited similar tail-specific AR-upregulation as was found under Z-REX (*Figure 3A* inset; the last two bars in left panel). Taken together, differences between head and tail response, and the fact that fold changes in AR are generally relatively small, collectively indicate that the measured response in the tail is indeed a biologically relevant change.

We next examined by qRT-PCR the extent of upregulation of *endogenous* AR-driven genes of conserved importance in higher eukaryotes, under both regimens, namely: global LDE-exposure and Z-REX-mediated Keap1-specific-modification. Representative genes associated with drug metabolism under control of Nrf2 (Gst-isoforms, Hmox1, and Abcb6) were activated to similar levels between Z-REX and bulk HNE-treatment, and AR modulation was most prominent in the tail (*Figure 3B*), particularly in

the case of *gsta.2*, but also for *abcd6a* that only showed a significant upregulation relative to controls in the tail. *gstpi1* showed hypomorphic responses in the head relative to the tail, but these were not significantly different, at this level of statistical power. We ascribe this modest AR-upregulation in the head [seen only by qRT-PCR analysis of several genes, and not by immunofluorescence(IF)-imaging of *Tg(gstp1:GFP)*] to increased sensitivity of qRT-PCR analysis compared to in vivo fluorescence-imaging, and the fact that the *gstp1* locus (used in the GFP-reporter fish) is not the most responsive in the head. By contrast, bolus LDE-treatment yielded mixed responses in most cases (*Figure 3B*). We further note that mRNA extracted from head or tail covers cells that are not necessarily examined in *Tg(gstp1:GFP)*, rendering data from head and tail in the qRT-PCR (e.g. *Figure 3B*) and IF-imaging (e.g. *Figures 2 and 3A*) assays not necessarily comparable.

## Replication of Z-REX using non-fused construct rules out off-target signaling and validates on-target/pathway interrogations—analyzed by: (i) AR-reporter Tg fish and (ii) qRT-PCR of endogenous AR-genes

To validate that AR-upregulation observed upon Z-REX is due to Keap1-specific HNEylation as opposed to off-target effects, we compared the extent of AR modulation in fish expressing either Halo-•-hKeap1, or the 'non-fused' Halo-P2A-•-hKeap1 (where '•' designates TEV-protease-site that is cleaved post lysis). The P2A-sequence allows for slipping of the translation machinery *Kim et al., 2011*, thereby expressing Halo and hKeap1 proteins separately (i.e. non-fused), from the same mRNA, during the same translation step. Based on our previous cell-based studies *Parvez et al., 2016*; *Long et al., 2017c*; *Parvez et al., 2015*; *Lin et al., 2015*; *Van Hall-Beauvais et al., 2018*, the non-fused system wherein Halo and POI are expressed as two separate proteins serves as a robust negative control in our proximity-directed LDE-targeting platform (*Figure 3C*): when electrophile release is executed in the non-fused system, both labeling of POI and downstream ramifications are ablated. A similar level of ubiquitous expression of hKeap1 was achieved following injection of mRNA encoding the non-fused construct (*Figure 3—figure supplement 2A*). Similar to our previous cell-based data, the P2A-integrated construct yielded little or no AR-upregulation upon Z-REX, compared to that achieved with Halo-•-hKeap1, demonstrated by both IF-imaging of GFP-reporter upregulation (*Figure 3D–E*) and qRT-PCR analyses targeting 3 different endogenous AR-driven genes (*Figure 3—figure supplement 2B*). These data also show that minor upregulation of AR-specific genes due to treatment with light alone, or photocaged-compound alone, do not synergize significantly and hence do not contribute dramatically to responses observed during Z-REX. Importantly, upon bolus exposure to Tecfidera (*Figure 1—figure supplement 1B*), an approved pleiotropic electrophilic drug that stimulates AR as part of its pharmaceutical program *Poganik et al., 2021*; *Poganik and Aye, 2020*, both fused and non-fused systems gave twofold AR-upregulation, particularly in the tail region (*Figure 3D–E*).

## LDE-labeling extent of hKeap1 is different between bolus conditions and Z-REX-target-specific interrogations

Importantly, at the level of target labeling by Z-REX, we showed that the non-fused expression system strongly-diminished the extent of LDE-signal on hKeap1 compared to the fused construct (*Figure 3—figure supplement 3A-B*). We demonstrated this outcome using 'biotin-Click-streptavidin pulldown' of LDE-modified proteins in fish, subsequent to Z-REX in vivo. Briefly, lysates from either control fish or Z-REX-treated fish—expressing either Halo-•-hKeap1 or Halo-•-P2A- hKeap1—were all subjected to: (in the former instance) TeV-protease to separate Halo and Keap1; Click-coupling with biotin-azide; and streptavidin enrichment to evaluate LDE-modified proteins (see **Appendix-Scheme** 1).

Using the same enrichment protocol, we examined the extent of hKeap1 labeling subsequent to bolus HNE-treatment of embryos (*Figure 3—figure supplement 4*). We unexpectedly found little discernable hKeap1-modifcation. This result is surprising because bolus dosing promoted AR-upregulation to a level similar to that elicited by Z-REX targeted hKeap1-HNEylation (*Figure 3A–B*). This result underscores the importance of using Z-REX to parse on-target electrophile signaling, especially in organisms because phenotypic effects of *on-target* LDE-modification can be directly and unambiguously interrogated by Z-REX. However, because bolus dosing affects numerous proteins, and changes redox balance, such outcomes are overall not surprising. To further investigate this phenomenon, we compared how AR was induced as a function of time under bulk-exposure and Z-REX.

## Latencies and duration of AR are similar between bolus exposure and Z-REX

AR increased as the fish aged, such that the DMSO-treated group had significantly higher AR 48 hpf than 34 hpf (*Figure 3—figure supplement 5A-B*). Thus, even though Z-REX-mediated Keap1-specific LDEylation led to a 1.5–1.7-fold increase in AR 4 hr post light-exposure (ple) (34 hpf), there was no significant difference between Z-REX and controls 18 hr ple (48 hpf) (*Figure 3—figure supplement 5C*). To further examine the time-dependent AR-upregulation following Z-REX, we measured AR-up-regulation at 34 hpf (*Figure 3—figure supplement 5D*). However, photouncaging was executed at different time points prior to harvesting (1.5, 4, and 8 h ple).

Interestingly, Z-REX gave a transient AR-upregulation, which peaked around 4 hr ple but returned to basal levels 8 hr ple (*Figure 3—figure supplement 5D*). This profile is consistent with the Z-REX model, wherein the transient release of LDE in an amount sub-stoichiometric to POI in vivo mimics signaling conditions *Liu et al., 2019*; *Long et al., 2021*, such that AR-upregulation is not sustained. Conversely, when we examined the effects of bolus HNE-treatment, a similar kinetic profile was observed (*Figure 3—figure supplement 5E*), although fish were constantly exposed to electrophile. It is likely that the off-target effects incurred during bolus-electrophile treatment cause insults that lead to severe negative effects on fish, which we showed above are not present in Z-REX (*Figure 3—figure supplement 1*). This finding makes precise comparisons between bolus dosing and Z-REX in terms of absolute outcomes difficult, and may explain the difficulty in correlating labeling of Keap1 with the extent of AR-upregulation under bolus dosing with that resulting from Z-REX. Regardless, the overall conclusions of the two methods in terms of head vs. tail AR are surprisingly similar.

## All LDEs have similar capacity to mount AR following hKeap1-specific LDEylation in live fish—analyzed by (i) AR-reporter Tg fish and (ii) qRT-PCR of endogenous AR-genes

The above results (primarily obtained with HNE, selected because it is the most well-known LDE) were intriguing, and we were keen to understand how Keap1 labeling contributed to overall AR for structurally-homologous native LDEs (*Figure 1* inset). HDE (4-hydroxydodecenal, identified in human urine *Florens et al., 2016* and heated oils *Seppanen and Csallany, 2004*), NE (nonenal, another endogenously-generated LDE exhibiting an age-dependent rise in production *Haze et al., 2001*), and DE (3-decen-2-one, an FDA-approved food additive with natural occurrence in certain fruits and mushrooms *Knowles and Knowles, 2012*) were chosen as representatives. As with HNE, the HaloTag-targetable photocaged precursors to HDE, NE, and DE have been successfully applied to target hKeap1 selectively in cells *Parvez et al., 2016*; *Lin et al., 2015*. The measured AR-upregulation in each case is of comparable efficiency to that obtained under whole-cell LDE flooding *Lin et al., 2015*. Using *Tg(gstp1:GFP)* fish, we found that these LDEs elicited similar AR-upregulation in fish tails upon Z-REX (*Figure 4A*), but the head was not responsive (*Figure 4B*).

Independent qRT-PCR analysis of endogenous AR-driven genes following Z-REX in casper fish gave broadly-consistent results across all the LDEs examined (*Figure 4C*). Using Click-biotin-pulldown following Z-REX in vivo, we validated using DE as an example that hKeap1-labeling extent was similar to that achieved with HNE (*Figure 4—figure supplement 1*). Given that AR under Z-REX conditions stems only from hKeap1-modification, these findings explain the similar magnitude of AR-outputs observed with Z-REX. These outcomes were further substantiated by qRT-PCR analysis of downstream genes discussed above (*Figure 3—figure supplement 2B*), where we showed that Halo-P2A-•-hKeap1 was hypomorphic for Z-REX-assisted hKeap1-HNEylation-promoted AR-upregulation.

## Whole-fish LDE-exposure elicits complex AR outputs that are dependent on LDE structure: allylic alcohols disfavor AR-upregulation—analyzed by (i) AR-reporter Tg fish and (ii) qRT-PCR of endogenous AR-genes

Having established that each LDE when targeted specifically to Keap1 could trigger tail-specific AR-upregulation, we compared effects on AR from whole-fish-LDE-exposure under otherwise identical conditions/timescales to Z-REX. Bolus dosing with these LDEs elicited widely different AR that was chemical-structure dependent. By both imaging of the AR-reporter fish (*Figure 5A–B*) and qRT-PCR

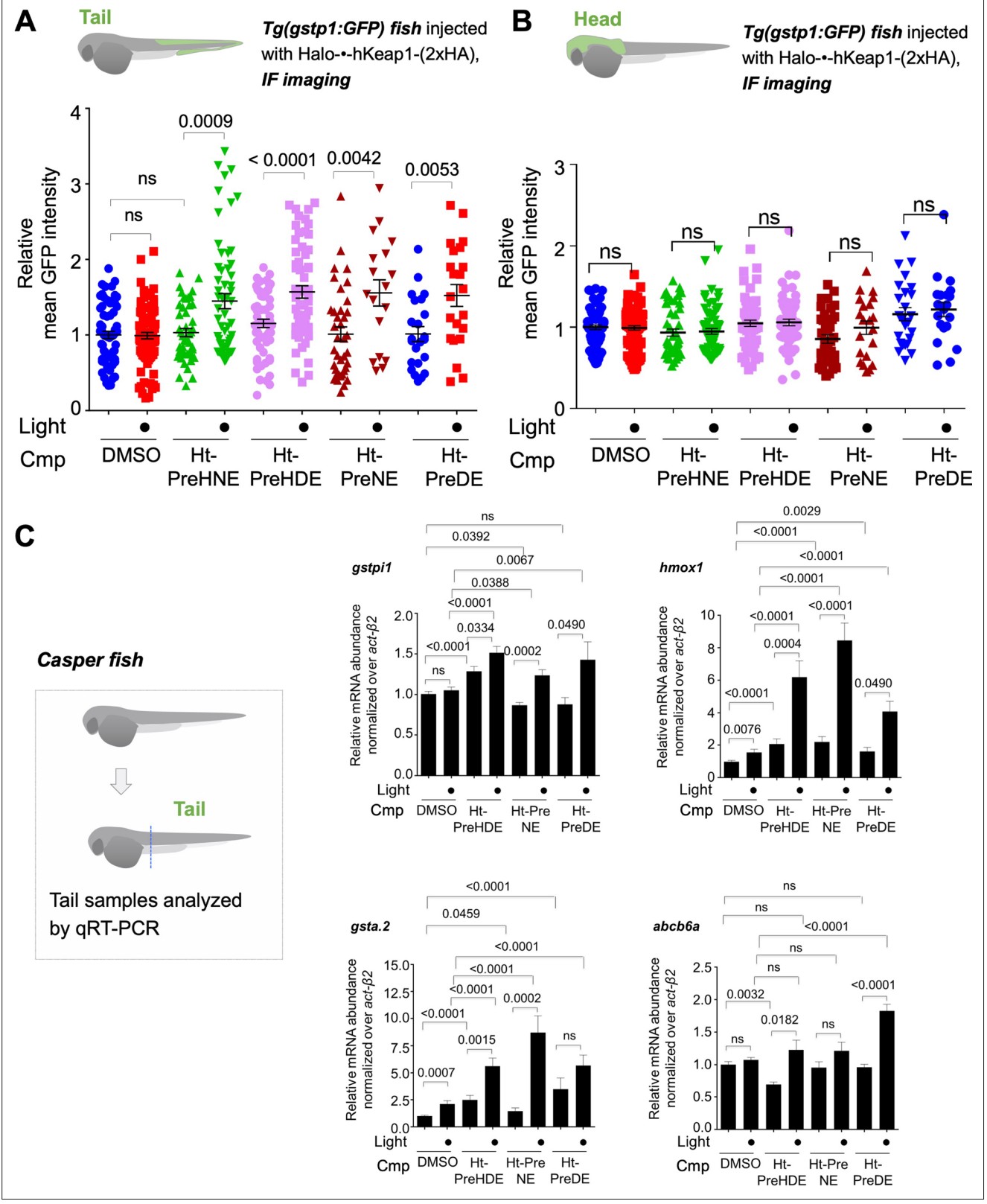

**Figure 4.** Z-REX delivery of 4-different electrophiles studied consistently labels hKeap1 and activates AR to similar extent (as previously observed in cell culture). Also see **Figures 5–6** and **Figure 3—figure supplement 1**, **Figure 3—figure supplements 3–5**, **Figure 4—figure supplement 1**, **Figure 5—figure supplement 1**. (**A**) Quantitation of mean AR-levels in the tail of embryos 4 h post Z-REX with indicated LDEs. Image quantitation was performed on the tail-regions as illustrated. No. embryos analyzed: DMSO: No light (65), with light (84); Ht-PreHNE: No light (47), with light (59); Ht-PreHDE: No

*Figure 4 continued on next page*

*Figure 4 continued*

light (59), with light (59); Ht-PreNE: No light (38), with light (18); Ht-PreDE: No light (23), with light (22). (**B**) Similar quantitation in the head shows no increase in AR post Z-REX. Image quantitation was performed on the head-regions as illustrated. No. embryos analyzed: DMSO: No light (65), with light (82); Ht-PreHNE: No light (49), with light (65); Ht-PreHDE: No light (63), with light (60); Ht-PreNE: No light (38), with light (21); Ht-PreDE: No light (23), with light (22). (**C**) hKeap1-modification alone is sufficient to drive *endogenous* AR-gene upregulation in the tail in casper zebrafish. Whole-head-/-tail separation was performed as indicated in inset (left), prior to RNA isolation selectively from the tails. 2 h post Z-REX with indicated LDEs, embryos were euthanized, and RNA was isolated, and qRT-PCR analyses were performed on tail samples (see inset, left) targeting indicated downstream genes (see Appendix for primer sequences). n>4 independent biological replicates and 2 technical repeats for each sample. Inset: schematic for fish separation. Note: tail was taken as a representative segment in these experiments. All numerical data present mean ± sem. Numbers above the bars represent analysis by two-tailed *t*-tests.

The online version of this article includes the following source data and figure supplement(s) for figure 4:

**Source data 1.** Quantification results.

**Figure supplement 1.** DE efficiently labels hKeap1 upon bolus treatment of whole embryos Cf. results from bolus treatment with 4-OH-bearing LDEs, such as HNE (*Figure 3—figure supplement 4*).

**Figure supplement 1—source data 1.** Full view blot image.

**Figure supplement 1—source data 2.** Raw blot image.

analysis of endogenous AR-driven genes (*Figure 5C*), LDEs that did not contain a 4-hydroxyl group were best at eliciting AR. Interestingly, 4-dehydroxy species are intrinsically less electrophilic than their 4-hydroxylated counterparts. This result was markedly different from those seen under Z-REX targeted conditions for the same LDEs.

We also assayed the effects of electrophiles of clinical relevance under bulk-exposure. Tecfidera gave outputs between those elicited by bolus HNE and NE (and of higher magnitude than the Z-REX responses) (*Figure 5A–B*; Cf. *Figure 3D*). Bardoxolone methyl (CDDO-Me; Phase II trials recently completed for pulmonary hypertension, *Figure 1—figure supplement 1B*) was severely toxic to the fish at this concentration, although AR was upregulated (*Figure 5—figure supplement 1A*). Sulforaphane (*Figure 1—figure supplement 1B*) did not elicit AR under these conditions (*Figure 5A–B*). However, regardless of the magnitude of the output in the tail, where responses were observed, there remained minimal, and indeed non-statistically-significant, increase in AR in the head (*Figure 5B*).

## Extent of the fish proteome labeling following bulk LDE-exposure closely mirrors that of AR induction

We next examined the ability of the LDEs to label the fish proteome following whole-fish treatment (*Figure 6A–B*, *Figure 5—figure supplement 1B*). Interestingly, this trend closely mirrored that of AR-upregulation elicited upon bolus dosing, with hydroxyl-bearing LDEs (HNE and HDE) manifesting a reduced level of proteome labeling and corresponding reduced AR-induction (*Figure 5* vs. *Figure 6A-B*, *Figure 5—figure supplement 1B*). Thus, AR-upregulation upon bulk LDE-exposure is dominated by permeation/pharmacokinetic-effects over inherent ligand electrophilicity. However, the divergent AR-induction in the front and hind portions of the fish was retained.

## Examination of zKeap1a and zKeap1b unveils divergent regulatory roles for these proteins in AR

We thus predicted that zKeap1a is a negative regulator of electrophile-stimulated AR, and zKeap1b is the principal means through which electrophile-stimulated AR is mounted in zebrafish. Based on this hypothesis, we examined the phylogeny between the two zebrafish genes and the human protein, for clues as to which residue could lead to the proposed unusual behavior of zKeap1a. We reasoned that signaling could be affected through mutation of a cysteine in the human protein (sensing competent) to a bulky residue in Keap1a (mimicking the electrophile-modified state *Poganik et al., 2018*). In Keap1b, this residue would remain a cysteine. Based on this logic (*Figure 7—figure supplement 1A*), we identified C273 (human numbering); the analogous residue in zKeap1a is an isoleucine, but it remains a cysteine in zKeap1b (*Figure 7A* and *Figure 7—figure supplement 1B*). We thus cloned hKeap1(C273I) in a bid to find a humanized form of zKeap1a. This mutation has already been identified as a loss-of-function mutation to hKeap1 (*Levonen et al., 2004*; *Saito et al., 2016*), although it has not been extensively characterized, particularly in terms of how it interacts with wild-type(wt)-hKeap1.

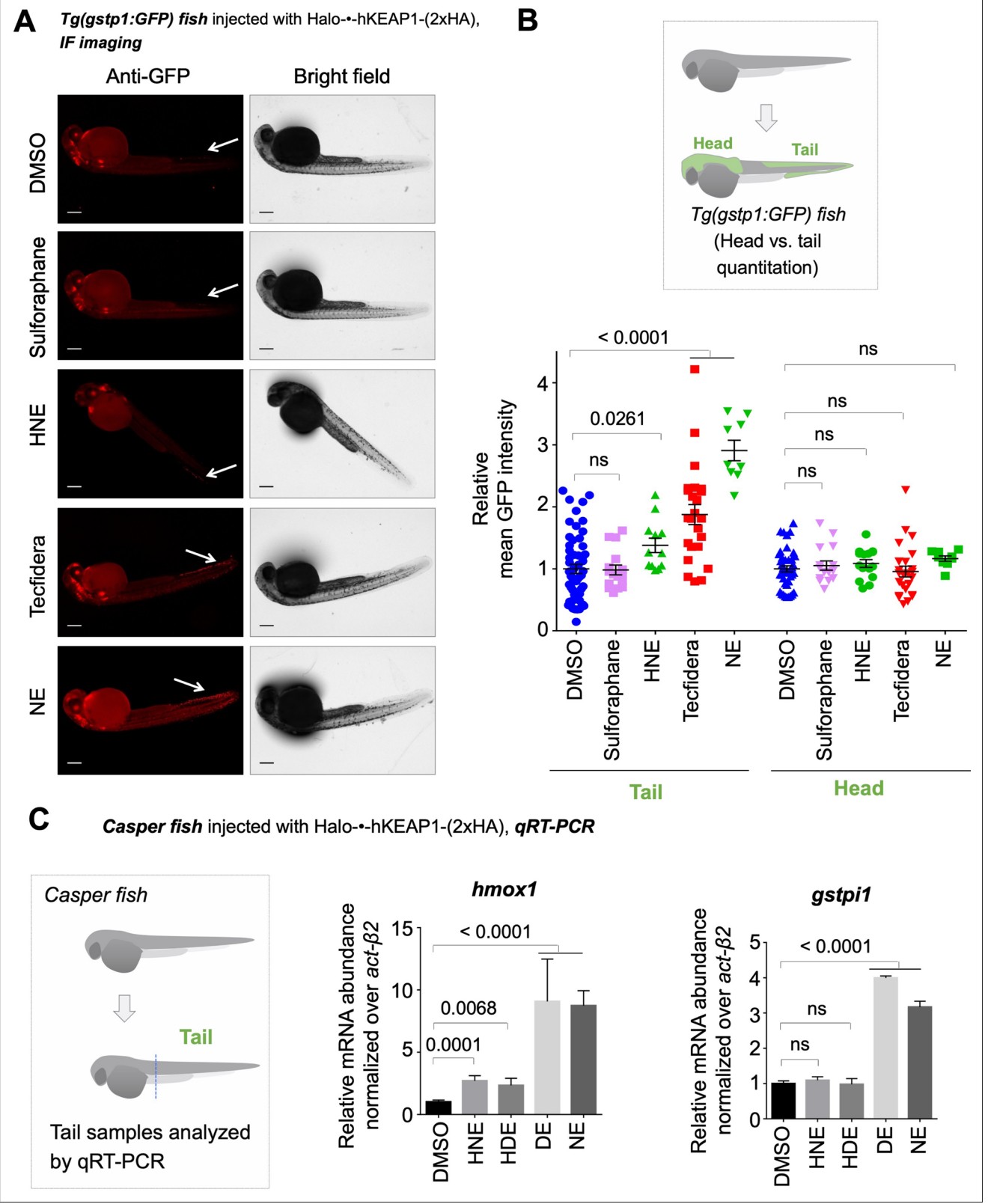

**Figure 5.** Bolus dosing with different LDEs or reactive covalent electrophilic drugs elicits complex AR responses. Also see **Figure 6**, **Figure 3—figure supplements 1 and 4** and 5E, **Figure 4—figure supplement 1**, and **Figure 5—figure supplement 1**. (**A**) Representative IF images of *Tg(gstp1:GFP)* fish expressing Halo-•-hKeap1-(2xHA), following bolus exposure to indicated electrophiles. (**B**) Quantitation of data in (**A**). Image quantitation was performed on the head/tail-regions as illustrated. No. embryos analyzed: Tail, DMSO (55), Sulforaphane (16), HNE (12), Tecfidera (24), NE (9); Head,

*Figure 5 continued on next page*

*Figure 5 continued*

DMSO (43), Sulforaphane (16), HNE (15), Tecfidera (24), NE (9). Sulforaphane and HNE elicit non-significant and 1.5-fold AR upregulation, respectively. Tecfidera gives medium (~2-fold) AR response and NE elicits the strongest AR upregulation (~3-fold) in tail. Consistent with data elsewhere (e.g., *Figure 3A and C-E*), head is not responsive. (**C**) qRT-PCR analysis of *endogenous* AR-responsive genes following bolus exposure of native reactive LDEs to whole fish similarly shows mixed responses. Whole-head/-tail separation was performed as indicated in inset (left), prior to RNA isolation selectively from the tails. 2 hr post Z-REX, embryos were euthanized, and RNA was isolated separately from tail (see inset, left). Data are presented as mean ± sem. n>3 independent biological replicates and two technical repeats for each sample. All numerical data present mean ± sem. Numbers above the bars represent analysis by two-tailed *t*-tests.

The online version of this article includes the following source data and figure supplement(s) for figure 5:

**Source data 1.** Quantification results.

**Figure supplement 1.** Uncontrolled bulk exposure of developing larvae with reactive electrophiles results in differential proteome labeling extent and adversely affects development/viability.

**Figure supplement 1—source data 1.** Quantification results.

We thus progressed to evaluate hKeap1(C273I) more deeply. We also amplified zKeap1a and zKeap1b from cDNA and cloned these genes into mammalian expression vectors with a HaloTag.

Behavior ascribed to zKeap1a could be due to an inability to sense LDE that leads to an overall muted AR, similar to what has been previously postulated *Kobayashi et al., 2009*. We thus first examined the sensing abilities of the four Keap1-variants to NE and HNE using T-REX in HEK293T cells ectopically expressing respective Halo-tagged proteins. All four constructs expressed similarly in HEK293T cells and at levels far above that of the endogenous Keap1-protein, allowing us to discard the influence of the endogenous hKeap1. hKeap1 was expressed at marginally higher levels than some of the other proteins (*Figure 7—figure supplement 2A*). Intriguingly, all four proteins sensed HNE equally well (*Figure 7B* and *Figure 7—figure supplement 2B-C*). hKeap1 and hKeap1(C273I) sensed NE marginally better than zKeap1a/b, both of which sensed NE to the same extent (*Figure 7B*). Thus,

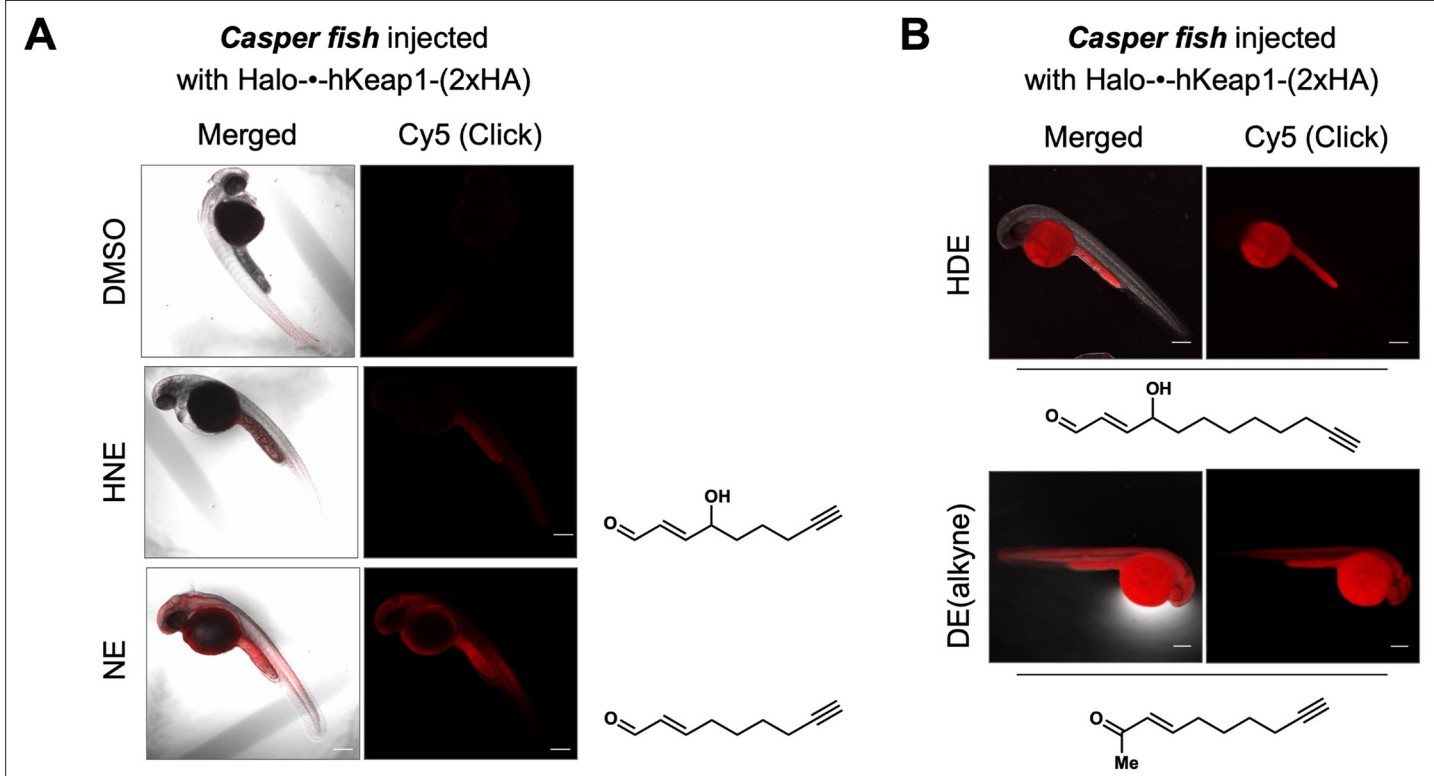

**Figure 6.** Bolus LDE treatment results in different extent of whole-reactive-proteome labeling that correlates with the magnitude of AR-upregulation (*Figure 5A–C*). See also *Figure 5—figure supplement 1*. (**A**) Casper zebrafish were treated with indicated LDEs for 2 hr before euthanasia and fixing. See Methods for whole-reactive-proteome labeling procedure in fish using Click coupling. (**B**) Comparison of whole-reactive-proteome labeling extent of HDE and DE over time (2 hr) shows allylic alcohol motif within LDEs is likely responsible for reduced Keap1-labeling degree under bolus conditions.

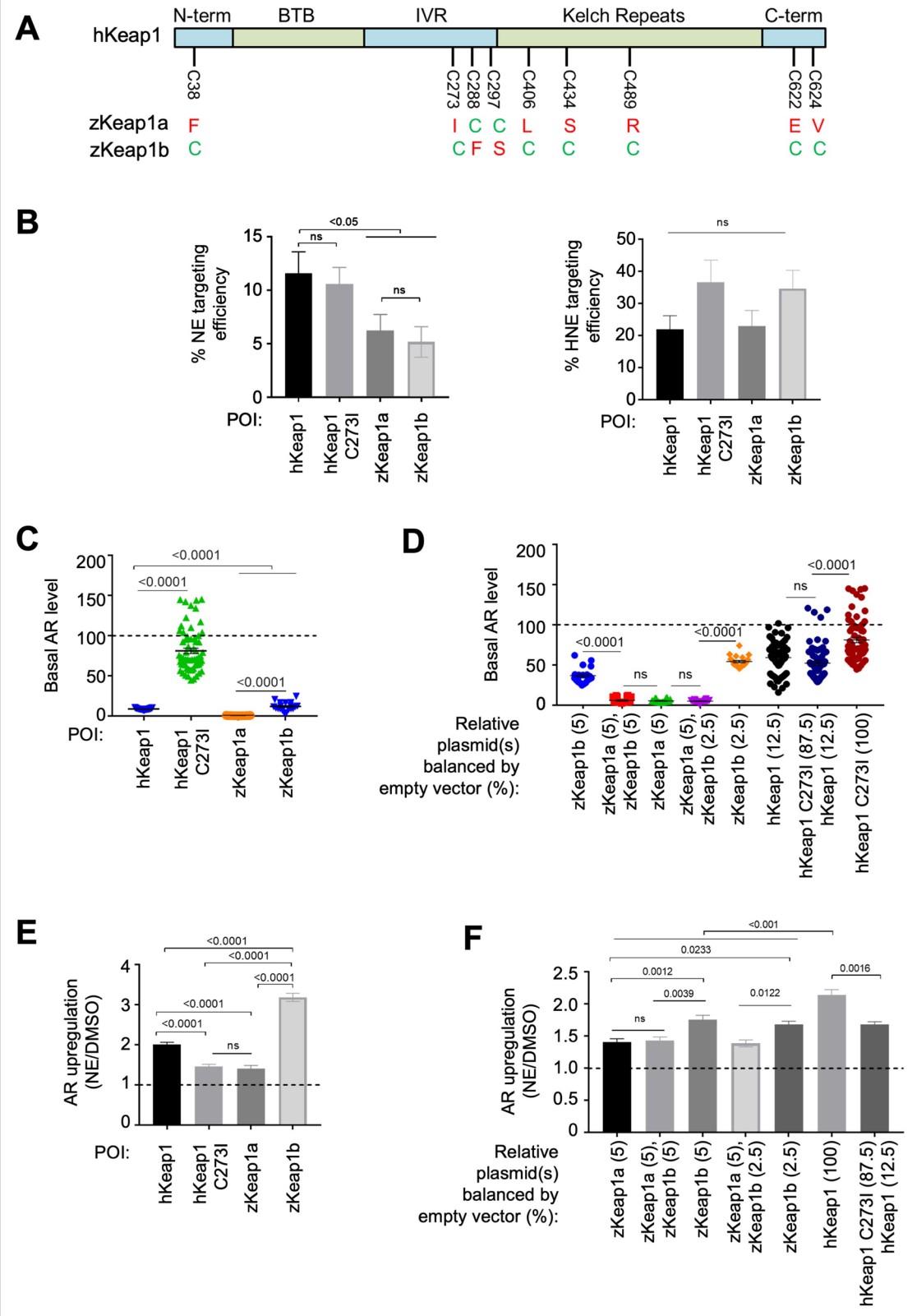

**Figure 7.** Cell-based studies of zKeap1a and zKeap1b recapitulate the dominant-negative behavior observed in developing embryos; cell-based T-REX analysis (*Figure 1—figure supplement 1A*) reveal similar electrophile sensitivity across all Keap1-variants. See also *Figure 8A–B*, *Figure 7—figure supplements 1–2* and *Figure 8—figure supplement 1*. (**A**) The nine cysteines within hKeap1 that are present in only one of the two Keap1 paralogs in zebrafish. (N-term, BTB, IVR, Kelch-Repeats, C-term are individual conserved domains of Keap1). All indicated cysteines are conserved between human

*Figure 7 continued on next page*

*Figure 7 continued*

and zebrafish Keap1. (**B**) HEK293T cells were transiently transfected to express indicated Halo-•-Keap1 constructs. (See *Figure 1—figure supplement 1D*, *Figure 7—figure supplement 2A and D* for validation of construct functionality). 36 hr post transfection, cells were treated with Ht-PreNE (10 µM, 2 hr), and after rinsing cycles, cells were then exposed to UV light (5 mW/cm² 365 nm lamp). Post lysis, samples were treated with TeV protease and subjected to Click coupling with Cy5-azide. The targeting efficiency of NE on each variant was calculated using a previously-reported procedure *Parvez et al., 2016*; *Van Hall-Beauvais et al., 2018*. See *Figure 7—figure supplement 2B-C* for representative in-gel fluorescence and western blot data. (**C**) HEK293T cells were transfected with plasmids encoding ARE:Firefly luciferase and CMV:Renilla Firefly reporters, human myc-Nrf2, and Keap1 (Halo-•-hKeap1, Halo-•-(3xFLAG)-zKeap1a, Halo-•-(3xFLAG)-zKeap1b, Halo-•-hKeap1 C273I, or empty vector). (See *Figure 1—figure supplement 1D*, *Figure 7—figure supplement 2A and D* for validation of construct functionality). Basal (non-electrophile-stimulated) AR levels were quantified using a standard procedure *Parvez et al., 2016*; *Long et al., 2017b*; *Van Hall-Beauvais et al., 2018*. The horizontal dotted line represents basal AR levels with no exogenous Keap1 introduction. All conditions show a significant drop compared to basal level (i.e. with no exogenous Keap1 overexpression). No. independent biological replicates: Halo-•-hKeap1 (n=65), Halo-•-hKeap1 C273I (n=63), Halo-•-(3xFLAG)-zKeap1a (n=19), Halo-•-(3xFLAG)-zKeap1b (n=19). These were all dosed at a plasmid loading equivalent to 100%. Also see *Figure 7—figure supplement 2A and D*. (**D**) HEK293T cells were transfected with a mixture of plasmids encoding Halo-•-(3xF)-zKeap1a and Halo-•-(3xF)-zKeap1b in various ratios this mix also contained empty vector as required, myc-Nrf2, and AR reporter plasmids, see (**C**) See *Figure 1—figure supplement 1D*, *Figure 7—figure supplement 2A and D* for validation of construct functionality. 36 hr post transfection, AR was measured using a standard procedure *Parvez et al., 2016*; *Long et al., 2017b*; *Van Hall-Beauvais et al., 2018*. The horizontal dotted line indicates the basal AR levels in the absence of exogenously-introduced Keap1. Percentages are relative to those analyzed in *Figure 7—figure supplement 2D*. No. independent biological replicates: n=22 for zKeap1a/b mixing, n=63 for WT/C273I hKeap1 mixing. (**E**) A similar experiment to (**C**) except AR in response to NE bolus dosing was measured in HEK293T cells transfected with: Halo-•-hKeap1(WT); Halo-•-(3xFLAG)-zKeap1b; Halo-•-hKeap1(C273I); and Halo-•-(3xFLAG)-zKeap1a. (See *Figure 1—figure supplement 1D*, *Figure 7—figure supplement 2A and D* for validation of construct functionality). The horizontal dotted line represents the normalized AR-level for respective Keap1-variants following DMSO-treatment in place of NE. No. independent biological replicates: hKeap1 WT (n=28), hKeap1 C273I (n=28), zKeap1a (n=20), zKeap1b (n=20). (**F**) A similar experiment to (**D**), except AR in response to NE bolus dosing was measured in HEK293T cells. (See *Figure 1—figure supplement 1D*, *Figure 7—figure supplement 2A and D* for validation of construct functionality). Note: the indicated mix of Halo-•-(3xFLAG)-zKeap1a and Halo-•-(3xFLAG)-zKeap1b upregulated AR to a similar extent as Halo-•-(3xFLAG)-zKeap1a alone. The horizontal dotted line represents the normalized AR-level for respective Keap1-variants following DMSO-treatment in place of NE. Percentages are relative to those described in *Figure 7—figure supplement 2D*. No. independent biological replicates: n=54 for zKeap1a/b mixing, n=20 for WT/C273I hKeap1 mixing. All numerical data present mean ± sem. Numbers above the bars represent analysis by two-tailed *t*-tests.

The online version of this article includes the following source data and figure supplement(s) for figure 7:

**Source data 1.** Quantification results.

**Figure supplement 1.** Keap1 is widely conserved and Keap1 variants from fish and human mutants sense NE to a similar extent.

**Figure supplement 2.** Keap1-paralogs/variants express at similar levels in cells; zKeap1a is an overall more-effective suppressor of basal AR than the other variants; but all manifest similar sensitivity to HNE and NE analyzed by cell-based T-REX assays.

**Figure supplement 2—source data 1.** Full view gel and blot image.

**Figure supplement 2—source data 2.** Raw gel and blot image.

**Figure supplement 2—source data 3.** Quantification results.

differences in sensing abilities of the different proteins cannot explain the data we, and indeed others, have observed with zKeap1a and zKeap1b.

There remains the possibility that the different zKeap1-paralogs have different modes of marshalling AR in response to electrophiles. We thus examined these behaviors, again using cell culture. When the drop in AR was measured following transfection of equal amounts of the different zKeap1 plasmids (which, based on the experiments above, gave roughly similar amounts of protein), we found that all zKeap1a/b and hKeap1 could all significantly suppress AR, again consistent with previous reports *Li et al., 2008*. zKeap1b appeared to be less efficient at suppressing basal AR than zKeap1a (*Figure 7—figure supplement 2D*), which does not completely agree with previous reports, although the previous reports are limited both in terms of statistical power, and direct comparability as they were carried out in early embryos where signal-to-noise can be low. zKeap1a was similar at suppressing AR relative to hKeap1. hKeap1(C273I) suppressed basal AR, but marginally (*Figure 7C*). Critically, none of these conditions were able to suppress basal AR to undetectable levels (*Figure 7—figure supplement 2D*). Intriguingly, when zKeap1b was co-transfected with sub-saturating amounts of zKeap1a, no decrease in basal AR was observed relative to zKeap1a alone (*Figure 7D*), implying that zKeap1a somehow affects the ability of zKeap1b to suppress AR. When we performed similar experiments mixing hKeap1 and hKeap1(C273I), a similar effect was observed (*Figure 7D*).

We progressed to examine response of the different Keap1-variants upon electrophile modification. For these purposes, we used NE as a bolus electrophile. We selected NE because in fish NE

was the most permeable molecule giving the most robust AR and labeling (*Figure 5A–C*, *Figure 6A*, *Figure 5Figure 1B*). Intriguingly, although hKEAP1 and zKeap1b were significantly responsive to bolus NE treatment, significantly upregulating AR, zKeap1a, and hKeap1(C273I) were much less responsive (*Figure 7E*). In terms of AR-upregulation ability, these data fully recapitulate data from zebrafish embryos expressing zKeap1a and zKeap1b *Kobayashi et al., 2009*. These data further imply that the C273I mutation is a likely cause of the zKeap1a's inability to mount AR. Furthermore, as the ability to suppress basal levels of AR were significantly different between zKeap1a and hKeap1(C273I) (*Figure 7C*), this suppressive effect is independent of fold basal-AR suppression. However, this finding does not fully explain what we found from the initial morphant data (*Figure 2B*); that is, why knock-down of zKeap1a could lead to hyper-elevated AR in zebrafish following bolus electrophile treatment. Such a mechanism could be explained if zKeap1a were a dominant-negative regulator of AR in the electrophile-stimulated states.

We next transfected equal amounts of zKeap1a/b plasmids, or 2:1 zKeap1a:zKeap1b and compared AR-levels upon bolus electrophile treatment to those found in cells transfected with zKeap1a- or zKeap1b-alone. Consistent with a dominant-negative effect on AR induction, NE caused zKeap1b-alone-expressing cells to induce AR significantly, whereas zKeap1a-alone or zKeap1a/Keap1b-expressing cells mounted AR significantly less efficiently upon NE treatment (*Figure 7F*). A similar effect was observed when hKeap1 and hKeap1(C273I) were mixed. These data explain why morpholinos targeting zKeap1a led to upregulation of electrophile-induced AR in both tail and especially the head and provide evidence for dominant-negative effects during Keap1/AR signaling.

## Differential extent of altered Nrf2-binding in response to electrophile signaling confers divergent AR-management by zKeap1a vs. zKeap1b

To examine how these paralog-divergent electrophile responses arose, we next examined how NE treatment affected the amount of Nrf2 bound to zKeap1a and zKeap1b, or the heterozygotic state. For these experiments we used HEK293T cells transiently expressing Nrf2, co-transfected with empty vector, zKeap1a, zKeap1b, or equal amounts zKeap1a and zKeap1b. We found that zKeap1b accumulated Nrf2 in the basal (i.e. non-electrophile-stimulated) state, whereas relative to zKeap1b, zKeap1a accumulated less Nrf2, and zKeap1a/zKeap1b accumulated an amount of Nrf2 that was significantly more than Keap1a alone and less than zKeap1b alone (*Figure 8—figure supplement 1*). When treated with NE, zKeap1b-bound Nrf2 was diminished by approximately 40% (*Figure 8A–B*). Conversely for zKeap1a and zKeap1a/zKeap1b no release of Nrf2 was observed. These data agree with the paralog-specific regulation of electrophile signaling observed above, which showed that zKeap1a and the heterozygotic state were unable to upregulate AR effectively following electrophile treatment. Furthermore, given that significant amount of Nrf2 is bound to zKeap1b in both the heterozygous- and zKeap1b-only-expressing states, these data are consistent with a model in which electrophile engagement of zKeap1b can trigger AR through causing net release of Nrf2, whereas zKeap1a (either with or without zKeap1b) cannot lead to net release of Nrf2 upon electrophile labeling, and hence cannot upregulate AR. Thus, zKeap1a (either electrophile modified or unmodified) acts to suppress loss of Nrf2 binding to zKeap1b.

## Discussion

This study has furnished several deliverables. On a technical level, we have established a simple, yet realistic and versatile organismal model system to interrogate on- and off-target effects of pathophysiologically-relevant native lipid-derived electrophilic metabolites represented by HNE, HDE, NE, and DE (*Figure 1A*). Because of its putative druggability and clear links to physiological well-being and disease, we used AR as a conserved model pathway *Poganik et al., 2019*; *Hayes and Dinkova-Kostova, 2014*; *Poganik and Aye, 2020*, using a reporter fish strain that is freely available *National BioResource Project Zebrafish, 2020*. Many of our assays using the reporter fish were also further supplemented with qRT-PCR, an analysis that is applicable to most signaling pathways and that interrogates endogenous genes. Furthermore, we have shown that our in vivo electrophile-targeting approach is functionally compatible with other reporter assays (e.g. Akt/FOXO pathway in live fish *Long et al., 2017c*). It is thus likely that this regimen will be readily adaptable to the study of how

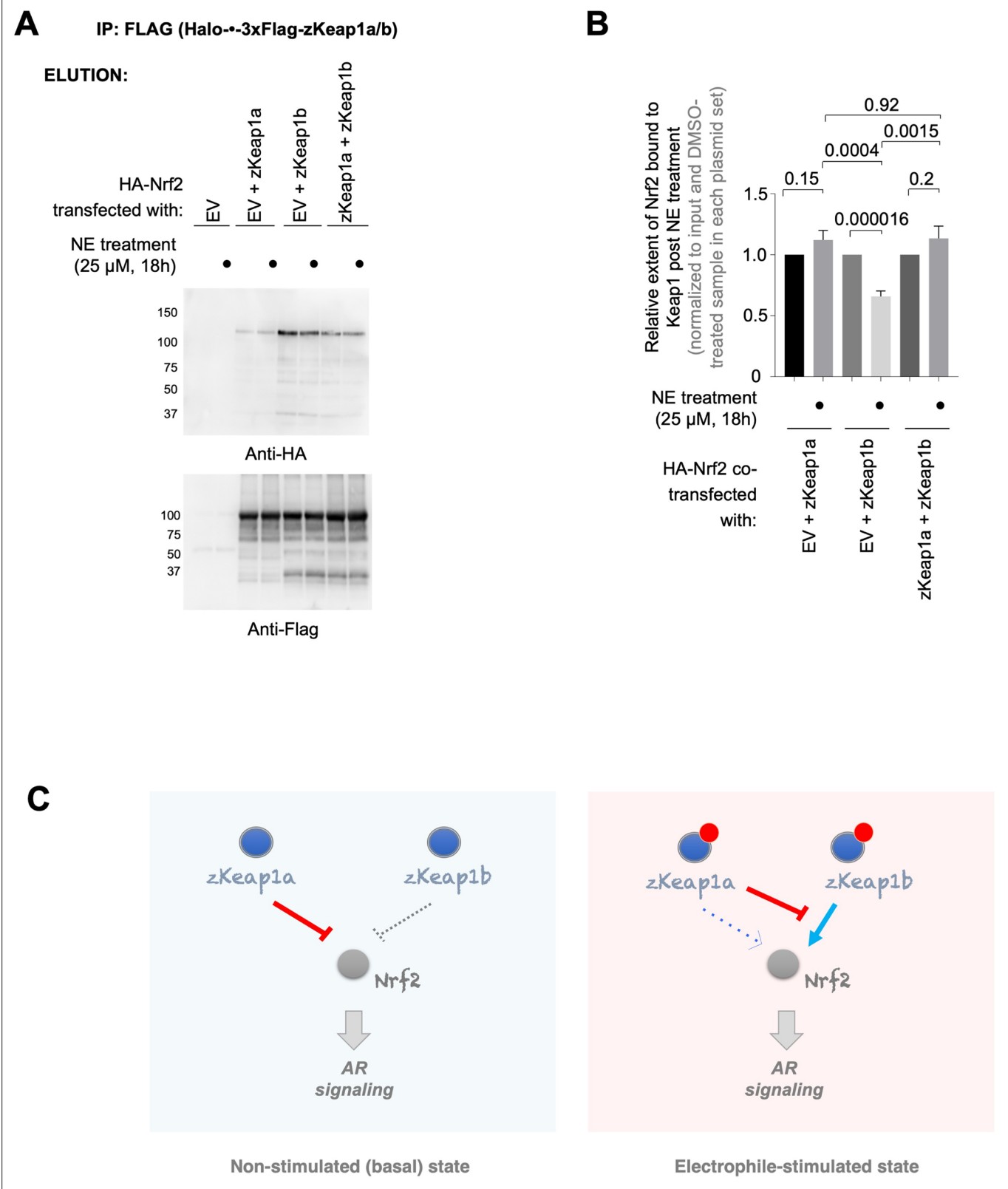

**Figure 8.** zKeap1a/b paralog-specific AR regulation is reflected by the differences in altered Nrf2 binding following electrophile stimulation. (**A**) HEK293T cells were transfected with a plasmid encoding HA-Nrf2 [used because anti-HA-antibody is orthogonal to our anti-FLAG-antibody; and because this anti-HA-antibody (for detecting HA-Nrf2) is of higher sensitivity, compared to anti-myc-antibody to myc-Nrf2], and (an)other plasmid(s) encoding: either a mix of Halo-•–3xFlag-zKeap1a and Halo-•–3xFlag-zKeap1b, Halo-•–3xFlag-zKeap1a and empty vector (EV), Halo-•–3xFlag-zKeap1b

*Figure 8 continued on next page*

*Figure 8 continued*

and EV, or EV alone. The plasmid amount of HA-Nrf2 was 50% in all co-transfection conditions, and the rest of the co-transfected plasmids made up the other 50% (with equal 1:1 or 1:1:1 contribution from each plasmid, as applicable). Following whole-cell NE treatment (25 µM, 18 hr), normalized cell lysates were treated with anti-Flag M2 resin to evaluate the relative extent of association between zKeap1a/b and Nrf2 following NE stimulation. Representative blots for Elution. The band around ~37 kDa in anti-Flag blot, although of unknown identity, is present almost equally in both 'zKeap1b' and 'zKeap1b+zkeap1 a' samples, and thus its presence cannot be sufficient to explain the differences observed between these two data sets. See *Figure 8—figure supplement 1* for the corresponding Input. (**B**) Quantification of (**A**) normalized over input (see *Figure 8—figure supplement 1*) and corresponding DMSO-treated samples in each set. Right panel: Quantification for Nrf2 association to zKeap1 upon NE bolus treatment for different ratios of zKeap1a:zKeap1b. (n=6 biological replicates). (**C**) Proposed model illustrating paralog-specific nuanced regulation of cellular antioxidant response (AR) under steady-state vs. electrophile-stimulated conditions. *Left panel:* under non-electrophile-stimulated conditions, zKeap1a is a more effective antagonist of Nrf2 (and hence, AR-signaling) than zKeap1b. *Right:* following electrophile stimulation, zKeap1b-modification results in a large upregulation in AR through significantly-reduced binding of Nrf2. By contrast, zKeap1a-modification gives rise to a weaker AR-upregulation, and zKeap1a—likely in the electrophile-modified or -non-modified state—functions as a negative regulator to suppress Nrf2/AR-pathway activation promoted by modified-zKeap1b. See also Figure 8A–B and *Figure 8—figure supplement 1*. All numerical data present mean ± sem. Numbers above the bars represent analysis by two-tailed *t*-tests.

The online version of this article includes the following source data and figure supplement(s) for figure 8:

**Source data 1.** Full view blot image.

**Source data 2.** Raw blot image.

**Source data 3.** Quantification results.

**Figure supplement 1.** IP assays investigating differential association of zKeap1a and zKeap1b to Nrf2, following electrophile stimulation (see also *Figure 8A–B*).

**Figure supplement 1—source data 1.** Full view blot image.

**Figure supplement 1—source data 2.** Raw blot image.

LDEs and drugs with Michael-acceptor electrophilic appendages affect numerous redox-dependent pathways *Long and Aye, 2017a*.

The setup is straightforward: the HaloTagged POI is expressed using mRNA injection; this functional fusion protein is expressed at close-to-endogenous levels; a bioinert photocaged compound is added to the fish water, and can be washed away prior to light-triggered electrophile-targeting at a user-prescribed time; and readouts are typically simple and required 10–30 fish. Although mRNA injection does have its detractors, we have found robust, reliable, and ubiquitous expression of Halo-POI can be readily obtained using this method up to at least 36 hpf, and is practically suited to studying impacts of transient protein-expression on ephemeral redox-/stress-dependent signaling responses. Furthermore, many POIs can be studied simply, without the need for tedious genetic manipulation steps. Such adaptability/simplicity is paramount for studying potential off-target proteins identified in large-scale screening assays, or for screening large numbers of compounds that could have numerous on- and off-the-path targets.

We also uncovered new insights into AR-regulation orchestrated by the different zKeap1-paralogs present in fish. These findings were derived from transient knockdowns, and Z-REX experiments that highlighted differences in responsivity to electrophiles in the head and the tail of the fish. Morphant data tied this difference to expression of zKeap1a, which we identified as a negative regulator of electrophile-induced AR. Intriguingly, our data in fish using Z-REX implied off the bat that this aspect was a dominant effect, as the in-trans expression of hKeap1 and its specific targeting was not sufficient to bypass this regulation. These results imply that Z-REX will be useful to identify such behaviors in other systems, a finding that complements the general ability of REX technologies to identify dominant electrophile-signaling events. We progressed to evaluate zKeap1a and zKeap1b function in cultured cells, where we found that expression of zKeap1a or a mixture of zKeap1a/zKeap1b could not upregulate AR following electrophile treatment. We point out that such systems are indeed apposite for study by this combination of fish and ectopic expression in human cells. This is because of the control offered by ectopic expression, and because of the overall dominant-negative effects conferred by the zKeap1a paralog. The latter render interpretation of data derived from MO rescue, particularly in a tissue-specific manner difficult to interpret.

In cells expressing zKeap1b, although this protein elicited effectively lower overall suppression of basal AR than zKeap1a, AR-upregulation upon electrophile treatment was robust. The divergent electrophile responses of zKeap1a and zKeap1b are entirely consistent with a previous report in

zebrafish *Kobayashi et al., 2009*, implying that these proteins function appropriately in cell culture. Furthermore, a single point mutation in hKeap1, to match a cysteine mutation present in zKeap1a but that remains a cysteine in zKeap1b, can recapitulate most of the zKeap1a-associated cell responses. Finally, T-REX—which allows for the most stringent and direct measurement of electrophile sensitivity known *Parvez et al., 2016*—showed that despite the disparate capabilities in basal vs. electrophile-stimulated AR-regulation, all Keap1-variants exhibit similar electrophile sensitivity. Given that we have previously published that single- or double-point mutations of LDE-sensing cysteines in Keap1 have little impact on electrophile sensing by Keap1 (*Parvez et al., 2015*; *Lin et al., 2015*), this result is unsurprising. On the contrary, the inability of zKeap1a to respond to electrophiles, and even zKeap1a's activity in the wake of the poor AR-suppressing activity shown by hKeap1(C273I) mutant, warrant more investigations in further work. Nevertheless, the electrophile-signal propagation programs associated with zKeap1b-modification (which promotes AR-signaling) are resisted by zKeap1a functioning as an overall negative-regulator of electrophile-stimulated AR-upregulation (*Figure 8C*), as opposed to differences in electrophile occupancy/sensing efficiencies influencing these behaviors.

To investigate this matter further, we showed that there are subtle differences in the way zKeap1a and zKeap1b function upon electrophile treatment. Whereas zKeap1a does not undergo net release of Nrf2 upon electrophile treatment, and further does not accumulate a large amount of Nrf2 in the steady-state prior to electrophile treatment, zKeap1b net relinquishes around 40% bound-Nrf2 upon electrophile treatment, and accrues a large amount of bound-Nrf2 in the basal state prior to electrophile treatment. The mixture of zKeap1a/zKeap1b also does not undergo net release of Nrf2 upon electrophile treatment, although it can still accrue substantial bound-Nrf2 in the state prior to electrophile treatment. These data allow rationalization of our results both from zebrafish and human cell culture, and favor a model in which decrease in affinity of electrophile-modified zKeap1b for Nrf2 is a means to upregulate AR in response to electrophilic stress. It is likely that such a mode of action leads to release of bound-Nrf2 from zKeap1b upon electrophile modification, given that turnover of Nrf2 on zKeap1b is slow [or otherwise, build-up of Nrf2 would not occur upon zKeap1b overexpression (just as it does not occur on zKeap1a)], and generally AR-upregulation is observed even at low-electrophile occupancy on Keap1[2,5-6]. Inhibition of rebinding of Nrf2 post dissociation, and inhibition of newly-synthesized Nrf2 binding to zKeap1b may also contribute to AR increase in such circumstances, as binding also contributes to zKeap1b–Nrf2 affinity. The contribution of zKeap1b re(binding) to Nrf2 to AR-upregulation vis-à-vis the contribution of release of bound-Nrf2 is difficult to parse, and indeed beyond the scope of this paper. Of course, other potential/synergistic mechanisms—such as inhibition of zKeap1b-promoted Nrf2 degradation—could occur *in tandem*. But the comparison of zKeap1a/zKeap1b and zKeap1b systems argues in favor of net release being the key component of AR upregulation.

There are further *potential* complications in data interpretation due to there being three possible zKeap1 dimeric forms (ignoring higher order structures) in the zKeap1a/zKeap1b-mixed system. However, an appreciable amount of Nrf2 is built up on zKeap1 in the zKeap1a/Keap1b system (unlike upon expression of zKeap1a alone), and *no* release of Nrf2 was observed upon NE treatment (unlike upon expression of zKeap1b alone). Thus, zKeap1a exerts a significant direct effect on how zKeap1b responds to electrophiles, and hence the heterodimer, or higher order state(s) containing both proteins, must be a significant component of the zKeap1 present in the assay.

We further note that a non-inconsiderable amount of Nrf2 builds up on Keap1 in human cells, as evidenced by primarily-cytosol-localized Keap1 promoting cytosolic Nrf2 accmulation (*Parvez et al., 2016*; *Parvez et al., 2015*; *Zhang and Hannink, 2003*). Such a mechanism further helps reconcile why relatively low electrophile occupancy on Keap1 is able to trigger large AR upregulation.

These studies further underscore subtle regulatory roles of lipid-derived electrophilic metabolites *Parvez et al., 2018* and applications of advanced chemical biology techniques *Long et al., 2020* in a model organism that enable nuanced target-specific electrophile-regulatory behaviors to be unmasked. They have unearthed interesting aspects of paralog-specific diversion and interplay, which continue to be of interest to the zebrafish community and to evolutionary biologists as a whole.

**Key resources table**

| Reagent type (species) or resource | Designation | Source or reference | Identifiers | Additional information |
|---|---|---|---|---|
| Strain, strain background (*Escherichia coli*) | XL10-Gold Ultracompetent Cells | Agilent Technologies | Cat# 200315 | N/A |
| Cell line (*Homo-sapiens*) | 293T (ATCC CRL-3216) | ATCC | Cat# CRL-3216, RRID:CVCL_0063 | https://www.atcc.org/products/crl-3216 |
| Genetic reagent (*Danio rerio*) | *Tg(–3.5gstp1:GFP)/it416b* | Japanese National BioResource Project | N/A | https://shigen.nig.ac.jp/zebra/index_en.html |
| Genetic reagent (*Danio rerio*) | Brian's wildtype | Professor Joseph Fetcho's lab (Cornell University) | N/A | A strain of wt-zebrafish that shows low pigmentation |
| Genetic reagent (*Danio rerio*) | Wild-type line *Casper* | Professor Joseph Fetcho's lab (Cornell University) | PMCID:PMC2292119 | N/A |
| Recombinant DNA reagent | pGL4.37[luc2P/ARE/Hygro] | Promega | Cat# E3641 | N/A |
| Recombinant DNA reagent | pGL4.75[hRluc/CMV] | Promega | Cat# E6931 | N/A |
| Recombinant DNA reagent | pCDNA3 myc-Nrf2 | Addgene | Cat# 21555, RRID:Addgene_21555 | N/A |
| Recombinant DNA reagent | pCS2 +8 vector | Addgene | Cat# 34931, RRID:Addgene_34931 | N/A |
| Recombinant DNA reagent | pFN21a Halo-TEV-Keap1 | Halo ORFeome, Promega Kazusa collection | Cat# FHC00420 | N/A |
| Antibody | Anti-GFP (FITC) (goat polyclonal) | Abcam | Cat# ab6662, RRID: AB_305635 | IF (1:1000) |
| Antibody | Anti-Flag (mouse monoclonal) | Sigma | Cat# F3165, RRID:AB_259529 | WB (1:4000) |
| Antibody | Anti-Keap1 (mouse monoclonal) | Abcam | Cat# ab119403, RRID: AB_10903761 | IF (1:500) |
| Antibody | Anti-HaloTag (rabbit polyclonal) | Promega | Cat# G9281, RRID:AB_713650 | WB (1:3000) |
| Antibody | Anti-HA HRP (mouse monoclonal) | Sigma | Cat# H3663, RRID: AB_262051 | IF and WB (1:500) |
| Antibody | Anti-β-actin HRP (mouse monoclonal) | Sigma | Cat# A3854, RRID: AB_262011 | WB (1:30,000) |
| Antibody | Anti-mouse-HRP (horse polyclonal) | Cell Signaling | Cat# 7076, RRID:AB_330924 | WB (1:2000) |
| Antibody | Anti-rabbit-HRP (goat polyclonal) | Cell Signaling | Cat# 7074, RRID:AB_2099233 | WB (1:2000) |
| Antibody | Anti-goat IgG H&L (AlexaFluor568) (rabbit polyclonal) | Abcam | Cat# ab175707, RRID: AB_2923275 | IF (1:2000) |
| Antibody | Anti-rat IgG H&L (AlexaFluor568) (donkey polyclonal) | Abcam | Cat# ab175475, RRID: AB_2636887 | IF (1:500) |
| Antibody | Anti-mouse IgG H&L (AlexaFluor568) (donkey polyclonal) | Abcam | Cat# ab175472, RRID: AB_2636996 | IF (1:1000) |
| Antibody | Anti-mouse IgG H&L (AlexaFluor647) (donkey polyclonal) | Abcam | Cat# ab150107, RRID:AB_2890037 | IF (1:1000) |

For additional information of other resources, See Appendix for detailed information on: Sequences of cDNAs, cloning primers, qRT-PCR primers, morpholinos (MOs) and primers for splice-blocking MO validation.

## Statistical analysis and reporting

Wherever applicable, figure legends contain information pertaining to SEM with associated P values, sample size (e.g. number of fish embryos analyzed, number of independent biological replicates).

Representative raw images are included with accompanying quantitation where relevant. Figure legends contain description of independent biological replicates, vs. technical replicates. Outliers were maintained in all data sets with error bars designating SEM and p-values from application of two-tailed Students' t-test included. Figure legends contain information pertaining to the following: the exact sample size (n) for each experimental group/condition, given as a discrete number and unit of measurement; whether measurements were taken from distinct samples or whether the same sample was measured repeatedly; null hypothesis testing (two-tailed Students' t-test) with exact p values noted whenever suitable. No statistical methods were used to pre-determine sample size. Size of datasets was chosen according to literature and based on our own experience, integrating similar methods of analysis. Number of technical replicates and biological replicates are reported in figure legends. Summary information related to sample allocation/handling is detailed in the supplementary text.Briefly, no masking was used. Prior to beginning each experiment, cells/embryos were allocated into groups randomly, for each sample group. When an experiment was commenced, groups of cells/embryos were allocated into treatment groups without pattern or bias. This ensured that each treatment group in an experiment was identical to account for any variation across cells/fish breeding. Cell counting was performed at each step of the experiment whenever relevant, to rigorously standardize conditions both within each experiment and across different experiments. Casper strain zebrafish, wild-type zebrafish, and previously-validated reporter strains were used for the experiments involving embryos and for the latter transgenic reporter strains, at consistent zygosity. As zebrafish are believed to exhibit polygenic sex determination, at the age at which experiments were undertaken, the sex of the fish was unable to be determined, but likely account for 50:50 male: female.

## Materials and methods

LDEs and photocaged LDEs all contain alkyne-functionalization and were synthesized in house as previously described. All primers were from IDT. Phusion HotStart II polymerase was from Thermo Scientific. All restriction enzymes were from NEB. pFN21a Halo-TEV-Keap1 (Kazusa Collection) was from Promega. Trizol RNA purification kit, RnaseZap RNA decontamination solution, DNaseI AMP grade, and Superscript III Reverse transcriptase were from ThermoFisher Scientific. iQ SYBR Green Supermix was from BioRad. Complete EDTA free protease inhibitor was from Roche. 1 X Bradford dye was from BioRad. Photocaged precursors and the corresponding uncaged LDEs were synthesized as described previously *Long et al., 2017b*; *Zhao et al., 2018*; *Surya et al., 2018*. Cy5 azide and Cu(TBTA) were from Lumiprobe. Dithiothreitol (DTT) and TCEP-HCl were from Goldbio Biotechnology. Streptavidin sepharose beads were from GE Healthcare. Bovine Serum Albumin (BSA) powder was from Thermo Scientific. CYP inhibitor, PF-4981517, was from ApexBio. Bardoxolone methyl (CDDO-Me), and dimethyl fumarate (Tecfidera) were from Selleckchem. Zirconia beads were from Biospec. Glass beads (150–200 μm) were from Sigma. Biotin-dPEG11-azide was from Quanta Biodesign. Lithium dodecyl sulfate (LDS) was from Chem-Impex. Sulforaphane was from Santa Cruz Biotech. Tryspin inhibitor from Glycine max and all other chemicals were from Sigma. The empty pCS2 +8 vector (Addgene #34931) was from Addgene. Morpholinos were synthesized by Gene Tools, LLC. 365 nm UV lamp was from Spectroline (XX-15N) for T-REX experiments in fish (Z-REX), 365 nm UV was from Camag (022.9160) for T-REX experiments in cells. For T-REX experiments, the lamps were positioned above zebrafish embryos in six-well plates such that the power of UV irradiation was ~5 mW/ cm$^2$ [as measured by a hand-held power sensor Spectroline, XDS-1000] and ~3 mW/cm$^2$ for cells. For all imaging experiments, a Leica M205-FA microscope equipped with a stereomicroscope was used, aside from *Figure 6A* that was imaged on a LSM710 (Zeiss). Quantitation of fluorescence intensity was performed using Image-J software (NIH, version 1.50 g). In-gel fluorescence analysis and imaging of western blots and Coomassie stained gel were performed using BioRad Chemi-Doc MP Imaging system. Densitometric quantitation was performed using Image-J (NIH). Cy5 excitation source was epi illumination and 695/55 emission filter was used. Quantitative PCR (qRT-PCR) was performed using Light Cycler 480 instrument (Roche Life Sciences). Anti-Flag M2 affinity agarose gel was from Sigma-Aldrich (A2220). Primer information and antibodies used are listed in **Appendix**.

## Plasmids

Plasmids to express (His$_6$)-Halo-TEV-Keap1-(2xHA) (hereafter, Halo-•-Keap1) and (His$_6$)-Halo-(2xHA)-P2A-TEV-Keap1-(2xHA) (hereafter, Halo-P2A-•-Keap1) in pCS2 +8 vectors were cloned using ligase-independent cloning method using primers specified in **Appendix**. Briefly, Halo-TEV-Keap1 was amplified from pFN21a Halo-TEV-Keap1 (Halo ORFeome, Promega Kazusa collection) using fwd1 and rev1 primers. The amplified product was then extended using fwd ext1 and rev ext1. An additional extension step was performed using the product from the first extension step as the template and using primers fwd ext2 and rev ext2 to generate the 'megaprimer'. The 'megaprimer' was inserted in empty pCS2 +8 vector linearized using EcoR1.

To generate Halo-P2A-•-Keap1, Halo and Tev-Keap1 were amplified separately from pFN21a Halo-TEV-Keap1 using primers fwd1 and rev1, and fwd1' and rev1', respectively. The amplified products were then extended using fwd ext1 and rev ext1, and fwd ext1' and rev ext1'. The two fragments were joined using PCR and amplified using another set of primers, fwd ext2 and rev ext2, to generate the 'megaprimer'. The 'megaprimer' was inserted in empty pCS2 +8 vector linearized using EcoR1.

To generate zKeap1a and zKeap1b in mammalian expression vectors, RNA from wild-type zebrafish was isolated and used to generate cDNA using an oligo(dT) primer. From the cDNA, primers designed to overlap with the zKeap1a/b sequences were used to amplify zKeap1a and zKeap1b. The primers also contained an overlapping region with the pCS2 +8 backbone. The empty pCS2 +8 plasmid was cut with EcoRI-HF enzyme and the zKeap1a/b PCR product with overlapping pCS2 +8 backbone was inserted into the vector. To obtain HaloTagged zKeap1a/b, the zKeap1a/b plasmids were cut with SpeI and a 'megaprimer' PCR sequence encoding Halo-TEV-Flag (3 x) was inserted to form the full construct: Halo-TEV-Flag3-zKeap1a/b, referred to as Halo-•-(3xF)-zKeap1a/b. The amino acid sequence for zKeap1a and zKeap1b from our fish matched the sequences reported in uniprot. Note: additional synonymous mutations were introduced in pCS2 +8 Halo-•–3xFlag-zKeap1a to prevent knockdown interference in the course of rescue experiments where embryos were injected with both Keap1a-ATG-MO MO and Halo-•–3xFlag-zKeap1a mRNA. The stop codon was added separately by site-directed mutagenesis. All primers used are listed in **Appendix**.

Successful inserts for each plasmid were identified with colony PCR. 'Hit' colonies were grown overnight in LB-AMP medium and purified using a miniprep kit (Bio Basic, BS614). The insert was fully sequenced at the Cornell Biotechnology Resource Center Genomics facility and Microsynth AG (Switzerland).

## Fish husbandry and crossing

All procedures performed at Cornell (2017–2018) and EPFL (2018-present) conform to the animal care, maintenance, and experimentation procedures followed by Cornell University's and EPFL's Institutional Animal Care and Use Committee (IACUC) guidelines and approved by the respective institutional committees. All experiments with zebrafish performed at EPFL (2018-present) have been performed in accordance with the Swiss regulations on Animal Experimentation (Animal Welfare Act SR 455 and Animal Welfare Ordinance SR 455.1), in the EPFL zebrafish unit, cantonal veterinary authorization VD-H23. Either Casper strain or *Tg(–3.5gstp1:GFP)/it416b* reporter [hereafter *Tg(gstp1:GFP)*] *Danio rerio* (zebrafish) were used for all experiments. Casper strain was a kind gift from the Fetcho Lab (Cornell University). *Tg(gstp1:GFP)* fish was from the Japanese National BioResource Project. Since zebrafish are believed to exhibit polygenic sex determination, at the age at which experiments were undertaken, sex of the fish was unable to be determined, although our samples are likely roughly equal mixtures of males and females. Animals were maintained and embryos were obtained according to standard fish husbandry procedures. For *Tg(gstp1:GFP)* fish, crosses were set between a homozygous transgenic parent and a WT parent such that the resulting progeny were all heterozygous for the reporter gene.

## Fish injection and Z-REX

For injection in fish embryos, mRNA for Halo-•-Keap1 and Halo-P2A-•-Keap1 was generated. First, the desired genes were amplified using RNA-fwd and RNA-rev primers (**Appendix**). mRNA was generated using an mMessage mMachine SP6 in vitro transcription kit (Ambion, AM1340) as per manufacturer's protocol except the reaction was scaled up for two preps.

Fertilized eggs at the one- to four-cell stage were injected with 2 nl mRNA (1.3–1.6 mg/ml) and/ or morpholino (2.7 mg/mL, approximately 5 ng of MO per fish) into the yolk sac. Immediately after injection, embryos were pooled, and separated into two petri dishes (10 cm) filled with 30 mL 10% Hank's salt solution with methylene blue and penicillin (100 U/ml) / streptomycin (100 µg/ml). To one set was added the photocaged precursor to designated LDE at a final concentration of 6 µM and to the other equal volume of DMSO in the dark. Embryos were maintained at 28 °C in the dark for 28 hr after which time fish larva were washed in 10% Hank's solution with no methylene blue/antibiotic (3 times for 30 min each). Larvae were moved to six-well plates. Half of the larvae (Ht-PreLDE-treated or -untreated) were exposed to light for 4 min, and the other half of each set was not. For bolus dosing, treatment of larvae with LDEs was staggered such that the harvest time was the same for all samples including larvae that underwent Z-REX (34 hpf). Further experiments were performed as illustrated in **Appendix**-Scheme 1 and described elsewhere in the supplementary methods, and main and supporting figure legends.

## qRT-PCR

All qPCR experiments were performed in casper strain. 2 hr post light illumination or bolus LDE treatment, 12–15 larvae per sample were euthanized by chilling, dechorionated, the head and tail separated using sharp forceps (11252–40 Dumont #5 Forceps - Biologie/Titanium), and transferred to separate Eppendorf tubes. The samples were washed twice with ice-cold PBS and homogenized in 1 mL Trizol (ThermoFisher Scientific, 15596018) together with vortexing with glass beads for 2 min. Total RNA was extracted per manufacturer's protocol. Glycoblue (ThermoFisher Scientific, AM9516) was used for visualization of the RNA pellet. Around 600 ng of total RNA was treated with AMP grade DNaseI (ThermoFisher Scientific, 18068015), reverse transcribed using Superscript III reverse tran-scriptase (ThermoFisher Scientific, 18080085) per manufacturer's instruction. qRT-PCR was performed for the indicated genes using primers specified in **Appendix**. All primers were validated as previously reported. qRT-PCR analysis was performed with iQ SYBR Green Supermix (Bio-Rad, 170–8880) on a Light Cycler 480 instrument (Roche). In a total volume of 10 µL the PCR reaction mix contained, in final concentrations, 1 X iQ SYBR Green Supermix, 0.30 µM each of the forward and reverse primers and 10–13 ng of template cDNA. The qPCR program was set for 3 min at 95 °C followed by 40-repeat cycles comprising heating at 95 °C for 10 s and at 55 °C for 10 s. The expected products were of ~100–130 bp in size. The data was analyzed using $\Delta\Delta C_t$ method and presented relative to zebrafish actin, β2.

## Immunofluorescence

To assess AR upregulation in *Tg*(*gstp1:GFP*) fish, larvae 4 hr post light illumination or bolus LDE treat-ment were dechorionated, washed twice in ice-cold PBS and fixed in 4% paraformaldehyde in 1 X PBS (Gibco 14190169) for at least overnight with gentle rocking at 4 °C. Fixed larvae were perme-abilized with chilled methanol at –20 °C for 4 hr–overnight. Fish were then washed 2 times with PBS-0.1%Tween-1% DMSO for 30 min each with gentle rocking, then blocked in PBS-0.1%Tween containing 2% BSA and 10% FBS, then stained with anti-GFP FITC conjugated (Abcam, ab6662) primary antibody overnight at 4 °C in blocking buffer. Subsequently, the larvae were washed twice (30 min each wash), re-blocked for 1 hr at room temperature, and incubated with the AlexaFluor 568-conjugated fluorescent secondary antibodies (Abcam, ab175707) in blocking buffer for 1.5 hr at room temperature with gentle rocking, and then washed three times. Fish were imaged on 2% agarose plates on a Leica M205-FA equipped with a stereomicroscope. Quantitation of IF data was performed using ImageJ/FIJI (NIH).

To assess protein expression in zebrafish, larvae were fixed at 34 hours post fertilization (hpf) after dechorionation. Permeabilization and immunostaining protocols are as above except antibodies to the desired protein/tag and appropriate secondary antibodies were used.

## Click coupling in whole fish

Casper zebrafish expressing Halo-•-Keap1, and either treated with the photocaged precursor (6 µM, overnight) or bolus treatment (2 hr) with the indicated LDE, were dechorionated at 34 hpf and fixed in 4% PFA for at least overnight with gentle rocking at 4 °C. Where relevant, PF-4981517 (1 or 5 µM) was added 4 hr prior to LDE treatment. Fish were then permeated in methanol (100%) at –20 °C for

at least 24 hr. Fish were then washed twice in PBS and two times in Hepes (50 mM, pH 7.6) for 30 min each (Note: Fish tend to float after MeOH wash. Allow them to settle prior to manipulation/washing). Fish were then exposed to a cocktail containing (in final concentration): 50 mM Hepes (pH 7.6), t-BuOH (5%), $CuSO_4$ (1.1 mM), sodium ascorbate (10 mM; made as a 100 mM stock in 500 mM Hepes (pH 7.6) with no further pH adjustment), and Cy5-azide (10 µM). This was shaken at room temperature for 1 h, then washed three times in 1 x PBS with 0.015% Tween-20. After third wash, fish were incubated overnight at 4 °C, then fresh 1 x PBS with 0.015% Tween-20 was added and fish were imaged. Fixed fish can stick to plastic. Tween helps to reduce this problem.

## Click coupling and enrichment of modified proteins

Casper zebrafish expressing Halo-•-Keap1 were treated with either the photocaged precursor or bolus treatment with the indicated LDEs (~120 per condition). Photocaged precursors were added to the fish water after injection of Halo-•-Keap1 mRNA at a final concentration of 6 µM and Z-REX was performed as specified above. Bolus dosing was done for 2 hr. Immediately after Z-REX, larvae were dechorionated and deyolked manually at 4 °C, washed twice with cold PBS to remove yolk proteins, and washed once with cold 50 mM Hepes (pH 7.6). The zebrafish pellet was flash frozen in liquid nitrogen and stored at −80 °C until lysis. Fish pellet was resuspended in 50 mM Hepes (pH 7.6), 1% Triton X-100, 0.1 mg/ml soybean trypsin inhibitor, and 2 X Roche protease inhibitor. Lysis was performed by vortexing with Zirconia beads for 20 s and subsequent 3 times freeze-thaw. Lysate protein was collected after centrifugation at 21,000×g for 10 min, and concentration determined using Bradford dye relative to BSA standard. 30–50 µg of the lysate protein was removed, quenched with Laemmli buffer and saved as input. The remaining lysate was diluted to 1 mg/ml with 50 mM Hepes (pH 7.6) and 0.2 mM TCEP, TEV protease was added at a final concentration of 0.2 mg/ml, and the sample incubated at 37 °C for 30 min. Next, 5% t-BuOH was added to the sample. A stock solution containing 10% SDS, 10 mM $CuSO_4$, 1 mM Cu-TBTA, 1 mM biotin-azide and 20 mM TCEP (made as a 100 mM stock in 500 mM HEPES pH 7.5) was prepared and added to the sample such that the final concentration are as follows: 1% SDS, 1 mM $CuSO_4$, 0.1 mM Cu-TBTA, 0.1 mM biotin-azide, and 2 mM TCEP. The mixture was mixed thoroughly and incubated at 37 °C for 15 min, after which another 1 mM TCEP was added, mixed, and the sample incubated for additional 15 min. Protein precipitation was performed by adding EtOH (prechilled at −20 °C) at a final concentration of 75% (v/v), vortexing the sample, and incubating at −80 °C for at least overnight. Precipitated protein was collected by centrifugation at 21,000×g at 4 °C for 2 hr, washed twice with prechilled EtOH (twice), once with 75% EtOH in water, and an additional wash with prechilled acetone. Precipitate was air-dried and subsequently redissolved in 8% LDS in 50 mM HEPES (pH 7.6), 1 mM EDTA by sonication at 50 °C and vortexing. The solubilized lysate protein was collected following centrifugation and diluted in 50 mM Hepes (pH 7.6) to give a final concentration of 0.5% LDS. The sample was added to pre-washed streptavidin high-capacity resin and incubated at 4–6 hr at rt. The supernatant was removed following a low-speed centrifugation (1000×g), and the beads washed thrice with 50 mM Hepes (pH 7.6) containing 0.5% LDS. Bound proteins were eluted by boiling beads in 2 x Laemmli buffer with 6% βME at 98 °C. Samples were analyzed using SDS-PAGE followed by western blot as specified below.

## SDS-PAGE and western blot

Up to 30 µl of input or elution samples were separated on a 10% polyacrylamide gel using SDS-PAGE. The gel was subsequently transferred to a PVDF membrane at 4 °C in ice-cold transfer buffer containing 25 mM Tris, 192 mM Glycine, and 15% Methanol (v/v). Membrane was blocked in 5% milk for 2 hr at rt, incubated with primary antibody in 1% milk for 5 hr at rt, washed three times with Tris Buffer Saline (100 mM Tris, pH 7.6, 150 mM NaCl) containing 0.2% Tween-20 (TBST). Where applicable, the membrane was incubated with secondary antibody in 1% milk for 5 hr at rt, washed twice with TBST, followed by an additional wash with TBS. Pierce ECL western blotting substrate was used for detection of the desired protein bands.

## Data quantitation and analysis

Imaging data was quantitated using ImageJ (NIH). For assessing AR upregulation in *Tg*(*gstp1:GFP*) fish, the area around the head (excluding the eyes), the tail (median fin fold), or the whole fish (excluding the yolk sac) were selected using freeform selection tool. Corresponding illustrations are included

in each sub-figure for clarity. For IF, the mean red fluorescence intensity of the selected region was measured and subtracted from the mean background fluorescence intensity (region with no fish). For live fish imaging, the mean GFP fluorescence intensity of the selected region was measured and subtracted from the mean background fluorescence intensity (non-transgenic fish).

The mean value for the control group was calculated from the raw, background-subtracted, values within that control group. Then all raw values were divided by the mean for the control. n for imaging experiments represent the number of single cells or fish embryos quantified from at least 7–8 fields of view with controls (empty vector controls for ectopically-overexpressed proteins, shRNA knockdown cell controls for endogenous proteins) shown in the figures.

Unless specified, all *t* tests were two-tailed analysis. n for western blot/gels, qRT-PCR, and luciferase assays represents the number of lanes on western blots/gels under identical experimental conditions and each lane is from a separate individual replicate, no. of independent biological replicates as indicated in the figure legends.

## Cell culture

HEK293T cells (obtained from ATCC) were cultured in complete 10% FBS medium (MEM Glutamax, 1 X sodium pyruvate, 1 X penicillin streptomycin, 1 X MEM NEAA, 10% FBS). Cells were grown at 37 °C with 5% $CO_2$ in a humidified incubator. Cell lines were verified to be free of mycoplasma contamination by Venor GeM Mycoplasma Detection Kit from Sigma.

## AR reporter screen bolus dosing

HEK293T cells were seeded in 48 (or 96) well plates (cell density was $0.25 \times 10^6$ cells/mL, 0.3 mL (or 0.1 mL) cells per well). Cells were transfected with 100 (or 33) ng pCDNA3 myc-Nrf2, 100 (or 33) ng luciferase plasmid mix (20:1 Firefly luciferase ARE promoter:Renilla CMV promoter), and 100 (or 33) ng total of either empty pCS2 +8, WT Keap1 (hKeap1, zKeap1a, or zKeap1b), or hKeap1 C273I. Transfection was carried out using Mirus TransIT-2020 transfection reagent. After 24 hr of transfection, the medium was removed and replaced with rinse medium (MEM glutamax, 1 X sodium pyruvate, 1 X penicillin streptomycin, 1 X MEM NEAA) containing 25 µM NE or HNE or corresponding volume of DMSO. Cells were incubated for a further 18 hr, at which point they were lysed in 65 (or 30) µL passive lysis buffer by shaking at room temperature for 25 min. 25 µL of sample was transferred to a 96-well opaque white plate for reading, and an additional 30 µL of sample was mixed with 10 µL of 4 X Laemmli dye with BME to run on a gel to confirm expression levels of the variants of Keap1. A BioTek Cytation3 or a Perkin Elmer 2300 microplate reader was used to perform Firefly and *Renilla* luciferase activity readings as previously reported *Zhao et al., 2018*; *Surya et al., 2018*.

## AR reporter assay following T-REX

Procedure was followed as above, except T-REX was performed as previously published after 24 hr of transfection and the cells were lysed 18 hr post light *Zhao et al., 2018*; *Surya et al., 2018*. Briefly, after 24 hr transfection, 10 µM Ht-PreHNE was added to the cells in rinse medium (or corresponding amount of DMSO). After 2 hr incubation in the dark, the medium was removed, and the cells rinsed three times with rinse medium, half an hour for each rinse. After the last rinse, the cells were exposed to UV light (~3 mW/cm²) for 5 min and returned to the incubator for 18 hr before being lysed and read as described above.

## T-REX electrophile-labeling assay in cells

HEK293T cells were transfected to express Halo-TEV-hKeap1, Halo-TEV-hKeap1 (C273I), or Halo-TEV-zKeap1a/b. After 36 hr of transfection, 25 µM of PreHNE or PreNE was introduced. After 2 hr of incubation in the dark, the cells were rinsed three times, 30 min each. The cells were exposed to light (~5 mW/cm²) for 10 min, incubated for a further 5 min, and then harvested. Click reactions with Cy5-azide were performed as previously published *Zhao et al., 2018*; *Long et al., 2019*.

## MO validation – translation inhibition reporter assay

Plasmids were generated that contained the binding site of the zKeap1a or zKeap1b ATG morpholino (the start codon and approximately 20 following bases) upstream of the gene encoding Firefly Luciferase (with a Gly-Ser linker). mRNA was generated using mMessage mMachine SP6 in vitro transcription

kit. To test the translation blocking ability of the MOs, 50 ng of mRNA (0.136 μM) was incubated with random control MO, standard control MO, water, or corresponding zKeap1 MO in indicated ratios for 5 min at room temperature. Then, each sample was added to Rabbit Reticulocyte Lysate (Promega) and incubated at 37 °C for 15 min. The sample was loaded onto a white plate (Corning 3912) and the Firefly Luciferase quantified using standard methods. A decrease in Firefly signal indicated blocked translation.

## MO validation – splice blocking analysis by RT-PCR

The yolk sac of *Tg(gstp1:GFP)* embryos at the one- to four-cell stage was microinjected with approximately 2 nL of splice-blocking (SPL) morpholino oligonucleotides [0.5 mM (GeneTools LLC; sequences in **Appendix**)]. Embryos were grown at 28.5 °C and euthanized at 30 hpf. Euthanized embryos were lysed by vortexing for approximately 30 s in Trizol (Life Technologies) with zirconia beads, and RNA was isolated following the manufacturer's protocol. The quality of the RNA was assessed by Nanodrop spectrophotometry (A260/A280 ratio ~2) and integrity was assessed by agarose gel electrophoresis. 1 μg of total RNA was treated with amplification-grade DNase I (NEB) and reverse transcribed using Oligo(dT)$_{20}$ as a primer and Superscript III Reverse Transcriptase (Life Technologies) following the manufacturer's protocols. The resulting cDNA was used as a template in PCR reactions with primers flanking the sites targeted by the splice-blocking MOs (**Appendix**). PCR products were resolved on an agarose gel and band intensity was quantitated using the Measure tool of Image-J(NIH).

## MO validation – assessment of relative protein expression levels by whole-mount immunofluorescence (IF)

The yolk sac of *Tg(gstp1:GFP)* embryos at the one- to four-cell stage was microinjected with approximately 2 nL of morpholino oligonucleotides, namely SPL-MOs targeting zKeap1a and zKeap1b [0.5 mM (GeneTools LLC; sequences in **Appendix**)]. Embryos were grown at 28.5 °C and euthanized at 30 hpf. Whole-mount IF was carried out as described above, using anti-Keap1.

## MO validation – zkeap1a/b-mRNA overexpression rescues effects of the zkeap1a/b MOs

Either Approach (**1**) or (**2**) below, was deployed as indicated in figure legends and text discussion: (**1**) The yolk sac of *Tg(gstp1:GFP)* embryos at the one- to four-cell stage was co-microinjected with 2 nL of 0.2 mM zKeap1a-ATG-MO and 250 ng/μL Halo-TEV-zKeap1a mRNA. Embryos were grown at 28.5 °C and treated with 10 μM NE alkyne at 4 hpf for 4 hr before being imaged. (**2**) The yolk sac of *Tg(gstp1:GFP)* embryos at the one- to four-cell stage was co-microinjected with 2 nL of 0.25 mM zKeap1b-ATG-MO and 250 ng/μL Halo-TEV-zKeap1b mRNA. Embryos were grown at 28.5 °C and treated with 20 μM NE alkyne at 30 hpf for 4 hr before image acquisition. Whole-mount IF was carried out as described above, using anti-GFP FITC-conjugated (Abcam, ab6662) primary antibody and AlexaFluor 568-conjugated fluorescent secondary antibody (Abcam, ab175707). **NOTE:** the bolus LDE dosage and MO concentrations deployed were reduced in Approach (1), since the blastula period (4 hpf) was found to be more sensitive to high-LDE/MO-dosage-induced viability loss.

## Anti-flag immunoprecipitation

HEK293T cells (~5–6×10⁶) were seeded in a 10 cm adherent tissue culture plate. After the cells reached 70–80% confluence (~18–24 h), the media were replaced with fresh complete media (8 mL). Cells were transfected with 7.5 μg (total amount) of indicated plasmid(s) using TransIT-2020 transfection reagent (per the manufacturer's recommendation, Mirus). Following 24–36 hr incubation period, the media were changed, and the cells were treated with fresh media containing either 25 μM NE or DMSO, and incubated for a further 18 h. Cells were harvested, pooled, washed twice with ice-chilled 1 X DPBS, and flash-frozen in liquid nitrogen. Cell lysis was performed by first resuspending the cell pellets in 100–200 μL (per 1.5×10⁶ cells) of lysis buffer [containing in final concentrations, 50 mM Hepes (pH 7.6), 150 mM NaCl, 1% Nonidet P-40 and 1 X Roche cOmplete, mini, EDTA-free protease inhibitor cocktail], then by subjecting the resulting cell suspension to rapid freeze-thaw cycles (x3). The lysate was clarified by centrifugation at 18,000 x g for 10 min at 4 °C. Total protein concentration was determined using Bradford assay using BSA as standard (triplicate measurements were made and average value was taken). The lysate was subsequently diluted to 2 mg/mL with binding buffer containing in

final concentrations, 50 mM Hepes (pH 7.6), 150 mM NaCl, 1 X Roche cOmplete, mini, EDTA-free protease inhibitor cocktail, and 0.1% Tween-20. This diluted lysate was subjected to either 50–100 µL bed volume of Anti-Flag M2 affinity agarose gel (A2220, Sigma) that had been pre-equilibrated with the binding buffer above. The sample was incubated with beads overnight at 4 °C by end-over-end rotation, after which time the supernatant was removed post-centrifugation at 500 x g. The beads were washed three times at 4 °C with 500 µL wash buffer containing in final concentrations, 50 mM Hepes (pH 7.6), 150 mM NaCl, 1 X Roche cOmplete, mini, EDTA-free protease inhibitor cocktail, and 0.1% Tween-20, using end-over-end rotation over 10 min during each wash. The bound protein was eluted by incubating with 0.15 mg/mL 3 X Flag peptide for 2 h at 4 °C. The sample was subjected to SDS-PAGE and transferred to a PVDF membrane for western blot analysis using antibodies indicated in corresponding figure legends.

## Materials availability statement

All plasmids generated by the authors are provided in **Appendix**. These plasmids are being deposited to Addgene upon manuscript publication. In the interim, they are available from the corresponding primary contact upon request, in the same way as for small-molecule reagents used in REX technologies. All other chemical and biological materials used are commercially available and their sources are indicated in Materials & Methods (**Key Resources Table**) and **Appendix**.

## Acknowledgements

Novartis FreeNovation Grant; European Research Council (ERC) grant (Project no. 101043303) funded by the State Secretairat for Education, Research and Innovation, Switzerland (SERI), and Swiss Federal Institute of Technology Lausanne (EPFL) (to YA). For zebrafish husbandry: Dr. Guillaume Valentin and Ms. Chloé C L Jollivet at EPFL; and Mr. Brian J Miller and Mrs. Nikki Gilbert, and Professor Joe Fetcho (NIH R01 NS026593, PI: JRF) at Cornell University. National BioResource Project Zebrafish (NBRP) grant funded by Japanese government for Tg(gstp1:GFP) fish line. NIH CBI training grant [NIH T32GM008500 (JRP as a trainee fellow)] and AHA predoctoral fellowship (17PRE33670395 to JRP); HHMI International Fellow (SP).

## Additional information

### Funding

| Funder | Grant reference number | Author |
|---|---|---|
| Novartis FreeNovation | | Yimon Aye |
| European Research Council | 101043303 | Yimon Aye |
| Swiss Federal Institute of Technology Lausanne | | Yimon Aye |
| National Institutes of Health | NIH T32GM008500 | Jesse R Poganik |
| AHA predoctoral Fellowship | 17PRE33670395 | Jesse R Poganik |
| HHMI International Fellow | | Saba Parvez |

The funders had no role in study design, data collection and interpretation, or the decision to submit the work for publication.

### Author contributions

Alexandra Van Hall-Beauvais, Jesse R Poganik, Kuan-Ting Huang, Saba Parvez, Data curation, Formal analysis, Investigation, Writing – review and editing; Yi Zhao, Data curation, Formal analysis, Investigation; Hong-Yu Lin, Xuyu Liu, Resources; Marcus John Curtis Long, Conceptualization, Resources, Data curation, Formal analysis, Investigation, Methodology, Writing – original draft, Writing – review

and editing; Yimon Aye, Conceptualization, Resources, Data curation, Formal analysis, Supervision, Funding acquisition, Methodology, Writing – original draft, Project administration, Writing – review and editing

**Author ORCIDs**
Alexandra Van Hall-Beauvais ![ORCID] http://orcid.org/0000-0003-2515-5191
Kuan-Ting Huang ![ORCID] http://orcid.org/0000-0001-7057-1448
Yi Zhao ![ORCID] http://orcid.org/0000-0002-6049-1943
Yimon Aye ![ORCID] http://orcid.org/0000-0002-1256-4159

**Ethics**
All procedures performed at Cornell (2017-2018) and EPFL (2018-present) conform to the animal care, maintenance, and experimentation procedures followed by Cornell University's and EPFL's Institutional Animal Care and Use Committee (IACUC) guidelines and approved by the respective institutional committees. All experiments with zebrafish performed at EPFL (2018-present) have been performed in accordance with the Swiss regulations on Animal Experimentation (Animal Welfare Act SR 455 and Animal Welfare Ordinance SR 455.1), in the EPFL zebrafish unit, cantonal veterinary authorization VD-H23.

**Decision letter and Author response**
Decision letter https://doi.org/10.7554/eLife.83373.sa1
Author response https://doi.org/10.7554/eLife.83373.sa2

## Additional files

**Supplementary files**
• MDAR checklist

**Data availability**
The data generated in this study using these materials are provided in main Figures 1–8, accompanied by 17 associated figure supplements, and the source data files associated with main Figures 1–8 and 17 associated figure supplements.

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

# Appendix 1

## Gene Sequences

(His$_6$)-Halo-TEV-hKeap1-(2×HA)

```
ATGGGCAGCAGCCATCATCATCATCATCATGGGTCAGGGATGGCAGAAATCGGTACTGGC
TTTCCATTCGACCCCCATTATGTGGAAGTCCTGGGCGAGCGCATGCACTACGTCGATGTT
GGTCCGCGCGATGGCACCCCTGTGCTGTTCCTGCACGGTAACCCGACCTCCTCCTACGTG
TGGCGCAACATCATCCCGCATGTTGCACCGACCCATCGCTGCATTGCTCCAGACCTGATC
GGTATGGGCAAATCCGACAAACCAGACCTGGGTTATTTCTTCGACGACCACGTCCGCTTC
ATGGATGCCTTCATCGAAGCCCTGGGTCTGGAAGAGGTCGTCCTGGTCATTCACGACTG
GGGCTCCGCTCTGGGTTTCCACTGGGCCAAGCGCAATCCAGAGCGCGTCAAAGGTATTGC
ATTTATGGAGTTCATCCGCCCTATCCCGACCTGGGACGAATGGCCAGAATTTGCCCGCGA
GACCTTCCAGGCCTTCCGCACCACCGACGTCGGCCGCAAGCTGATCATCGATCAGAACG
TTTTTATCGAGGGTACGCTGCCGATGGGTGTCGTCCGCCCGCTGACTGAAGTCGAGATGG
ACCATTACCGCGAGCCGTTCCTGAATCCTGTTGACCGCGAGCCACTGTGGCGCTTCCCAA
ACGAGCTGCCAATCGCCGGTGAGCCAGCGAACATCGTCGCGCTGGTCGAAGAATACATGG
ACTGGCTGCACCAGTCCCCTGTCCCGAAGCTGCTGTTCTGGGGCACCCCAGGCGTTCTGA
TCCCACCGGCCGAAGCCGCTCGCCTGGCCAAAAGCCTGCCTAACTGCAAGGCTGTGGACA
TCGGCCCGGGTCTGAATCTGCTGCAAGAAGACAACCCGGACCTGATCGGCAGCGAG
ATCGCGCGCTGGCTGTCGACGCTCGAGATTTCCGGCGAGCCAACCACTGAGGATCTGTAC
TTTCAGAGCGATAACGCGATCGCCATGCAGCCAGATCCCAGGCCTAGCGGGGCTGGGGCC
TGCTGCCGATTCCTGCCCCTGCAGTCACAGTGCCCTGAGGGGGCAGGGGACGCGGTGA
TGTACGCCTCCACTGAGTGCAAGGCGGAGGTGACGCCCTCCCAGCATGGCAACCGCACCT
TCAGCTACACCCTGGAGGATCATACCAAGCAGGCCTTTGGCATCATGAACGAGCTGCGGC
TCAGCCAGCAGCTGTGTGACGTCACACTGCAGGTCAAGTACCAGGATGCACCGGCCGCCC
AGTTCATGGCCCACAAGGTGGTGCTGGCCTCATCCAGCCCTGTCTTCAAGGCCATGTTCA
CCAACGGGCTGCGGGAGCAGGGCATGGAGGTGGTGTCCATTGAGGGTATCCACCCC
AAGGTCATGGAGCGCCTCATTGAATTCGCCTACACGGCCTCCATCTCCATGGGCGAGAAG
TGTGTCCTCCACGTCATGAACGGTGCTGTCATGTACCAGATCGACAGCGTTGTCCGTGCC
TGCAGTGACTTCCTGGTGCAGCAGCTGGACCCCAGCAATGCCATCGGCATCGCCAACTTC
GCTGAGCAGATTGGCTGTGTGGAGTTGCACCAGCGTGCCCGGGAGTACATCTACATGCAT
TTTGGGGAGGTGGCCAAGCAAGAGGAGTTCTTCAACCTGTCCCACTGCCAACTGGTGACC
CTCATCAGCCGGGACGACCTGAACGTGCGCTGCGAGTCCGAGGTCTTCCACGCCTG
CATCAACTGGGTCAAGTACGACTGCGAACAGCGACGGTTCTACGTCCAGGCGCTGCTGCG
GGCCGTGCGCTGCCACTCGTTGACGCCGAACTTCCTGCAGATGCAGCTGCAGAAGTGCGA
GATCCTGCAGTCCGACTCCCGCTGCAAGGACTACCTGGTCAAGATCTTCGAGGAGCTCAC
CCTGCACAAGCCCACGCAGGTGATGCCCTGCCGGGCGCCCAAGGTGGGCCGCCTGA
TCTACACCGCGGGCGGCTACTTCCGACAGTCGCTCAGCTACCTGGAGGCTTACAACCCCA
GTGACGGCACCTGGCTCCGGTTGGCGGACCTGCAGGTGCCGCGGAGCGGCCTGGCC
GGCTGCGTGGTGGGCGGGCTGTTGTACGCCGTGGGCGGCAGGAACAACTCGCCCGA
CGGCAACACCGACTCCAGCGCCCTGGACTGTTACAACCCCATGACCAATCAGTGGTCGCC
CTGCGCCCCCATGAGCGTGCCCCGTAACCGCATCGGGGTGGGGGTCATCGATGGCCACAT
CTATGCCGTCGGCGGCTCCCACGGCTGCATCCACCACAACAGTGTGGAGAGGTATGAGCC
AGAGCGGGATGAGTGGCACTTGGTGGCCCCAATGCTGACACGAAGGATCGGGGTGG
GCGTGGCTGTCCTCAATCGTCTCCTTTATGCCGTGGGGGGCTTTGACGGGACAAACCGCC
TTAATTCAGCTGAGTGTTACTACCCAGAGAGGAACGAGTGGCGAATGATCACAGCAATGA
ACACCATCCGAAGCGGGGCAGGCGTCTGCGTCCTGCACAACTGTATCTATGCTGCTGGGG
GCTATGATGGTCAGGACCAGCTGAACAGCGTGGAGCGCTACGATGTGGAAACAGAGACGT
GGACTTTCGTAGCCCCCATGAAGCACCGGCGAAGTGCCCTGGGGATCACTGTCCACCAGG
GGAGAATCTACGTCCTTGGAGGCTATGATGGTCACACGTTCCTGGACAGTGTGGAGTGTT
ACGACCCAGATACAGACACCTGGAGCGAGGTGACCCGAATGACATCGGGCCGGAGT
GGGGTGGGCGTGGCTGTCACCATGGAGCCCTGCCGGAAGCAGATTGACCAGCAGAA
CTGTACCTGTGGCAGCTACCCATACGATGTTCCAGATTACGCTGGCAGCTACCCA     TACG
ATGTTCCAGATTACGCTTAA
```

## (His₆)-Halo-(2×HA)-P2A-TEV-hKeap1-(2×HA)

ATGGGCAGCAGCCATCATCATCATCATCATGGGTCAGGGATGGCAGAAATCGGTACTGGC
TTTCCATTCGACCCCCATTATGTGGAAGTCCTGGGCGAGCGCATGCACTACGTCGATGTT
GGTCCGCGCGATGGCACCCCTGTGCTGTTCCTGCACGGTAACCCGACCTCCTCCTACGTG
TGGCGCAACATCATCCCGCATGTTGCACCGACCCATCGCTGCATTGCTCCAGACCTGATC
GGTATGGGCAAATCCGACAAACCAGACCTGGGTTATTTCTTCGACGACCACGTCCGCTTC
ATGGATGCCTTCATCGAAGCCCTGGGTCTGGAAGAGGTCGTCCTGGTCATTCACGACTGG
GGCTCCGCTCTGGGTTTCCACTGGGCCAAGCGCAATCCAGAGCGCGTCAAAGGTATTGCA
TTTATGGAGTTCATCCGCCCTATCCCGACCTGGGACGAATGGCCAGAATTTGCCCGCGAG
ACCTTCCAGGCCTTCCGCACCACCGACGTCGGCCGCAAGCTGATCATCGATCAGAACGTT
TTTATCGAGGGTACGCTGCCGATGGGTGTCGTCCGCCCGCTGACTGAAGTCGAGATGGAC
CATTACCGCGAGCCGTTCCTGAATCCTGTTGACCGCGAGCCACTGTGGCGCTTCCCAAAC
GAGCTGCCAATCGCCGGTGAGCCAGCGAACATCGTCGCGCTGGTCGAAGAATACATGGAC
TGGCTGCACCAGTCCCCTGTCCCGAAGCTGCTGTTCTGGGGCACCCCAGGCGTTCTGATC
CCACCGGCCGAAGCCGCTCGCCTGGCCAAAAGCCTGCCTAACTGCAAGGCTGTGGA
CATCGGCCCGGGTCTGAATCTGCTGCAAGAAGACAACCCGGACCTGATCGGCAGCGAGAT
CGCGCGCTGGCTGTCGACGCTCGAGATTTCCGGCTATCCTTACGACGTCCCAGACTACGC
CGGCAGCTACCCATACGATGTTCCAGATTACGCCGGAAGCGGAGCTACTAACTTCAGCCT
GCTGAAGCAGGCTGGAGACGTGGAGGAGAACCCTGGACCTGGCAGCGAGCCAACCA
CTGAGGATCTGTACTTTCAGAGCGATAACGCGATCGCCATGCAGCCAGATCCCAGGCCTA
GCGGGGCTGGGGCCTGCTGCCGATTCCTGCCCCTGCAGTCACAGTGCCCTGAGGGG
GCAGGGGACGCGGTGATGTACGCCTCCACTGAGTGCAAGGCGGAGGTGACGCCCTC
CCAGCATGGCAACCGCACCTTCAGCTACACCCTGGAGGATCATACCAAGCAGGCCTTTGG
CATCATGAACGAGCTGCGGCTCAGCCAGCAGCTGTGTGACGTCACACTGCAGGTCAAGTA
CCAGGATGCACCGGCCGCCCAGTTCATGGCCCACAAGGTGGTGCTGGCCTCATCCAGCCC
TGTCTTCAAGGCCATGTTCACCAACGGGCTGCGGGAGCAGGGCATGGAGGTGGTGTCCAT
TGAGGGTATCCACCCCAAGGTCATGGAGCGCCTCATTGAATTCGCCTACACGGCCTCCAT
CTCCATGGGCGAGAAGTGTGTCCTCCACGTCATGAACGGTGCTGTCATGTACCAGATCGA
CAGCGTTGTCCGTGCCTGCAGTGACTTCCTGGTGCAGCAGCTGGACCCCAGCAATGCCAT
CGGCATCGCCAACTTCGCTGAGCAGATTGGCTGTGTGGAGTTGCACCAGCGTGCCCGGGA
GTACATCTACATGCATTTTGGGGAGGTGGCCAAGCAAGAGGAGTTCTTCAACCTGTCCCA
CTGCCAACTGGTGACCCTCATCAGCCGGGACGACCTGAACGTGCGCTGCGAGTCCGAGGT
CTTCCACGCCTGCATCAACTGGGTCAAGTACGACTGCGAACAGCGACGGTTCTACGTCCA
GGCGCTGCTGCGGGCCGTGCGCTGCCACTCGTTGACGCCGAACTTCCTGCAGATGCAGCT
GCAGAAGTGCGAGATCCTGCAGTCCGACTCCCGCTGCAAGGACTACCTGGTCAAGATCTT
CGAGGAGCTCACCCTGCACAAGCCCACGCAGGTGATGCCCTGCCGGGCGCCCAAGG
TGGGCCGCCTGATCTACACCGCGGGCGGCTACTTCCGACAGTCGCTCAGCTACCTGGAGG
CTTACAACCCCAGTGACGGCACCTGGCTCCGGTTGGCGGACCTGCAGGTGCCGCGG
AGCGGCCTGGCCGGCTGCGTGGTGGGCGGGCTGTTGTACGCCGTGGGCGGCAGGAA
CAACTCGCCCGACGGCAACACCGACTCCAGCGCCCTGGACTGTTACAACCCCATGACCAA
TCAGTGGTCGCCCTGCGCCCCCATGAGCGTGCCCCGTAACCGCATCGGGGTGGGGGTCAT
CGATGGCCACATCTATGCCGTCGGCGGCTCCCACGGCTGCATCCACCACAACAGTGTGGA
GAGGTATGAGCCAGAGCGGGATGAGTGGCACTTGGTGGCCCCAATGCTGACACGAAGGAT
CGGGGTGGGCGTGGCTGTCCTCAATCGTCTCCTTTATGCCGTGGGGGGCTTTGACGGGAC
AAACCGCCTTAATTCAGCTGAGTGTTACTACCCAGAGAGGAACGAGTGGCGAATGATCAC
AGCAATGAACACCATCCGAAGCGGGGCAGGCGTCTGCGTCCTGCACAACTGTATCTATGC
TGCTGGGGGCTATGATGGTCAGGACCAGCTGAACAGCGTGGAGCGCTACGATGTGGAAAC
AGAGACGTGGACTTTCGTAGCCCCCATGAAGCACCGGCGAAGTGCCCTGGGGATCACTGT
CCACCAGGGGAGAATCTACGTCCTTGGAGGCTATGATGGTCACACGTTCCTGGACAGTGT
GGAGTGTTACGACCCAGATACAGACACCTGGAGCGAGGTGACCCGAATGACATCGGGCCG
GAGTGGGGTGGGCGTGGCTGTCACCATGGAGCCCTGCCGGAAGCAGATTGACCAGC
AGAACTGTACCTGTGGCAGCTACCCATACGATGTTCCAGATTACGCTGGCAGCTACCCAT
ACGATGTTCCAGATTACGCTTAA

## Myc-Halo-TEV-3×Flag-zKeap1a

```
ATGGAACAAAAACTCATCTCAGAAGAGGATCTGATGGCAGAAATCGGTACTGGCTTTCCA
TTCGACCCCATTATGTGGAAGTCCTGGGCGAGCGCATGCACTACGTCGATGTTGGTCCG
CGCGATGGCACCCCTGTGCTGTTCCTGCACGGTAACCCGACCTCCTCCTACGTGTGGCGC
AACATCATCCCGCATGTTGCACCGACCCATCGCTGCATTGCTCCAGACCTGATCGGTATG
GGCAAATCCGACAAACCAGACCTGGGTTATTTCTTCGACGACCACGTCCGCTTCATGGAT
GCCTTCATCGAAGCCCTGGGTCTGGAAGAGGTCGTCCTGGTCATTCACGACTGGGGCTCC
GCTCTGGGTTTCCACTGGGCCAAGCGCAATCCAGAGCGCGTCAAAGGTATTGCATTTATG
GAGTTCATCCGCCCTATCCCGACCTGGGACGAATGGCCAGAATTTGCCCGCGAGACCTTC
CAGGCCTTCCGCACCACCGACGTCGGCCGCAAGCTGATCATCGATCAGAACGTTTTTATC
GAGGGTACGCTGCCGATGGGTGTCGTCCGCCCGCTGACTGAAGTCGAGATGGACCATTAC
CGCGAGCCGTTCCTGAATCCTGTTGACCGCGAGCCACTGTGGCGCTTCCCAAACGAGCTG
CCAATCGCCGGTGAGCCAGCGAACATCGTCGCGCTGGTCGAAGAATACATGGACTGGCTG
CACCAGTCCCCTGTCCCGAAGCTGCTGTTCTGGGGCACCCCAGGCGTTCTGATCCCACCG
GCCGAAGCCGCTCGCCTGGCCAAAAGCCTGCCTAACTGCAAGGCTGTGGACATCGG
CCCGGGTCTGAATCTGCTGCAAGAAGACAACCCGGACCTGATCGGCAGCGAGATCG
CGCGCTGGCTGTCGACGCTCGAGATTTCCGGCTCAGGGGAAAACTTGTATTTCCAGGGCT
CAGGGATGGATTATAAAGATCATGATGGCGATTATAAAGATCATGATATTGATTATAAAGATGA
TGATGATAAAATGATATGTCCAAGAAAGAAGAGGCCCATCAAAGATGAGGATTTCTCCGC
CATCGTGGTCCCCTCCATGAGGGGTCACGGTTACTTGGATTACACGGTTGAAAGTCATCC
GTCTAAAGCTCTGCAGAACATGGACGAGCTGCGTCATCATGAAATGCTGTGTGATCTGGT
TCTGCATGTCACATACAAGGACAAGATAGTGGATTTTAAGGTGCATAAGCTGGTTCTGGC
CGCCTCCAGTCCTTACTTCAAAGCCATGTTCACCAGCAACTTCAAGGAGTGCCACGCGTC
GGAAGTCACCCTTCGAGACGTTTGTCCTCAAGTCATCAGCCGTCTCATTGACTTTGCCTA
CACCTCGCGCATCACAGTTGGCGAGACCTGCGTTCTTCACGTCCTCTTGACCGCCATGCG
CTACCAAATGGAAGAAGTGGCCAAAGCCTGCTGCGATTTCCTCATGAAGAACCTGGAGCC
ATCCAATGTCATCGGCATCTCGAGATTCGCTGAGGAGATCGGCTGCACTGACCTACACCT
TCGCACCAGAGAGTATATCAACACTCACTTCAATGAGGTAACCAAAGAAGAAGAGTTCTT
CAGCTTGTCCCATTGCCAGCTGCTTGAGCTGATCAGTCAGGACAGTCTGAAGGTGCTCTG
CGAGAGCGAGGTCTACAAGGCCTGCATAGACTGGGTACGCTGGGACGCAGAGAGCC
GTGCGCAGTACTTCCATGCCCTCCTCAATGCCGTCCACATCTACGCCCTTCCACCCACTT
TCCTCAAAAGACAACTGCAGTCCTGCCCCATCCTCAGCAAGGCCAACTCCTGCAAAGACT
TCCTATCAAAGATCTTCCATGAAATGGCTCTCCGAAAACCCCTGCCGCCAACACCTCATC
GTGGGACGCAGCTCATTTACATAGCGGGAGGTTACAAGCAACACTCTCTGGACACCTTGG
AGGCCTTCGACCCGCACAAGAACGTCTGGCTCAAACTAGGTAGCATGATGTCTCCTTGTA
GCGGGCTTGGGGCGTGTGTTTTGTTCGGGCTTCTTTATACAGTCGGCGGACGCAATCTCT
CCCTGCAGAACAACACAGAATCTGGATCTTTGTCCTGCTACAACCCCATGACTAACCAGT
GGACCCAGCTGGCTCCGCTCAACACACCCAGAAACCGAGTGGGCGTCGGGGTCATTGATG
GGAGCATTTATGCTGTTGGGGGGTTCACATGCCTCTACGCATCACAACAGCGTCGAGAGGT
ATGACCCAGAAACAAACCGCTGGACGTTTGTGGCCCCTATGTCAGTGGCGCGACTAGGGG
CCGGTGTGGCGGCATGTGGAGGTTGCCTGTATGTGGTAGGAGGGTTTGACGGGGACAACC
GGTGGAACACAGTGGAGCGATACCAACCAGACACCAACACCTGGCAGCATGTGGCACCTA
TGAACACAGTGCGCAGCGGGCTGGGGGTGGTGTGTATGGATAACTACCTCTATGCAGTTG
GAGGCTATGATGGACAAACCCAACTCAAAACCATGGAGAGATATAACATCACTCGAGATG
TGTGGGAACCCATGGCTTCGATGAACCACTGCCGCAGTGCACATGGAGTCTCAGTCTACC
AGTGCAAGATTTTTGTGTTAGGTGGATTTAACCAAGGTGGTTTCCTGTCCAGTGTGGAGT
GCTACTGTCCCGCCAGTAATGTATGGACGCTTGTAACAGATATGCCCGTGGGACGCAGTG
GAATGGGTGTAGCTGTGACCATGGAACCGTGTCCTGGTATCCTGCCAGAGGAGGAGGAAG
AAGTGGACGAGGAGATGTGA
```

## Myc-Halo-TEV-3×Flag-zKeap1a (synonymous mutations preventing zKeap1a-ATG-MO binding)

```
ATGGAACAAAAACTCATCTCAGAAGAGGATCTGATGGCAGAAATCGGTACTGGCTTTCCA
TTCGACCCCCATTATGTGGAAGTCCTGGGCGAGCGCATGCACTACGTCGATGTTGGTCCG
CGCGATGGCACCCCTGTGCTGTTCCTGCACGGTAACCCGACCTCCTCCTACGTGTGGCGC
AACATCATCCCGCATGTTGCACCGACCCATCGCTGCATTGCTCCAGACCTGATCGGTATG
GGCAAATCCGACAAACCAGACCTGGGTTATTTCTTCGACGACCACGTCCGCTTCATGGAT
GCCTTCATCGAAGCCCTGGGTCTGGAAGAGGTCGTCCTGGTCATTCACGACTGGGGCTCC
GCTCTGGGTTTCCACTGGGCCAAGCGCAATCCAGAGCGCGTCAAAGGTATTGCATTTATG
GAGTTCATCCGCCCTATCCCGACCTGGGACGAATGGCCAGAATTTGCCCGCGAGACCTTC
CAGGCCTTCCGCACCACCGACGTCGGCCGCAAGCTGATCATCGATCAGAACGTTTTTATC
GAGGGTACGCTGCCGATGGGTGTCGTCCGCCCGCTGACTGAAGTCGAGATGGACCATTAC
CGCGAGCCGTTCCTGAATCCTGTTGACCGCGAGCCACTGTGGCGCTTCCCAAACGAGCTG
CCAATCGCCGGTGAGCCAGCGAACATCGTCGCGCTGGTCGAAGAATACATGGACTGGCTG
CACCAGTCCCCTGTCCCGAAGCTGCTGTTCTGGGGCACCCCAGGCGTTCTGATCCCACCG
GCCGAAGCCGCTCGCCTGGCCAAAAGCCTGCCTAACTGCAAGGCTGTGGACATCGG
CCCGGGTCTGAATCTGCTGCAAGAAGACAACCCGGACCTGATCGGCAGCGAGATCG
CGCGCTGGCTGTCGACGCTCGAGATTTCCGGCTCAGGGGAAAACTTGTATTTCCAGGGCT
CAGGGATGGATTATAAAGATCATGATGGCGATTATAAAGATCATGATATTGATTATAAAGATGA
TGATGATAAAATGATATGTCCAAGAAAGAAGAGGCCCATCAAAGATGAGGATTTCTCCGC
CATCGTGGTCCCCTCCATGAGGGGTCACGGTTACTTGGATTACACGGTTGAAAGTCATCC
GTCTAAAGCTCTGCAGAACATGGACGAGCTGCGTCATCATGAAATGCTGTGTGATCTGGT
TCTGCATGTCACATACAAGGACAAGATAGTGGATTTTAAGGTGCATAAGCTGGTTCTGGC
CGCCTCCAGTCCTTACTTCAAAGCCATGTTCACCAGCAACTTCAAGGAGTGCCACGCGTC
GGAAGTCACCCTTCGAGACGTTTGTCCTCAAGTCATCAGCCGTCTCATTGACTTTGCCTA
CACCTCGCGCATCACAGTTGGCGAGACCTGCGTTCTTCACGTCCTCTTGACCGCCATGCG
CTACCAAATGGAAGAAGTGGCCAAAGCCTGCTGCGATTTCCTCATGAAGAACCTGGAGCC
ATCCAATGTCATCGGCATCTCGAGATTCGCTGAGGAGATCGGCTGCACTGACCTACACCT
TCGCACCAGAGAGTATATCAACACTCACTTCAATGAGGTAACCAAAGAAGAAGAGTTCTT
CAGCTTGTCCCATTGCCAGCTGCTTGAGCTGATCAGTCAGGACAGTCTGAAGGTGCTCTG
CGAGAGCGAGGTCTACAAGGCCTGCATAGACTGGGTACGCTGGGACGCAGAGAGCC
GTGCGCAGTACTTCCATGCCCTCCTCAATGCCGTCCACATCTACGCCCTTCCACCCACTT
TCCTCAAAAGACAACTGCAGTCCTGCCCCATCCTCAGCAAGGCCAACTCCTGCAAAGACT
TCCTATCAAAGATCTTCCATGAAATGGCTCTCCGAAAACCCCTGCCGCCAACACCTCATC
GTGGGACGCAGCTCATTTACATAGCGGGAGGTTACAAGCAACACTCTCTGGACACCTTGG
AGGCCTTCGACCCGCACAAGAACGTCTGGCTCAAACTAGGTAGCATGATGTCTCCTTGTA
GCGGGCTTGGGGCGTGTGTTTTGTTCGGGCTTCTTTATACAGTCGGCGGACGCAATCTCT
CCCTGCAGAACAACACAGAATCTGGATCTTTGTCCTGCTACAACCCCATGACTAACCAGT
GGACCCAGCTGGCTCCGCTCAACACACCCAGAAACCGAGTGGGCGTCGGGGTCATTGATG
GGAGCATTTATGCTGTTGGGGGTTCACATGCCTCTACGCATCACAACAGCGTCGAGAGGT
ATGACCCAGAAACAAACCGCTGGACGTTTGTGGCCCCTATGTCAGTGGCGCGACTAGGGG
CCGGTGTGGCGGCATGTGGAGGTTGCCTGTATGTGGTAGGAGGGTTTGACGGGGACAACC
GGTGGAACACAGTGGAGCGATACCAACCAGACACCAACACCTGGCAGCATGTGGCACCTA
TGAACACAGTGCGCAGCGGGCTGGGGGTGGTGTGTATGGATAACTACCTCTATGCAGTTG
GAGGCTATGATGGACAAACCCAACTCAAAACCATGGAGAGATATAACATCACTCGAGATG
TGTGGGAACCCATGGCTTCGATGAACCACTGCCGCAGTGCACATGGAGTCTCAGTCTACC
AGTGCAAGATTTTTGTGTTAGGTGGATTTAACCAAGGTGGTTTCCTGTCCAGTGTGGAGT
GCTACTGTCCCGCCAGTAATGTATGGACGCTTGTAACAGATATGCCCGTGGGACGCAGTG
GAATGGGTGTAGCTGTGACCATGGAACCGTGTCCTGGTATCCTGCCAGAGGAGGAGGAAG
AAGTGGACGAGGAGATGTGA
```

## Myc-Halo-TEV-3×Flag-zKeap1b

```
ATGGAACAAAAACTCATCTCAGAAGAGGATCTGATGGCAGAAATCGGTACTGGCTTTCCA
TTCGACCCCATTATGTGGAAGTCCTGGGCGAGCGCATGCACTACGTCGATGTTGGTCCG
CGCGATGGCACCCCTGTGCTGTTCCTGCACGGTAACCCGACCTCCTCCTACGTGTGGCGC
AACATCATCCCGCATGTTGCACCGACCCATCGCTGCATTGCTCCAGACCTGATCGGTATG
GGCAAATCCGACAAACCAGACCTGGGTTATTTCTTCGACGACCACGTCCGCTTCATGGAT
GCCTTCATCGAAGCCCTGGGTCTGGAAGAGGTCGTCCTGGTCATTCACGACTGGGGCTCC
GCTCTGGGTTTCCACTGGGCCAAGCGCAATCCAGAGCGCGTCAAAGGTATTGCATTTATG
GAGTTCATCCGCCCTATCCCGACCTGGGACGAATGGCCAGAATTTGCCCGCGAGACCTTC
CAGGCCTTCCGCACCACCGACGTCGGCCGCAAGCTGATCATCGATCAGAACGTTTTTATC
GAGGGTACGCTGCCGATGGGTGTCGTCCGCCCGCTGACTGAAGTCGAGATGGACCATTAC
CGCGAGCCGTTCCTGAATCCTGTTGACCGCGAGCCACTGTGGCGCTTCCCAAACGAGCTG
CCAATCGCCGGTGAGCCAGCGAACATCGTCGCGCTGGTCGAAGAATACATGGACTGGCTG
CACCAGTCCCCTGTCCCGAAGCTGCTGTTCTGGGGCACCCCAGGCGTTCTGATCCCACCG
GCCGAAGCCGCTCGCCTGGCCAAAAGCCTGCCTAACTGCAAGGCTGTGGACATCGG
CCCGGGTCTGAATCTGCTGCAAGAAGACAACCCGGACCTGATCGGCAGCGAGATCGCGCG
CTGGCTGTCGACGCTCGAGATTTCCGGCTCAGGGGAAAACTTGTATTTCCAGGGCTCAGG
GATGGATTATAAAGATCATGATGGCGATTATAAAGATCATGATATTGATTATAAAGATGATGAT
GATAAAATGTTGGCGGCGGCGGGCATGACGGAGTGTAAGGCGGAGGTGACGCCGTC
GGCCAGCAATGGGCACCGCGTGTTCAGCTACACGTTGGAGAGCCACACGGCCGCCG
CCTTCGCCATCATGAACGAGCTGCGGCGCGAGAGACAGCTGTGTGACGTCACACTCCGCG
TGCGCTACTGCCCGCTCGACACACACGTCGACTTCGTGGCGCATAAGGTGGTGCTGGCCT
CGTCCTCGCCTGTGTTCCGCGCCATGTTCACCAACGGCCTGAAGGAGTGCGGCATGGAGG
TGGTGCCCATCGAGGGGATACACCCCAAGGTCATGGGCCGGCTCATTGAGTTTGCGTACA
CGGCGAGCATCTCAGTGGGTGAGAAGTGTGTGATCCACGTGATGAACGGCGCCGTGATGT
ACCAGATCGACAGCGTGGTTCAGGCCTGCTGTGATTTCCTGGTGGAGCAGCTGGACCCCA
GTAACGCCATCGGCATCGCCAGCTTCGCCGAGCAGATCGGCTGCACGGAGCTCCACCAGA
AGGCCAGAGAGTACATCTACATGAACTTCAGCCAGGTGGCGACGCAGGAGGAGTTCTTCA
CCCTGTCTCACTGTCAGCTGGTGACCCTGATCAGCCGGGACGAGCTGAACGTGCGCTGCG
AGTCGGAGGTGTTCCACGCGTGTGTGGCGTGGGTTCAGTACGACCGTGAGGAGCGG
CGTCCGTATGTGCAGGCGCTGCTGCAGGCCGTCCGCTGCCACTCGCTCACGCCGCACTTC
CTGCAGCGGCAGCTGGAGCACTTCGAGTGGGACGCGCAGAGCAAAGACTACCTGTC
GCAGATCTTCCGGGACCTGACGCTGCACAAGCCCACCAAGGTCATCCCCCTGCGCACGCC
CAAGGTGCCGCAGCTGATCTACACGGTGGGCGGATACTTCCGGCAGTCGCTCAGCTTCCT
GGAGGCCTTCAACCCCTGCAGCGGCGCGTGGCTGCGGCTGGCGGACCTGCAGGTGC
CCCGCAGCGGGCTGGCGGCCTGCGTCATCAGCGGCCTGCTGTACGCCGTGGGCGGA
CGCAACAACGGGCCCGACGGGAACATGGACTCACACACACTCGACTGCTACAACCCCATG
AACAACTGCTGGCGGCCCTGCGCACACATGAGCGTCCCGCGCAACCGCATCGGCGT
CGGCGTCATCGACGGCATGATCTACGCCGTGGGCGGATCACACGGCTGCACACACCACAA
CAGCGTGGAGAGGTATGACCCGGAGCGGGACAGCTGGCAGCTGGTGTCGCCAATGC
TGACGCGGCGGATCGGAGTGGGCGTGGCCGTGATCAACCGGCTGCTGTATGCGGTG
GGCGGCTTCGATGGGACGCACCGGCTGAGCTCCGCGGAATGCTACAACCCCGAGCG
GGACGAGTGGAGGAGCATAGCGGCCATGAACACAGTCCGCAGCGGCGCAGGTGTGT
GTGCGCTGGGGAACTACATCTATGTGATGGGTGGATATGACGGCACCAACCAGCTGAACA
CGGTGGAGCGCTACGATGTGGAGAAGGACAGCTGGAGCTTCAGCGCATCCATGCGGCACC
GGCGCAGCGCTCTGGGGGTCACCACACACCACGGACGCATCTATGTGCTGGGTGGCTATG
ATGGAACACGTTCCTGGACAGTGTGGAGTGTTTTGACCCAGAGACGGACTCATGGACAG
AGGTCACACACATGAAGTCGGGCCGCAGCGGAGTCGGAGTCGCCGTCACCATGGAG
CCCTGTCACAAAGAGCTGATCCCCTGTCAGTGCTAA
```

## Self-cloned plasmids*:

| |
|---|
| **pCS2 +8 HA-Nrf2** |
| pCS2 +8 His6-Halo-TEV-hKeap1(C273I) |
| pCS2 +8 His6-Halo-TEV-hKeap1 |
| pCS2 +8 His6-Halo-TEV-hKeap1-(2xHA) |
| pCS2 +8 His6-Halo-(2xHA)-P2A-TEV-hKeap1-(2xHA) |
| pCS2 +8 zKeap1a |
| pCS2 +8 myc-Halo-TEV-3xFlag-zKeap1a |
| pCS2 +8 myc-Halo-TEV-3xFlag-zKeap1a, synonymous mutant preventing zKeap1a-ATG-MO binding |
| pCS2 +8 zKeap1b |
| pCS2 +8 myc-Halo-•–3xFlag-zKeap1b |

## Primers for cloning plasmids

| | | |
|---|---|---|
| Primers for cloning pCS2 +8 His₆-Halo-•-hKeap1-(2×HA) | Primers for gene amplification (His₆-Halo-•-hKeap1) *template: pFN21a Halo-TEV-Keap1 | Fwd 1:<br>CATGGGCAGCAGCCATCATCATCATCATCAT GGGTCAGGGATGGCAGAAATCGGTACTGG<br>Rev 1:<br>CCAGCGTAATCTGGAACATCGTATGGGTAGC TGCCACAGGTACAGTTCTGCTGGTCAATC |
| | Extension primers 1 *template: PCR product from the above amplification step | Fwd ext 1:<br>AGGTGACACTATAGAATACAAGCTACTTGTT CTTTTCCACCATGGGCAGCAGCCATCATC<br>Rev ext 1:<br>AGCGTAATCTGGAACATCGTATGGGTAGCTG CCAGCGTAATCTGGAACATCGTATG |
| | Extension primers 2 *template: PCR product from the above extension step *PCR product was inserted into pCS2 +8 empty vector | Fwd ext 2:<br>GTCGGAGCAAGCTTGATTTAGGTGACACTATA GAATACAAGCTACTTGTTCTTTTCCACC<br>Rev ext 2:<br>CGGCCTTTAATTAATGGCGCGCCACTAGTTTA AGCGTAATCTGGAACATCG |
| Primers for cloning pCS2 +8 His₆-Halo-(2×HA)-P2A-•-hKeap1-(2×HA) | Primers for gene amplification (Halo) *template: pFN21a Halo-TEV-Keap1 | Fwd 1:<br>CATGGGCAGCAGCCATCATCATCATCATCAT GGGTCAGGGATGGCAGAAATCGGTACTGG<br>Rev 1:<br>ATGGGTAGCTGCCGGCGTAGTCTGGGACGT CGTAAGGATAGCCGGAAATCTCGAGCGTCG |
| | Primers for gene amplification (hKeap1) *template: pFN21a Halo-TEV-Keap1 | Fwd 1':<br>GCTGGAGACGTGGAGGAGAACCCTGGACCT GGCAGCGAGCCAACCACTGAGGATCTGTAC<br>Rev 1':<br>CCAGCGTAATCTGGAACATCGTATGGGTAGC TGCCACAGGTACAGTTCTGCTGGTCAATC |
| | Extension primers 1 (Halo) *template: PCR product from the above amplification (Halo) step | Fwd ext 1:<br>AGGTGACACTATAGAATACAAGCTACTTGTTC TTTTCCACCATGGGCAGCAGCCATCATC<br>Rev ext 1:<br>AAGTTAGTAGCTCCGCTTCCGGCGTAATCTGG AACATCGTATGGGTAGCTGCCGGCGTAG |
| | Extension primers 1' (hKeap1) *template: PCR product from the above amplification (hKeap1) step | Fwd ext 1':<br>CGCCGGAAGCGGAGCTACTAACTTCAGCCTG CTGAAGCAGGCTGGAGACGTGGAGGAGAA<br>Rev ext 1':<br>AGCGTAATCTGGAACATCGTATGGGTAGCTG CCAGCGTAATCTGGAACATCGTATG |
| | Extension primers 2 *template: PCR product from the above extension (hKeap1) step *Additional PCR with the two extended product (Halo and hKeap1) was done to yield a megaprimer to be inserted into pCS2 +8 empty vector | Fwd ext 2:<br>GTCGGAGCAAGCTTGATTTAGGTGACA CTATAGAATACAAGCTACTTGTTCTTTTCCACC<br>Rev ext 2:<br>CGGCCTTTAATTAATGGCGCGCCACTAG TTTAAGCGTAATCTGGAACATCG |

*Continued on next page*

*Continued*

| | | |
|---|---|---|
| Primers for cloning pCS2 +8 His$_6$-Halo-•-hKeap1-(2×HA) | Primers for gene amplification (His$_6$-Halo-•-hKeap1) *template: pFN21a Halo-TEV-Keap1 | **Fwd 1:** CATGGGCAGCAGCCATCATCATCATCATCAT GGGTCAGGGATGGCAGAAATCGGTACTGG **Rev 1:** CCAGCGTAATCTGGAACATCGTATGGGTAGC TGCCACAGGTACAGTTCTGCTGGTCAATC |
| | Extension primers 1 *template: PCR product from the above amplification step | **Fwd ext 1:** AGGTGACACTATAGAATACAAGCTACTTGTT CTTTTCCACCATGGGCAGCAGCCATCATC **Rev ext 1:** AGCGTAATCTGGAACATCGTATGGGTAGCTG CCAGCGTAATCTGGAACATCGTATG |
| | Extension primers 2 *template: PCR product from the above extension step *PCR product was inserted into pCS2 +8 empty vector | **Fwd ext 2:** GTCGGAGCAAGCTTGATTTAGGTGACACTATA GAATACAAGCTACTTGTTCTTTTCCACC **Rev ext 2:** CGGCCTTTAATTAATGGCGCGCCACTAGTTTA AGCGTAATCTGGAACATCG |
| Primers for cloning pCS2 +8 zKeap1a | Primers for gene amplification (zKeap1a) *template: zKeap1a cDNA | Fwd: ATTAAAGGCCGGCCAGCGATCGCCGGA CCCACC ATGATATGTCCAAGAAAGAAGAGGC Rev: TCTAGAGGCTCGAGAGGCCTTGAATTC GATCACATCTCCTCGTCCACTTC |
| | Extension primers *template: PCR product from the above amplification step *PCR product was inserted into pCS2 +8 empty vector | Fwd: GCTACTTGTTCTTTTGCAGGATCCACT AGTGGCGCGCCATTAATTAAAGGCCGGCCAGC Rev: CTTATCATGTCTGGATCTACGTAATACG ACTCACTATAGTTCTAGAGGCTCGAGAGGCCT |
| Primers for cloning pCS2 +8 myc-Halo-•–3×Flag-zKeap1a | Primers for gene amplification (myc-Halo-•–3×Flag) *template (first PCR with Fwd and Rev): pCS2 +8 myc-Halo-TEV-3x Flag *template (second PCR with Fwd and Rev extender): PCR product from first reaction *PCR product was inserted into pCS2 +8 zKeap1a | Fwd: GCTACTTGTTCTTTTGCAGGATCCACTA GTGGCGCGCCATTAATTAAAGGCCGGCCAGC Rev: GCCTCTTCTTTCTTGGACATATCATTTTAT CATCATCATCTTTATAATCAATATCATGAT Rev extender: ACGATGGCGGAGAAATCCTCATCTTTGAT GGGCCTCTTCTTTCTTGGACATATCAT |
| Primers for introducing synonymous mutations in pCS2 +8 myc-Halo-•–3×Flag-zKeap1a, preventing zKeap1a-ATG-MO binding | Primers for gene amplification *template (first PCR with Fwd and Rev): pCS2 +8 myc-Halo-•–3×Flag-zKeap1a *template (second PCR with Fwd and Rev extender): PCR product from first reaction *PCR product was inserted into pCS2 +8 myc-Halo-•–3×Flag-zKeap1a | Fwd: CAGAAATCGGTACTGGCTTTCCA Rev: GCGTTTCTTGCGCGGGCAGATCATTTT ATCATCATCATCTTTATAATCAATATCATGATC Rev ext: GTCGGAGCAAGCTTGATTTAGGTGACA CTATAGAATACAAGCTACTTGTTCTTTTCCACC |
| Primers for cloning pCS2 +8 zKeap1b | Primers for gene amplification (zKeap1b) *template: zKeap1b cDNA | Fwd: ATTAAAGGCCGGCCAGCGATCGCCGG ACCCACC ATGTTGGCGGCGGC Rev: TCTAGAGGCTCGAGAGGCCTTGAATT CGAGCACTGACAGGGGATCAGC |
| | Extension primers *template: PCR product from the above amplification step *PCR product was inserted into pCS2 +8 empty vector | Fwd: GCTACTTGTTCTTTTGCAGGATCCACTA GTGGCGCGCCATTAATTAAAGGCCGGCCAGC Rev: CTTATCATGTCTGGATCTACGTAATACGA CTCACTATAGTTCTAGAGGCTCGAGAGGCCT |
| | Stop codon mutation primers (site-directed mutagenesis) *template: plasmid obtained from the above gene insertion step | Fwd: CACAAAGAGCTGATCCCCTGTCAGTGC**T AA**TCGAATTCAAGGCCTCTCGAGCCTCTAGA Rev: TCTAGAGGCTCGAGAGGCCTTGAATTCG **ATT**AGCACTGACAGGGGATCAGCTCTTTGTG |
| Primers for cloning pCS2 +8 myc-Halo-•–3×Flag-zKeap1b | Primers for gene amplification (myc-Halo-•–3×Flag) *template (first PCR with Fwd and Rev): pCS2 +8 myc-Halo-TEV-3x Flag *template (second PCR with Fwd and Rev extender): PCR product from first reaction *PCR product was inserted into pCS2 +8 zKeap1b | Fwd: GCTACTTGTTCTTTTGCAGGATCCACTA GTGGCGCGCCATTAATTAAAGGCCGGCCAGC Rev: CGTCATGCCCGCCGCCGCCAACATTTTAT CATCATCATCTTTATAATCAATATCATGAT Rev ext: CCATTGCTGGCCGACGGCGTCACCTCCGC CTTACA CTCCGTCATGCCCGC |

## Primers for mRNA preparation from pCS2 +8 vector

| RNA-fwd | CAATGGGGAGGGGCAATG |
|---|---|
| RNA-rev | CCAAGCGCGCAATTAACC |

## Morpholino sequence

| Name | Sequence |
|---|---|
| Nrf2a ATG Morpholino | 5'-CATTTCAATCTCCATCATGTCTCAG-3' |
| Nrf2a SPL Morpholino | 5'-ATTAAATATTATTTACCTGTTGGCT-3' |
| Nrf2b ATG Morpholino | 5'-AGCTGAAAGGTCGTCCATGTCTTCC-3' |
| Keap1a ATG Morpholino | 5'-GCCTCTTCTTTCTTGGACATATCAT-3' |
| Keap1a SPL Morpholino | 5'-GCTGCACTTAAAAATTGACTTACCT-3' |
| Keap1b ATG Morpholino | 5'-CCAACATCAGCGCGGGCACATCC-3' |
| Keap1b SPL Morpholino | 5'-GGCCCATGACCTGGAGACAAGAACA-3' |
| Control Morpholino 1 | Random control morpholino from Gene Tools. It is a mixture of many oligo sequences. |
| Control Morpholino 2 | Standard control oligo from Gene Tools 5'-CCT CTT ACC TCA GTT ACA ATT TAT A 3' |

## Morpholino validation sequence

| Name | Sequence |
|---|---|
| Keap1a MO validation sequence | ATG-ATATGTCCAAGAAAGAAGAGGCCC-GGCAGC-Firefly |
| Keap1b MO validation sequence | ATG-GGATGTGCCCGCGCTGATGTTGGC-GGCAGC-Firefly |

Schematic of construct:

## Primers for splice-blocking morpholino (SPL-MO) validation

| Name | Sequence |
|---|---|
| Keap1a SPL-MO F | 5'–ATGATATGTCCAAGAAAGAAGAGGCCCATC–3' |
| Keap1a SPL-MO R1 | 5'–CACATTTCAGTAAACCACAAAGCTGTCACC–3' |
| Keap1a SPL-MO R2 | 5'–CATTGAAGTGAGTGTTGATATACTCTGGTGC–3' |
| Keap1b SPL-MO F1 | 5'–ATGTTGGCGGCGGCGG–3' |
| Keap1b SPL-MO F2 | 5'–CACACTCACACACACACACACACAC–3' |
| Keap1b SPL-MO R | 5'–CTGAAGTTCATGTAGATGTACTCTCTGGCC–3' |

## Primers for qRT-PCR

| Gene of interest | Fwd Primer sequence | Rev Primer sequence |
|---|---|---|
| *gstpi1* | CTTCGCAGTCAAAGGCAGATG | CGCCCTTCATCCACTCTTCA |

*Continued on next page*

Continued

| Gene of interest | Fwd Primer sequence | Rev Primer sequence |
| --- | --- | --- |
| *hmox1* | ACAGAGACTGAGAGAGATTGGC | TCTATTGGCGCTCGTCACTC |
| *gsta.2* | AGAGCGAGCCATGATCGAC | ACTGTAGGTCTTTTCCTTGTTTTC |
| *abcb6a* | TACTGGGCAGTAGCTTTCGC | ACTCCATCTGTTGCTCGGAC |
| *gstpi2* | CGTGCTGGCCCTTTGAAGAT | GCTGTCCAAAGAGACATGTGG |

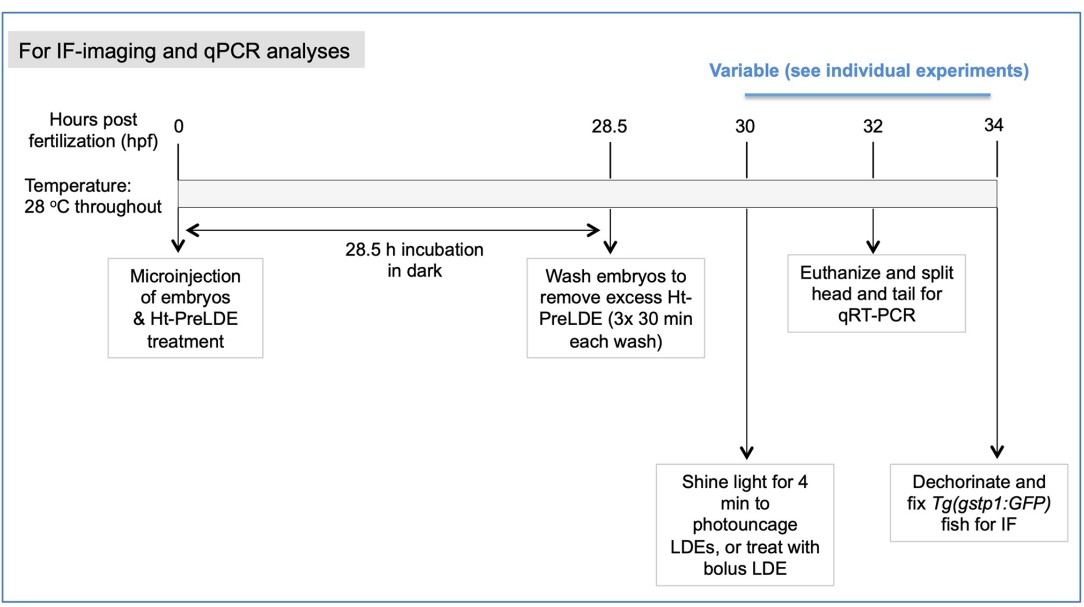

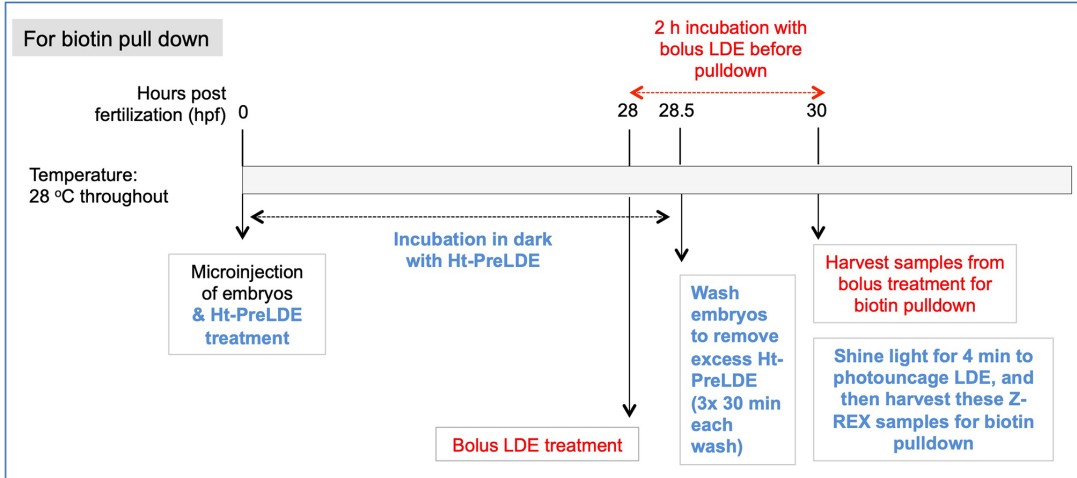

**Appendix 1—scheme 1.** Workflow for IF-imaging, qRT-PCR, and biotin pulldown experiments.

