## [Editor Report]

This is an elegant, solid, carefully performed, and substantial study investigating the divergent functions of two zebrafish paralogs of Keap1, which, in mammals, is the main negative regulator of transcription factor Nrf2, which controls cell responses to antioxidants. Curiously, one zebrafish paralog augments and the other opposes Nrf2 signaling. Creative use is made of photocaged lipid-derived electrophiles to activate one Keap1 paralog at a time without stimulating other electrophile sensors. The results will be of interest to redox biologists and those interested in the regulation of stress responses through Keap1 and Nrf2.

---

## [Decision Letter]

**Decision letter after peer review:**

[Editors’ note: the authors submitted for reconsideration following the decision after peer review. What follows is the decision letter after the first round of review.]

Thank you for submitting your work entitled "Z-REX uncovers a bifurcation in function of Keap1 paralogs" for consideration by *eLife*. Your article has been reviewed by 3 peer reviewers, and the evaluation has been overseen by a Reviewing Editor and a Senior Editor. The reviewers have opted to remain anonymous.

Our decision has been reached after consultation between the reviewers. Based on these discussions and the individual reviews below, we regret to inform you that your work will not be considered further for publication in *eLife*.

As you will see, there was a mixed opinion about the manuscript. While Reviewer #3 was enthusiastic, Reviewer #2 identified numerous shortcomings involving the execution and interpretation of the zebrafish experiments. Reviewer #2 also highlighted the lack of accounting for the distinct results reported in a 2008 J Biol Chem paper for functions of Keap1a and Keap1b. Reviewer #1 also raised several important concerns including a discrepancy between the behavior of Cys273 mutation here and in prior work as well as some other experimental weaknesses. Overall, it was felt that, to respond adequately to all of these important issues, the degree of extensive additional work would exceed the two month time frame expected for manuscripts invited for revision at *eLife*. However, if you can satisfactorily address the major criticisms at some point in the future, we would be happy too consider a version of this manuscript at that time.

*Reviewer #1:*

This is a very elegant study implicating a bifurcation in the function of the zebrafish paralogs of Keap1, the main negative regulator of transcription factor Nrf2, which controls the cellular antioxidant response (AR). The authors have used unique approaches and tools that have been previously developed in their laboratory. The findings are new and exciting, and the manuscript was a pleasure to read. However, some of the conclusions are not fully justified by the data. The authors may wish to consider addressing the following points:

1. Do any of the treatments and/or treatment combinations affect the fluorescence of GFP?

2. There seems to be induction of both gstp1 and gsta2 (Figure 3B, Figure 4C) by the light exposure alone. Has this been accounted for?

3. Does the endogenous Keap1/Nrf2 system in HEK293 cells interfere with the AR measurements?

4. Whereas the study provides an explanation for the lack of AR in the head of the zebrafish, the functional significance and the mechanism by which zKeap1a exerts its 'dominant negative function' have not been addressed and remain unclear. It has been shown for mammalian Keap1 that the substitution of C273 with I makes Keap1 unable/less able to suppress Nrf2-dependent transcription (Saito et al. Mol Cell Biol. 2015;36(2):271-84), thus providing a gain of function for Nrf2 rather than repressing Nrf2-dependent transcription. In this study, the transcriptional activity of Nrf2 (basal AR) is greater in cells expressing the Keap1 C273I mutant compared to WT Keap1 (Figure 7C) as well as in cells expressing zKeap1b compared to zKeap1a (Figure 7C,D and Figure S12), suggesting in both cases partial loss of function for Keap1 and consequently, gain of function for Nrf2. Yet, it is zKeap1a, and not zKeap1b, that has 'I' rather than 'C' at position 273 and thus is expected to resemble the mammalian Keap1 C273I mutant. How could this be rationalized?

5. The proposed model shown in Figure 8 is difficult to understand. The ascribed 'dominant negative function' of zKeap1a could be not on Nrf2-driven transcription (AR), but rather on Nrf2 turnover. An alternative mechanism to that proposed by the authors could be that a zKeap1a/zKeap1b heterodimer binds Nrf2, but is (partially) impaired to target Nrf2 for degradation thus trapping the zKeap1a/Nrf2/zKeap1b complex in the bound state, and consequently allowing Nrf2 that is synthesized de novo to accumulate and activate Nrf2-dependent transcription (AR). Testing this potential mechanism requires additional experiments.

*Reviewer #2:*

This manuscript by Long et al. investigates the functionality of T-REX in zebrafish, using a HaloTag-functionalized form of human KEAP1. The authors also examine the zebrafish homologs Keap1a and Keap1b, and they propose with a model in which: (1) Keap1a is the primary antagonist of Nrf2 under basal conditions; and (2) electrophile-modified Keap1a suppresses the Nrf2 activation that results from Keap1b modification/suppression. The authors further speculate that differences between Keap1a and Keap1b function are due to the absence of the cysteine residue that is equivalent to C273 in human KEAP1.

The authors' demonstration that T-REX works in zebrafish is fairly compelling, although it is not clear to this reviewer why extending T-REX from previous work in cultured cells and worms merits a new name for this approach (Z-REX). Ht-PreHNE conveys light-inducible expression of Nrf2 targets in zebrafish overexpressing HaloTag-functionalized KEAP1, whereas bolus HNE does not. These findings extend to other T-REX electrophiles such as Ht-PreNE and Ht-PreDE. It also appears that Nrf2 signaling in the tail but not head is responsive to these exogenous electrophiles, though the mechanistic basis for this difference remains unknown.

The authors' studies of Keap1/Nrf2 signaling in zebrafish are problematic, and my specific concerns are summarized below:

(1) The authors inexplicably fail to cite the 2008 paper by Li et al. (J. Biol. Chem., 266:3248-3255), which examines Keap1a and Keap1b function in zebrafish. This prior study demonstrates: (1) both Keap1a and Keap1b suppress Nrf2 activity in zebrafish upon their transient overexpression by mRNA injection (as assessed by gstp1 transcription); (2) both Keap1a and Keap1b promote Nrf2 degradation; (3) keap1b is the predominant homolog expressed during the first day of zebrafish development, and keap1a is transcribed at much lower levels; (4) Keap1a and Keap1b form homodimers and heterodimers; (5) cysteine residues corresponding to C288 and C273 in mammalian KEAP1 are important for the anti-Nrf2 activities of Keap1a and Keap1b, respectively.

(2) Long et al. fail to validate reagents that are essential for their studies. For example, they do not confirm that the functionality of nrf2a, nrf2b, keap1a, or keap1b MOs against their endogenous targets. Nor do they disclose the MO doses used for their studies. They instead show that keap1a and keap1b MOs can block the in vitro translation of a synthetic luciferase-encoding construct. However, the reagents are only effective at very high MO:RNA ratios for reasons that are not explained (and the actual concentrations of RNA and MO used for these studies are not described).

(3) Even if these MO reagents are truly effective, the authors over-interpret their morphant phenotypes. For instance, they state that the keap1a/keap1b double knockdown increases GFP reporter expression in the tails of Tg(gstp1:GFP) embryos relative to a control MO (Figure 2A). While the change may be statistically significant (P value = 0.0014), the magnitude appears to be only ~1.3-fold. It is unclear that this difference is biologically significant or mechanistically meaningful. The authors also state that nrf2b knockdown leads to "hyperelevated AR response in the tail." Again, the P value may be <0.0001, but fold-change is ~1.6 and the individual data points are broadly distributed. How confident can we really be that "Nrf2b countermands electrophile-induced AR-upregulation"? These issues extend to their cell culture experiments as well. For example, since co-expression of Keap1a and Keap1b does not suppress basal antioxidant responses to a greater extent than Keap1a alone (Figure 7D), the authors conclude that Keap1a affects the ability of Keap1b to affect these responses. An alternative explanation (and arguably more likely) is that Keap1a overexpression alone can maximally suppress basal antioxidant responses, especially since Keap1b appears to be less active in these assays (Figure 7C).

(4) Some of the other MO phenotypes are difficult understand in light of other results in the paper and/or missing information. For instance, Figure 1D shows that nrf2a is expressed at much higher levels than nrf2b in the zebrafish embryo tails. Yet nrf2a and nrf2b MOs have essentially the same effects on tail GFP expression in Tg(gstp1:GFP) embryos (Figure 2A)-a surprising result if "Nrf2b is a minor contributor" as described by the authors in the Discussion section. Were the nrf2a and nrf2b MOs used at the same dose? Do they have the same knockdown efficiency? Also, in Figure 2A-B. the keap1b MO does not alter tail GFP expression in Tg(gstp1:GFP) embryos in basal conditions, but it suppresses the differential effects of NE and DMSO on this reporter. If the morphant phenotypes reflect on-target activities, this means that Keap1b must promote NE-mediated activation of antioxidant responses, an idea that counters the conventional wisdom that NE directly suppresses the ability of Keap1 proteins to induce Nrf2 degradation. The authors should provide a mechanism that can account for these findings.

(5) Important controls are missing from some experiments. For example, the authors examine how "bolus LDE treatment results in different extent of hKEAP1-labeling" in Figure 6. Presumably they are using the HNE(alkyne) and NE(alkyne) labeling and Cy5 click chemistry results as evidence that these electrophiles efficiently modify KEAP1 expressed in zebrafish embryos. However, without testing the effects of these electrophiles on embryos that are not expressing exogenous KEAP1, it is not possible to know how much of the observed Cy5 signal is due to KEAP1 labeling vs. non-specific labeling of endogenous proteins.

(6) The authors overlook inconsistencies in their observations. For example, the ability of Ht-PreHNE to convey light-inducible endogenous antioxidant responses in both the tail and head of zebrafish embryos expressing HaloTag-functionalized KEAP1 (Figure 3B) seems to contradict the inability of Ht-PreHNE to convey light-inducible GFP expression in the heads of Tg(gstp1:GFP) embryos expressing the same construct (Figure 3C-D). If the GFP reporter does not faithfully recapitulate endogenous antioxidant responses, what do the differences in tail and head GFP expression mean? This is a particularly important question since the Tg(gstp1:GFP) embryos are used extensively in the authors' study.

(7) Even if the gstp1:GFP reporter is a reliable surrogate for endogenous antioxidant signaling, the authors' reliance on immunofluorescence imaging to quantify these responses is questionable. This approach is complicated by the autofluorescence and non-specific antibody binding of zebrafish embryos, and a more rigorous approach would be to assess reporter expression by quantitative western blots or qRT-PCR.

(8) The authors should also explain why their studies of KEAP1 function have focused on zebrafish embryos that are at least 24 hours post fertilization (hpf) (and certain T-REX experiments utilized 48-hpf embryos). Zebrafish studies using mRNA-mediated overexpression ae typically limited to the first 24 hours of development since the exogenous mRNA is rapidly degraded. Accordingly, the 2008 J. Biol. Chem. paper by Li et al. examined Keap1a and Keap1b activity in zebrafish gastrulae (8 hpf).

(9) The electrophile-labeling results for Ht-PreHNE and Ht-PreNE shown in Figure 7B and Supp. Figure 13 are puzzling. They suggest that ~20% of KEAP1 is HNE(alkyne)-modified and ~12% is NE(alkyne)-modified. Since it is believed that electrophile-mediated activation of NRF2 acts by stoichiometrically suppressing KEAP1 function, the authors should explain how these results relate to the ability of Ht-PreHNE and Ht-PreNE to induce antioxidant responses in embryos expressing HaloTag-functionalized KEAP1 (e.g., Figure 3, Figure 4, Supp. Figure 4, Supp. Figure 7, and Supp. Figure 9).

(10) The authors' studies of Keap1a and Keap1b function in HEK293T cells assumes that both zebrafish proteins maintain their functions in mammalian cells. There is no guarantee that this will be the case, as there will be species-specific structural differences in E3 ligase components and their substrates (not to mention different culture temperatures). Based on these cell-based assays, the authors claim that Keap1b is less efficient than Keap1a at suppressing antioxidant responses. However, this interpretation is at odds with the previous studies by Li et al. (J. Biol. Chem., 2008) that show Keap1a and Keap1b have comparable effects on Nrf2 stability and antioxidant responses in zebrafish embryos. The authors should comment on this difference.

(11) In general this manuscript is very challenging to read, and as a result, mechanisms proposed by the authors are difficult to follow. For example, the scheme in Figure 8 suggests that the electrophile-modified form of Keap1b potentiates Nrf2 function. Is this really what the authors mean to convey, as opposed to proposing that electrophile modifications of Keap1b prevent this E3 ligase component from promoting Nrf2 degradation? Or are the authors proposing that modified Keap1b can negatively regulate a suppressor of Nrf2 function? The authors also suggest the Keap1a is "a dominant-negative regulator of the antioxidant response in both basal and electrophile-stimulated states." However, Keap1-mediated Nrf2 degradation under basal conditions is believed to be a catalytic rather than a dominant-negative mechanism. And if electrophile-modified Keap1a acts as a dominant-negative regulator, what is it inhibiting in a stoichiometric manner?

(12) For the multiple reasons described above, authors' model for Keap1a and Keap1b function (Figure 8) does not have strong experimental support. Their model is also mechanistically counterintuitive. If Keap1a is the primary antagonist of Nrf2/antioxidant responses under basal conditions, then electrophile-induced Nrf2 signaling should be predominantly due to Keap1a modification and suppression. The authors propose instead that electrophile modification of the purportedly less effective antagonist, Keap1b, is the primary driver of increased Nrf2 activity and that modified Keap1a actually suppresses Nrf2 function further. It is difficult to see how electrophiles could mount 3- to 4-fold increases in antioxidant responses under these mechanistic constraints.

*Reviewer #3:*

Investigating specific protein functions in electrophile sensing pathways in cells is challenging in terms of deciphering on vs off-target interaction of electrophiles with your target of interest vs other targets. In this study, the authors investigate the specific function of Keap1 paralogs in zebrafish using their Z-REX system and show different functions between Keap1a and Keap1b paralogs, which they also recapitulate in cell culture models. The study is performed very rigorously and the conclusions of the paper are supported by the data. This review recommends publication as is.

[Editors’ note: further revisions were suggested prior to acceptance, as described below.]

Thank you for resubmitting your work entitled "Z-REX uncovers a bifurcation in function of Keap1 paralogs" for further consideration by *eLife*. Your revised article has been evaluated by three reviewers, including a a new reviewer with zebrafish expertise who substituted for the prior zebrafish specialist.

Summary:

This is a very interesting study implicating a bifurcation in the function of the zebrafish paralogs of Keap1, the main negative regulator of transcription factor Nrf2, which controls the cellular antioxidant response (AR). The authors' experimental approach is unique and very elegant. The manuscript has been improved but there are some remaining issues that need to be addressed, as outlined below. The principal remaining concern relates to the zebrafish experiments, and in particular, ruling out off-targets in the morpholino knockdown studies. As stated below, it is important to meet the Stanier guidelines paper to ensure the robustness of these experiments. In addition, we hope you can address the other issues by text revision, eg. toning down the conclusions and/or providing alternative explanations for the findings as indicated.

Essential revisions:

– The revised document includes a number of 'validation' studies, but they are not consistent with expectations in the field in terms of controls and other validation. That does not mean their interpretation is necessarily wrong, it's that from the controls they do present, we do not know the boundaries where the work is likely limited in interpretation. Morpholino validation includes on-targeting and off-targeting questions. The authors present A LOT of work, but there are issues with both. (A) the authors appear to spend all of their time worried about on-targeting/efficacy measurements. For the translational blocking reagents, they do appear to be inhibiting their targets (and they now report their dosing per embryo). But their splice-site reagents are showing 30% or more wt RNA still present in all but one circumstance. Will only 70% knockdown be enough to show a phenotype? Not clear. (B) Off-targeting questions. In the Stainier guidelines paper, there are clear approaches to experimental design to address this question. Comparing to a mutant is the preferred approach, followed by RNA Rescue etc. We see none of this in this manuscript, or even a reason why they did not do these logical experiments.

– Statistical assessment – We are concerned about the confidence interval numbers (we think that's what's shown) compared to the data distribution shown in this manuscript. Just one example – Figure 2A: comparing control MO1 with Keap1a+Keap1b MO injections. This should show the largest differential. Seems to be about 30 individual data points. The authors claim these distributions are 99.86% likely to be different. But the visual data set suggests that the only primary difference is that 1/30 data points are above 1.6x in control MO1 versus 4/30 data points are above 1.6x. Visually, the numerical value just does not align with the graphical representation they present. This same issue is found throughout the manuscript. Indeed, the data distribution graphs raise a key question of study design – how many times were these experiments independently run? We are not sure combining biological replicates is the best way to go, if that's what they did. It is not clear from the text how they derived the data points they show.

– Why was myc-Nrf2 used for the experiments shown in Figure 7, whereas HA-Nrf2 was used for the experiments shown in Figure 8?

– Figure 7. The authors say in the text (beginning on line 454): 'Intriguingly, when zKeap1b was co-transfected with sub-saturating amounts of zKeap1a, no decrease in basal AR was observed relative to zKeap1a alone (Figure 7D), implying that zKeap1a somehow affects the ability of zKeap1b to suppress AR.' Is it possible that the absence of C273 makes zKeap1a a more efficient repressor, because this cysteine when present (as in zKeap1b) senses endogenous electrophile(s)/oxidant(s) at basal state? If repression of AR by zKeap1a is already maximal (as suggested by the data), is co-transfection with zKeap1b expected to have any further effect?

– Figure 8. The authors say in the text (beginning on line 490): 'We found that zKeap1b accumulated Nrf2 in the basal (i.e., non-electrophile-stimulated) state, whereas relative to zKeap1b, zKeap1a accumulated less Nrf2, and zKeap1a/zKeap1b accumulated an amount of Nrf2 that was significantly more than Keap1a alone and less than zKeap1b alone (Figure 8A-B).'

• Could these data be interpreted that zKeap1a degrades Nrf2 better whereas zKeap1b does not? This would be consistent with a scenario where the absence of C273 makes zKeap1a a more efficient repressor, because this cysteine senses endogenous electrophiles/oxidants at basal state, causing partial inactivation of zKeap1b.

• Is the presence of less Nrf2 bound to zKeap1b upon treatment with NE necessarily a consequence of release of Nrf2? Was an 18-h treatment necessary to see this effect? It seems a very long time; we would have thought that if NE was causing a release of Nrf2 from zKeap1b, the release should be evident at a much earlier time point. Can the authors be fully confident that release does occur when the released protein has not been observed/accounted for?

• There seems to be less HA-Nrf2 following NE treatment in the input samples (Suppl Figure 15), and it is thus possible that the NE treatment affects the turnover of HA-Nrf2. The normalization for the input addresses this; nonetheless, the semi-quantitative nature of the immunoblotting technique should be kept in mind.

• Do the authors know the identity of the ~37 kDa fragment in the anti-FLAG blot?

• The results from the zKeap1a/zKeap1b co-transfection experiment are not straightforward to interpret, because Keap1 is a dimer, and thus the simultaneous presence of several dimeric combinations (e.g. zKeap1a/zKeap1b, zKeap1a/zKeap1a, zKeap1b/zKeap1b) is possible.

[Editors’ note: further revisions were suggested prior to acceptance, as described below.]

Thank you for submitting your work entitled "Z-REX uncovers a bifurcation in function of Keap1 paralogs" for consideration by eLife. Your article has been reviewed by 2 peer reviewers, and the evaluation has been overseen by a Reviewing Editor and a Senior Editor. The reviewers have opted to remain anonymous.

Our decision has been reached after consultation between the reviewers. Based on these discussions and the individual reviews below, we regret to inform you that your work will not be considered further for publication in eLife.

While two of the original three reviewers are now satisfied, the third reviewer who is an expert on zebrafish has indicated that the zebrafish morpholino experiments are insufficiently rigorous to be acceptable. Below are the latest reviewer comments. We regret having to communicate this disappointing news. We hope that if you ultimately choose to address Reviewer 3's concerns with the requisite additional experimental data, that you will come back to eLife with a manuscript that includes this information when ready.

*Reviewer #1:*

The authors have addressed my remaining concerns appropriately.

*Reviewer #3:*

None of my technical concerns tied to either the on-target or off-target effects were addressed.

1) On-target. The maximal described knockdown was 70%. In our experience, effective morpholinos are readily able to go well beyond 90 or 95% knockdown - one of the key distinguishing features of MOs over siRNA is the normal ability to be well beyond 80% knockdown.

2) Off-target. The authors were given several options to address this concern, and they chose to not offer any new data. The specific suggestion to follow the Stainier guidelines was rebutted, arguing they did not have mutants etc.

I recommend rejection of this manuscript. The zebrafish work does not achieve the level of rigor expected in the field. If the authors were to come back after confirming their results using mutants and/or true rescue experiments with morpholinos that show strong on-target efficacy, the paper could be considered appropriate to publish in eLife.

---

## [Author Response]

[Editors’ note: the authors resubmitted a revised version of the paper for consideration. What follows is the authors’ response to the first round of review.]

As you will see, there was a mixed opinion about the manuscript. While Reviewer #3 was enthusiastic, Reviewer #2 identified numerous shortcomings involving the execution and interpretation of the zebrafish experiments. Reviewer #2 also highlighted the lack of accounting for the distinct results reported in a 2008 J Biol Chem paper for functions of Keap1a and Keap1b. Reviewer #1 also raised several important concerns including a discrepancy between the behavior of Cys273 mutation here and in prior work as well as some other experimental weaknesses. Overall, it was felt that, to respond adequately to all of these important issues, the degree of extensive additional work would exceed the two month time frame expected for manuscripts invited for revision at eLife. However, if you can satisfactorily address the major criticisms at some point in the future, we would be happy too consider a version of this manuscript at that time.Reviewer #1:This is a very elegant study implicating a bifurcation in the function of the zebrafish paralogs of Keap1, the main negative regulator of transcription factor Nrf2, which controls the cellular antioxidant response (AR). The authors have used unique approaches and tools that have been previously developed in their laboratory. The findings are new and exciting, and the manuscript was a pleasure to read.

We are most grateful to the Reviewer for a series of mostly very fair and careful points. We have amended the manuscript with some data and in areas of the text where necessary. We were, however, a little perplexed by the Reviewer’s final main point.

We hope our response is satisfactory. We would be very happy to engage with the reviewers in further open discussion.

However, some of the conclusions are not fully justified by the data. The authors may wish to consider addressing the following points:1. Do any of the treatments and/or treatment combinations affect the fluorescence of GFP?

We are not detecting GFP fluorescence; we are using IF (please see legend to Figure 2, for example, where we stated “Note: GFP expression was detected using immunofluorescence (IF) in fixed fish, analyzed by red fluorescence. The IF protocol is used because auto-fluorescence in the green channel is high in fish and prevents accurate quantitation”). This strategy also obviates the need to worry about GFP quenching (or even activation), which is another benefit of using the IF strategy (above reducing background, etc.). GFP is also not labeled by HNE under Z-REX, and GFP and other FPs^1^ are also compatible with T-REX in cell culture, giving similar outputs to what is observed using luciferase as a reporter^2^. Furthermore, for the most part, these outputs are also independently supported by qRT-PCR experiments assessing endogenous genes (please see Figure 3B, Figure 4C, Supporting Figure 2B, Supporting Figure 7B). We further note that we can modulate the changes/lack thereof (often in both directions) observed using MOs, arguing these results are independent of GFP; these data correlate with known outputs and also data from IP studies now newly added (Figure 8A-B and Supporting Figure 15).

To further support the statements above, we provide Author response image 1:

**Author response image 1. sa2fig1:** Comparison between live imaging (A) and whole-mount immunofluorescence (IF) (B). Scale bars, 500 μm. (A) Live *Tg(gstp1:GFP)* embryos (1.5 dpf, injected with control MOs) were dechorionated and placed on an agarose pad and imaged using GFP fluorescence (Ex. 470 nm, Em. 525 nm) and bright field. Signal to noise (S/N) ratio was estimated by dividing fluorescence intensity of the fluorescent signal by the fluorescence intensity of the surrounding area for 4-6 fluorescent areas using Image-J (NIH). (B) *Tg(gstp1:GFP)* embryos (1.5 dpf, injected with control MOs) were dechorionated, fixed, and immunostained for GFP as described in the original manuscript. AlexaFluor568 (Ex. 560 nm, Em. 630 nm) channel shows GFP signal. GFP localization reported by GFP intrinsic fluorescence [in (A)] is similar to that reported by red fluorescence from IF in this figure. S/N ratio was calculated as in (A).

2. There seems to be induction of both gstp1 and gsta2 (Figure 3B, Figure 4C) by the light exposure alone. Has this been accounted for?

The fold-induction by light alone (which is only seen in qPCR) is minimal, compared to that which occurs upon Z-REX. Unfortunately, the effect of light alone and the effect of compound alone cannot account for what occurs during Z-REX as there may other hidden variables that such simple comparisons cannot account for. Thus, to fully account for all potential hidden variables, we also deploy ‘perfect’ negative controls for Z-REX, by deploying a P2A-mediated ‘non-fused construct’ (Figure 3C). We note that this control takes into account both single variables (light and small-molecule probe), and also other “hidden variables”, such as effect of photouncaging itself, released of HNE to system. Data with this non-fused construct unambiguously validate that the responses observed are due to on-target LDE modification [please see Figure 3C-E (IF-imaging data) and also Supporting Figure 7A-B (qRT-PCR data sets) in the revised manuscript]. We showed this control using two different readouts (IF-imaging and qRT-PCR), two different Z-REX-LDE-precursor probes, evaluating multiple endogenous downstream AR-genes (Supporting Figure 7B) and also compared the outcomes under bolus dosing (Figure 3D-E). We have now made these more clearly with an accompanying illustration in Figure 3C.

3. Does the endogenous Keap1/Nrf2 system in HEK293 cells interfere with the AR measurements?

This is a fair question, which also worried us for some time in the past. However, our data are consistent between fish and cells (in this current manuscript as well as in our recent publications and ongoing manuscripts that involve the use of both cells and fish^3^) in instances where the endogenous : ectopic protein component differs widely. They further agree with overexpression of the different-paralogs in zebrafish embryos, derived from another laboratory^4^. Such consistency is the first clear piece of evidence that there is little interference between the different ectopic constructs and the endogenous copy.

Furthermore, our data from 7 recent publications and additional work pending publications involving Keap1/Nrf2/AR-pathway studies^5^, where we used multiple orthogonal assays investigating endogenous AR-driven downstream genes as well as using ectopic transfection systems all point to the fact that there is no such interference. In each case, we have also consistently established the optimum plasmid ratios for desired ectopic transgenes (typically Keap1, Nrf2, and reporters) such that measurements are made well within the dynamic range of the AR-reporter assay. In the current manuscript, Figure S14B showed such a determination of the amounts/ratios of Keap1a/b-plasmids.

We also assessed the importance of the endogenous copy of Keap1 in cells as follows: western blot analyses show that ectopic Keap1-overexpression is significantly higher than the endogenous construct for any situation examined (Author response image 2). Thus, one would expect endogenous human Keap1 to be unimportant in our cell-based assays. *We investigated this proposition systematically as follows:* Nrf2-alone-overexpression gives a high level of basal AR (around 1000-fold higher than in a transfection with an empty vector) (Author response image 2, left panel). Under these conditions, AR is not dependent on endogenous Keap1, as 50% Keap1-knockdown by 2 different siRNAs targeting Keap1 had no overall significant effect suppressed (up to 100-fold) by co-overexpression of the specific Halo-Keap1 construct (data in our original manuscript). This ectopic Keap1-dependent AR-state is also independent of endogenous Keap1-activity, as knockdown of the endogenous protein has no effect on AR either (50% knockdown by 2 different siRNAs targeting Keap1) (Author response image 2 right panel, and Author response image 2). Thus, we conclude based on stoichiometry and effect on AR; that endogenous Keap1 causes no issue for our interpretations of these data. Taken together, we believe that both our previously published work and additional reviewers’ data relevant to this current manuscript fully address this reviewer’s concern.

**Author response image 2. sa2fig2:** Endogenous human Keap1/Nrf2 expression does not interfere with AR measurement in cell-based assays. (**A**) *Left*, Expression of ectopic Keap1 (labeled here as ‘HTK’, designating the construct used: Halo-TeV-Keap1) is large relative to endogenous hKEAP1 present in cells. HEK293T cells were transfected as indicated in the dual luciferase assay described in Figure 7C-F (original manuscript) and after 42 hours harvested for western blotting. *Right*, quantification of endogenous (blue bars) and overexpressed Keap1 (red bars) for each transfection condition from n=6 replicates. The affinity of the antibody to each Keap1 variant (Supp. Figure 1D, original manuscript) was used to normalize the Keap1 signal for zKeap1a and zKeap1b. The horizontal dotted line represents the average level of overexpressed hKEAP1. (**B**) *Left,* introduction of ectopic Nrf2 alone increases the AR signal by 1000-fold (note: maximal AR is normalized to 100; lowest AR is 0.1). *Center and Right*; using the dual luciferase assay described in Figure 7C-F (original manuscript), two different siRNAs targeting endogenous hKEAP1 were used to reduce endogenous hKEAP1 levels, while the cells were also transfected to express hNrf2 (*Graph at center*) only, or low levels of zKeap1a/b (*Graph on right*), using transfection conditions as in Figure 7D,F (original manuscript). siRNAs have little effect on AR output in either condition. si(control) i, ii, iii correspond to Santa Cruz siRNA controls “A, C, and E (vendor’s labels)”, respectively. si(hKEAP1) i and ii correspond to Dharmacon siGENOME human KEAP1 siRNAs D-012453-03 and D-012453-04. Dharmafect Duo was used for transfection. (**C**) *Left,* representative blot for siRNA knockdown efficiency assessment under assay conditions deployed in the original manuscript (data shown is from a single continuous blot); *Right,* quantification (n=4). Horizontal dotted line represents average level of Keap1 signal of all siControls.

4. Whereas the study provides an explanation for the lack of AR in the head of the zebrafish, the functional significance and the mechanism by which zKeap1a exerts its 'dominant negative function' have not been addressed and remain unclear. It has been shown for mammalian Keap1 that the substitution of C273 with I makes Keap1 unable/less able to suppress Nrf2-dependent transcription (Saito et al. Mol Cell Biol. 2015;36(2):271-84), thus providing a gain of function for Nrf2 rather than repressing Nrf2-dependent transcription. In this study, the transcriptional activity of Nrf2 (basal AR) is greater in cells expressing the Keap1 C273I mutant compared to WT Keap1 (Figure 7C) as well as in cells expressing zKeap1b compared to zKeap1a (Figure 7C,D and Figure S12), suggesting in both cases partial loss of function for Keap1 and consequently, gain of function for Nrf2. Yet, it is zKeap1a, and not zKeap1b, that has 'I' rather than 'C' at position 273 and thus is expected to resemble the mammalian Keap1 C273I mutant. How could this be rationalized?

Our newly-acquired data (Figure 8A-B and Supporting Figure 15) show that following electrophile treatment, Keap1a is unable to release Nrf2 bound to it (explaining a previous report on the non-permissivity of zKeap1a to electrophile-induced AR^4^). Keap1b can release a large amount of bound Nrf2 upon electrophile treatment. In the heterodimer (a mix of Keap1a and Keap1b), release of Nrf2 is also inhibited. This explains how Keap1a can regulate AR in a dominant-negative fashion, as the AR increase upon electrophile labeling of zKeap1b is ascribable to release of Nrf2 from zKeap1b.

Our data in the basal state for cells overexpressing only hKEAP1(C273I) are overall qualitatively consistent with that reported in Saito et al. *Mol Cell Biol.* 2015; 36(2):271-84 on mKeap1(C273I). These authors did not look at interaction between different Keap1 constructs, like we did.

In our mixing studies, we found that hKEAP1(C273I) mutant shows dampening of the electrophilic response of hKEAP1(WT), similarly to how zKeap1a exerts a dampening of the electrophilic response of zKeap1b [or hKEAP1(WT)], both in cells and fish (i.e., suppression of electrophile-induced AR-upregulation in mixed-plasmid systems in cells, compared to zKeap1b [or hKEAP1(WT)]-overexpression alone in Figure 7F, and alleviation of electrophile-induced AR-upregulation in double-knockdown systems in fish, compared to Keap1b-knockdown alone, in Figure 2B left panel). Note that the fact that hKEAP1(C273I) and zKeap1a show the same effect even though their respective abilities to suppress basal AR are dissimilar, shows that the effect of C273I is independent of basal AR (Figure 7C).

We believe that this difference in basal AR-suppression abilities is due to other mutations that allow zKeap1a to retain its basal function better than hKEAP1 C273I. We have tried to ID the specific residue in zKeap1a that would allow for this increase in Nrf2suppression function in the basal state (through analysis of structural and sequence data), but it appears to be not due to a single mutation/residue. We show these data as Author response image 3. We decided to not include these data/discussions as we feel they deviate from the main focus of the manuscript as we try to frame the Discussion back to heterozygous systems both in fish and human. What these data do tell us though is that overall it is quite surprising that a single mutation (C273I) is responsible for the regulatory role of Keap1/paralogs (and how they bifurcate) under electrophile-stimulated state.

Similar reasoning applies to zKeap1a/1b. As shown in Figure 7E and 7F, Keap1b can mount a 2-3-fold response in electrophileinduced AR, although zKeap1b shows a lower level of basal AR suppression (Figure 7C). However, in the presence of zKeap1a, where if anything, basal AR is lower (Figure 7D), indicating that the output in terms of AR-upregulation is muted. Accordingly, the electrophile-induced AR upregulation due to zKeap1b [which was ~3-fold (Figure 7E), even though basal AR due to zKeap1b was higher (Figure 7C)] is weakened.

Given that release of Nrf2 appears to be the predominant means through which AR is mounted, at least in the fish, understanding how zKeap1a functions in conjunction with zKeap1b in regulating Nrf2/AR is itself an important advancement. However, given the complexity of the system, the overall poor understanding of Keap1 structurally, and the fact that we have covered a large amount of ground already in this revised manuscript, we think that some of the remaining interesting questions that the reviewer raised in this specific query are questions that can be pursued in another paper.

We hope that our rationalization herein, supported further by additional data and newly-acquired revision data, fully address the reviewer’s concern.

**Author response image 3. sa2fig3:** Initial explorations regarding potential rescue of basal AR-activity of hKEAP1 C273I through mutation. (A) *Top;* No full-length hKEAP1 crystal structure is available, therefore crystal structure of human Kelch-like protein-11 (PDB: 4AP2) was used as a model onto which the hKEAP1 sequence was threaded using Swiss-Prot. *Bottom;* Amino acid sequence alignment (performed using MEGA-X) between hKEAP1, zKeap1a, and zKeap1b. Residues with side chains within 5 Å of C273 (magenta) are shown and boxed in red in the sequence alignment shown below. These residues served as putative residues for potentially rescuing the basal AR-activity of hKEAP1 C273I. (B) The cell-based AR assay from Figure 7 C-F (original manuscript) was used to determine the extent of basal AR suppression elicited by each double/triple mutant. C273I is the single point mutant that recapitulated zKeap1a behavior but lost the ability to suppress basal AR to the same extent as hKEAP1 WT can do. The double, quadruple, and triple mutants (last three sets of hKEAP1-mutant on graph) introduce additional zKeap1a residues into hKEAP1 C273I.

5. The proposed model shown in Figure 8 is difficult to understand. The ascribed 'dominant negative function' of zKeap1a could be not on Nrf2-driven transcription (AR), but rather on Nrf2 turnover. An alternative mechanism to that proposed by the authors could be that a zKeap1a/zKeap1b heterodimer binds Nrf2, but is (partially) impaired to target Nrf2 for degradation thus trapping the zKeap1a/Nrf2/zKeap1b complex in the bound state, and consequently allowing Nrf2 that is synthesized de novo to accumulate and activate Nrf2-dependent transcription (AR). Testing this potential mechanism requires additional experiments.

We have now performed experiments where we measured release of Nrf2 from zKeap1-paralogs post electrophile treatment (Figure 8A-B, and Supporting Figure 15). We found that only zKeap1b of the two zKeap1-paralogs, could release Nrf2 following electrophile exposure. Furthermore, zKeap1b could build up a larger amount of associated Nrf2 than zKeap1a in the steady state (Supporting Figure 15). The mixed state, consisting of equal amounts of Keap1a and Keap1b relative to above, was able to accumulate Nrf2 bound to Keap1, but could *not* release Nrf2 following electrophile treatment (Figure 8A-B, and Supporting Figure 15). Given these data, we believe that the release model is sufficient to explain our data. It is worth noting however that we cannot rule out that the zKeap1b protein is also more susceptible to inhibition than zKeap1a, although clearly this would at most have to be due to a partitioning since zKeap1a has the same extent of electrophile-labeling/sensing as zKeap1b (Figure 7B), and around 40% of Nrf2 bound to zKeap1b is released (Figure 8B) and less than ~40% occupancy on Kzeap1b (Figure 7B) is sufficient to trigger maximal AR-upregulation. However, as the heterodimeric state does not release Nrf2 bound to it (Figure 8B), and no AR increase is observed in this state either (Figure 7F), these data could not be fully explained by the alternative model proposed by the reviewer.

Nonetheless, we do understand that complex processes do not always break down to a single factor, so we have written our revised Discussion section to mention that other factors (such as inhibition of zKeap1b, or zKeap1a marginally) could contribute to AR.

Reviewer #2:This manuscript by Long et al. investigates the functionality of T-REX in zebrafish, using a HaloTag-functionalized form of human KEAP1. The authors also examine the zebrafish homologs Keap1a and Keap1b, and they propose with a model in which: (1) Keap1a is the primary antagonist of Nrf2 under basal conditions; and (2) electrophile-modified Keap1a suppresses the Nrf2 activation that results from Keap1b modification/suppression. The authors further speculate that differences between Keap1a and Keap1b function are due to the absence of the cysteine residue that is equivalent to C273 in human KEAP1.The authors' demonstration that T-REX works in zebrafish is fairly compelling, although it is not clear to this reviewer why extending T-REX from previous work in cultured cells and worms merits a new name for this approach (Z-REX).

The change in name allows us to more clearly distinguish and/or refer to experiments performed in fish and cells in a paper where the twain is conducted.

Ht-PreHNE conveys light-inducible expression of Nrf2 targets in zebrafish overexpressing HaloTag-functionalized KEAP1, whereas bolus HNE does not.

With respect, this is untrue. Please see Figure 3A, B, C-D, and Figure 5B for example. However, the reviewer may be aware that permeation is a large issue for drug efficacy in general. Hence in live organisms, often REX-technologies are more able to efficiently target the intended protein than bolus dosing, as we showed, for instance, with ectopic human-Keap1 in *C. elegans*^6^.

These findings extend to other T-REX electrophiles such as Ht-PreNE and Ht-PreDE.

This is partly true but we would like to clarify that these other electrophiles, where studied, also upregulate AR upon bolus exposure.

It also appears that Nrf2 signaling in the tail but not head is responsive to these exogenous electrophiles, though the mechanistic basis for this difference remains unknown.

We believe that we have explained the basis for the differences in the responsivity. This and many of the reviewer’s comments were below appear to stem from misunderstanding: e.g., aspects of data that the reviewer stated to be missing were actually present in our original manuscript, and issues that the reviewer stated that we had ignored were explained in the text. We have also doubled our input to make all of the points more obvious and improve clarity in our data discussion.

The authors' studies of Keap1/Nrf2 signaling in zebrafish are problematic, and my specific concerns are summarized below:1) The authors inexplicably fail to cite the 2008 paper by Li et al. (J. Biol. Chem., 266:3248-3255),

The cited paper (for which we believe the citation volume is incorrect, and should be *J Biol Chem* 2008; 283:3248-3255) is from Yamamoto, whose more recent work (building on these and other data from his laboratory) is cited by us in the original manuscript (2011 Tsujita et al. *Genes Cells*). We note that the timescales, differentiation, and modus operandi of the experiments were largely different between our own paper and the aforementioned JBC paper. That being said, overall where there are, albeit minor, overlaps between the two papers, our data and the data from the JBC paper do agree. Note that our paper is much more quantitative and investigates the interplay between the two proteins, especially in the electrophile-stimulated state. The JBC paper does *not* study electrophile-regulated state.

which examines Keap1a and Keap1b function in zebrafish. This prior study demonstrates: (1) both Keap1a and Keap1b suppress Nrf2 activity in zebrafish upon their transient overexpression by mRNA injection (as assessed by gstp1 transcription);

This experiment was performed using overexpression (of Keap1 individually, in the presence of ectopic Nrf2) *at a very early stage of development, where AR is often globally high* (see Figure 4B of the JBC paper referred by the reviewer, but since this is with Nrf2(*fragment*)-GFP injected fish, the direct relevance is not clear), and cells have clearly not differentiated, so regulation/responsivity may be very different to our systems. However, the overall message is the same as we conclude, namely that, both Keap1 molecules *are active* in the basal state.

As the JBC paper did not look to a huge extent to the interplay between the two, nor does it look at electrophile sensitivity, nor does it perform much quantitative analysis, the data therein are not directly relatable to nor comparable against our data.

(2) both Keap1a and Keap1b promote Nrf2 degradation;

We respectfully do question the relevance of this (and the above) point, unless it is to say that our analysis agrees with previously published literature. We have now cited the JBC 2008 paper, but in the spirit of experimental rigor and to avoid misleading the readers, we are reluctant to not draw comparisons where experimental setup/conditions are not directly comparable and/or quantitative analyses have not been performed. In this specific degradation-related data in the JBC paper, the data have not been validated using any standard protein degradation controls (e.g., inhibitors, ubiquitination), so it does not directly show degradation per se.

(3) keap1b is the predominant homolog expressed during the first day of zebrafish development, and keap1a is transcribed at much lower levels;

Our experiment was done at 36 hpf, and indeed our overall expression agrees with the data in the JBC paper at this time point as much as it is possible to do so, as expression varies quite significantly at this time (See Figure 6B of the JBC paper; these data are also derived from only a single data point).

(4) Keap1a and Keap1b form homodimers and heterodimers;

These experiments were performed in vitro and do not show that dimerization occurs in embryos. This outcome is, in any case, *not* at odds with our model, and the dimerization of Keap1 was never called into question by us (and is indeed it is known that Keap1 is a dimeric protein).

(5) cysteine residues corresponding to C288 and C273 in mammalian KEAP1 are important for the anti-Nrf2 activities of Keap1a and Keap1b, respectively.

We are not sure what the relevance of this statement is (as we do not change these residues in Keap1a/b).

We do apologize if we somehow offended the reviewer by not citing this paper (which mostly deals with information that is covered in the other literature we cited). As stated above, we are happy to cite this work and hope that the reviewers and the Editors can see that there is an overall agreement between our data and the JBC paper data in basal-AR state, but it is clearly unreasonable for the reasons delineated above to make direct comparisons neither at quantitative level, nor at electrophilestimulated state which was not at all touched upon in the JBC paper.

2) Long et al. fail to validate reagents that are essential for their studies. For example, they do not confirm that the functionality of nrf2a, nrf2b, keap1a, or keap1b MOs against their endogenous targets. Nor do they disclose the MO doses used for their studies. They instead show that keap1a and keap1b MOs can block the in vitro translation of a synthetic luciferase-encoding construct. However, the reagents are only effective at very high MO:RNA ratios for reasons that are not explained (and the actual concentrations of RNA and MO used for these studies are not described).

We have now additional independent validation of MOs targeting zKeap1a and zKeap1b, using whole-mount immunofluorescence analysis (Supporting Figure 5A-B) [using anti-Keap1 antibodies that we also in parallel validated through ectopic expression in cell culture (Figure 7, Supporting Figure 1D, 14A-B; and Author response images 2 and 3)], as well as by RT-PCR for SPL-MOs (Supporting Figure 4A-B). These additional data have added robust validations, beyond the original data validating zKeap1a and zKeap1b-targeting MOs using luciferase translational assays (Supporting Figure 3B; figure numbering in the revised manuscript) which are also routinely used in the zebrafish field for validating translationally-blocking MOs.

In the revised manuscript, we have now listed the MOs all clearly: in figure legends beyond the Source Data file; including the mRNA amount injected (1.3-1.6 mg/ml; 2 nl volume) [0.5 mM, (approximately 2.8 mg/ml), 2 nl] (Figure 2; Supporting Figure 2, 3, 4, and 5).

The reviewer likely missed our validation data of Nrf2-MOs and they may not also be aware of the state of Nrf2 antibodies in the field. Briefly, Nrf2 antibodies, except for the mouse protein, are *not* considered effective (even in mammalian cells on the human protein), and so AR effects are the best way to validate Nrf2 manipulation. Given the number of papers citing how much Nrf2 antibodies have misled people even in cells (e.g., see Lau et al., 2013 *Antioxid Redox Signal* 18 91-93), we do not use them in our lab. But our previous manuscript clearly showed that zNrf2a/b-gene knockdown negatively impacts AR in the tail in the basal (or non-electrophile-stimulated) state, a standard phenotype expected by these MOs, and hence the validation was not an issue for us (e.g., see Supporting Figure 3A in revised manuscript). Furthermore, these MOs have been very widely used^7^, and agree with previously-reported knockout fish where appropriate^8^ and consistent with previously-reported^9^, as well as our own observations from Keap1-overexpression (e.g., see Supporting Figure 1C,2AB,7A)

3) Even if these MO reagents are truly effective,

Our validation data (Supporting Figure 3A-B, 4A-B, 5A-B) collectively showed that all MOs employed are effective, based on standard assays that we and others have used in the past as well as and are accepted in the zebrafish community, and also by phenotypic assays on expected downstream response. On the other hand, as we explained above, antibodies to fish/human proteins are not always available/of high-quality, and isoform specific.

The authors over-interpret their morphant phenotypes. For instance, they state that the keap1a/keap1b double knockdown increases GFP reporter expression in the tails of Tg(gstp1:GFP) embryos relative to a control MO (Figure 2A). While the change may be statistically significant (P value = 0.0014), the magnitude appears to be only ~1.3-fold.

We note that the observed magnitude is actually the same as the magnitude of GFP-signal upregulation following HNE treatment of fish not treated with MO (Figure 5A,B,C), and thus these responses are genuine. Furthermore, the dynamic range of AR responses in cells and animals is known to be modest across studies made by independent labs; please see our responses to Reviewer 1 (Query #3) and to Guest Editor (Point #2) where we discussed the dynamic range of AR in detail, and our further explanations below.

It is unclear that this difference is biologically significant or mechanistically meaningful. The authors also state that nrf2b knockdown leads to "hyperelevated AR response in the tail." Again, the P value may be <0.0001, but fold-change is ~1.6 and the individual data points are broadly distributed. How confident can we really be that "Nrf2b countermands electrophile-induced AR-upregulation"?

Please also see our responses above and those to Reviewer 1 (Query #3) and to Guest Editor (Point #2). First, Nrf2b is known to countermand AR, “Nrf2b represses activity of an ARE-GFP reporter construct in vivo”^10^. Second, it is well-known in the field that AR has a relatively low dynamic range in most organisms and cells. The maximum output from the AR reporter is around 2-3 fold in many orthogonal assays (targeting both ectopic and endogenous genes), as well as within the same readouts such as ARreporter assay used across many laboratories in multiple model animals. This is true even when bolus dosing with excess electrophiles is performed. So, indeed, the values the reviewer claims to be “irrelevant” are 10-20% of maximal output. Our lab is able to recapitulate the same fold change of response that was reported by other independent labs under bolus flavonol-derived Michaelacceptor electrophile in Figure 4B: Zhang et al. *Molecules* 2019;24(4):708; in zebrafish, following treatment with nitro-olefin-derived Michael acceptor electrophiles in Figure 5A-B: Tsujita et al. *Genes Cells* 2011;16(1):46-57.

In the paper that the Reviewer cites above (2008 JBC) Nrf2-mRNA-injection was used, which can give higher AR fold changes. But tampering with the regulation to such an extent, especially in an embryo, is typically not a good idea, especially for MO experiments where we are trying to examine specific changes. (Note: we are careful to ensure that our Keap1-expression in zebrafish embryos, under Z-REX setting, is similar to endogenous levels; see Supporting Figure 1C).

Statistical significance is indeed the standard tool used to define whether a change is real or not; without this level of numerical insight and the concept of statistical power, in real biological experiments, often no objective would be gleaned. The necessity of using statistical significance to help assign biological significance is actually due to the inherent variation of biological data. We believe that it would be not be of responsible nature for any researcher to brush off such statistically-significant effects by passing them off as ‘not biologically significant’, when they are significant with respect to the tolerance of the system, and are carefully controlled for. *We would point out that there was no significant change in the head, even in the same fish under at least some of the conditions mentioned.* That being said, our focus is to examine the reason behind the interesting head versus tail responsivity, for which we performed correct and careful analyses and came to realize that Nrf2 expression is not closely linked to responsivity differences between head and tail, and zKeap1-paralog expression. Finally, as the p-value gives an estimate of the chances of being fooled by the experimental data and thus the level of confidence, we are perplexed by the question “how confident can we really be…”.

These issues extend to their cell culture experiments as well. For example, since co-expression of Keap1a and Keap1b does not suppress basal antioxidant responses to a greater extent than Keap1a alone (Figure 7D), the authors conclude that Keap1a affects the ability of Keap1b to affect these responses. An alternative explanation (and arguably more likely) is that Keap1a overexpression alone can maximally suppress basal antioxidant responses, especially since Keap1b appears to be less active in these assays (Figure 7C).

This conclusion proposed by the reviewer is ruled out by supporting Figure 12B wherein we went at length to ensure a good dynamic window of the assay is maintained and saturation is not reached in our system. Please also see Author response image 2 and 3.

We also explained this point clearly within the original manuscript text.

4) Some of the other MO phenotypes are difficult understand in light of other results in the paper and/or missing information. For instance, Figure 1D shows that nrf2a is expressed at much higher levels than nrf2b in the zebrafish embryo tails. Yet nrf2a and nrf2b MOs have essentially the same effects on tail GFP expression in Tg(gstp1:GFP) embryos (Figure 2A)-a surprising result if "Nrf2b is a minor contributor" as described by the authors in the Discussion section.

This question stems from misunderstanding/the reviewer missing a section of our text:

On Page 7 of the manuscript, we explained clearly: “Note: the data here (*referring to qPCR data*) are for the *whole* tail and *whole* head, and thus, do not necessarily reflect the expression levels in responsive tissues, or those present in the areas where GFP is expressed”; that is tail- versus head-sectioning (which is also clearly diagramed in Figure 1C, original manuscript) in qPCR experiments, is *not* the same as the reporter fish [for which images , and schematics (note the green patches denoting expression of GFP therein) are provided are provided], since the latter (reporter fish) only report on a fraction of the head and the tail.

We are sorry for the confusion: Nrf2b is not mentioned in the Discussion section, but if the reviewer were referring to the statement in the relevant section in the body of the text, Nrf2b is referred to as a minor contributor *in the whole of the tail*; *not in the part of the tail where we measure AR*. We hope this is now clear.

Were the nrf2a and nrf2b MOs used at the same dose? Do they have the same knockdown efficiency?

MOs injected were standardized (we have now expanded the section). The effects on Nrf2a/b on basal AR are similar, although how this correlates with magnitude of knockdown depends on numerous factors that are beyond the scope of this paper.

Also, in Figure 2A-B. the keap1b MO does not alter tail GFP expression in Tg(gstp1:GFP) embryos in basal conditions, but it suppresses the differential effects of NE and DMSO on this reporter. If the morphant phenotypes reflect on-target activities, this means that Keap1b must promote NE-mediated activation of antioxidant responses, an idea that counters the conventional wisdom that NE directly suppresses the ability of Keap1 proteins to induce Nrf2 degradation. The authors should provide a mechanism that can account for these findings.

The reviewer appears to have missed the point of the paper. We also point out that these data are consistent with previous reports derived from mRNA injection^8^. We show that zKeap1b can upregulate AR to a higher extent than zKeap1a upon electrophile treatment in cells (Figure 7E and 7F). This would explain why knockdown of zKeap1b suppresses electrophile-induced AR-upregulation (Figure 2B). zKeap1a also suppresses zKeap1b’s ability to mount AR (Figure 7F), hence zKeap1a’s knockdown would stimulate AR in the electrophile-stimulated state (Figure 2B). These outputs are therefore self-consistent, and consistent with previous data.

5) Important controls are missing from some experiments. For example, the authors examine how "bolus LDE treatment results in different extent of hKEAP1-labeling" in Figure 6. Presumably they are using the HNE(alkyne) and NE(alkyne) labeling and Cy5 click chemistry results as evidence that these electrophiles efficiently modify KEAP1 expressed in zebrafish embryos. However, without testing the effects of these electrophiles on embryos that are not expressing exogenous KEAP1, it is not possible to know how much of the observed Cy5 signal is due to KEAP1 labeling vs. non-specific labeling of endogenous proteins.

The corresponding result section title (page 13, original manuscript) indicates: “Extent of the fish proteome labeling following bulk LDE exposure closely mirrors that of AR Induction”, and the subsequent goal of the stated experiments details (Click labeling of LDE treated fish). These specific experiments are *only to examine electrophile permeation into the fish*; *not to assess Keap1 labeling*. We have also doubly ensured that the associated figure legends are also as clear as possible.

The reviewer might have missed the data where Keap1-specific labeling in vivo was directly assessed by Click-Biotin pulldown experiments (see Supporting Figure 8,9,11), using hKeap1 as a proxy (which we know is overall a similarly good sensor to zKeap1’s).

6) The authors overlook inconsistencies in their observations. For example, the ability of Ht-PreHNE to convey light-inducible endogenous antioxidant responses in both the tail and head of zebrafish embryos expressing HaloTag-functionalized KEAP1 (Figure 3B) seems to contradict the inability of Ht-PreHNE to convey light-inducible GFP expression in the heads of Tg(gstp1:GFP) embryos expressing the same construct (Figure 3C-D). If the GFP reporter does not faithfully recapitulate endogenous antioxidant responses, what do the differences in tail and head GFP expression mean? This is a particularly important question since the Tg(gstp1:GFP) embryos are used extensively in the authors' study.

We are sorry that the reviewer is of the opinion that we “overlook” this point. In reality, we actually ceded in the original manuscript that the head can weakly show upregulation of some endogenous genes as assessed by qPCR (which is consistent with the spirit of the reporter data, but not the completely muted response) thus, we wrote in the original manuscript:

“Representative genes associated with drug metabolism under control of Nrf2 (Gst-isoforms, Hmox1, and Abcb6) were activated to similar levels between Z-REX and bulk HNE-treatment, and AR modulation was most prominent in the tail (Figure 3B). Z-REX mounted a weak but measurable AR in the head. We ascribe this modest AR-upregulation in the head [seen only by qRT-PCR analysis, and not by imaging of *Tg(gstp1:GFP*)] to increased sensitivity of qRT-PCR analysis compared to in vivo fluorescence-imaging, and the fact that the *gstp1* locus (used in the GFP-reporter fish) is not the most responsive in the head. By contrast, bolus HNE yielded mixed responses in most cases (Figure 3B).:”

We have also added additional illustrations (e.g., Figure 1B-C, and insets with similar illustration in all other applicable figures) and further clarified the figure legends such that the above point is now clear.

The fold changes in the qPCR are sometimes larger than in the reporter assay, hence some more fine detail can be gleaned from these experiments; furthermore tissues where GFP expression is not high, or GFP is degraded may not be taken into account in this assay. However, overall, the response to a good number of these genes is muted in the head relative to the tail, and many other genes are very clearly trending that way. To explain these minor differences, we would like to point out once again that in the qRT-PCR, we are looking at many different tissues than what we are looking at in the reporter assay, but overall trend/results remain consistent between the two readouts.

We would like to also point out that the reviewer appears to have missed one of the much-important controls: the data sets in Figure 3C (now Figure 3D) and 3D (now Figure 3E) (see from 6^th^ to 9^th^ bar from left) integrate a genuine control for Z-REX induction of AR, where Halo and hKEAP1 are expressed as two separate proteins. This control construct is shown schematically in Supporting Figure 4A in the original manuscript (and we further expanded to have it featured as a main figure, Figure 3C); and this system represents the best control for Z-REX setup in studying signaling downstream and to rule out off-target responses. (Note: it is also clearly demarked that these fish do not express the same construct as in Figure 3A or 3B). Using this split-control construct, we saw no upregulation, in neither head nor tail (Figure 3C-D and 3E), analyzed by independent readouts (qRT-PCR vs. IF-imaging).

Taken together, there is a consistent vein of logic from fish, to human cells, using several varied orthogonal approaches via which we have carefully measured AR and cell/animal responses in this work.

7) Even if the gstp1:GFP reporter is a reliable surrogate for endogenous antioxidant signaling, the authors' reliance on immunofluorescence imaging to quantify these responses is questionable. This approach is complicated by the autofluorescence and non-specific antibody binding of zebrafish embryos, and a more rigorous approach would be to assess reporter expression by quantitative western blots or qRT-PCR.

We first stress that The GFP reporter fish was developed by the same authors as the paper cited by the reviewer and this reporter line is widely available, have been validated by numerous labs, and show expected responses upon treatment with ARstimulating agents: e.g., Tsujita et al. Genes Cells 2011;16(1):46-57; Zhu et al. Free Radical Biol and Med 2016; 95:243-254.

Importantly, our original manuscript did show all the qPCR data on endogenous genes that are broadly consistent with our interpretations. We do not think that the reviewer missed these qPCR data since the reviewer’s points above (see reviewer 2 Query#6 above) mentioned some of these qPCR-data.

Furthermore, the reviewer thinks that western blot (for tail only?) would be more accurate than imaging? Or for head only? We can explain precisely why we used IF, and indeed Reviewer 1 specifies further reasons for using IF [please see our responses to Reviewer 1 Query (1) above and Author response image 1. We can rule out non-specific binding by comparing signal in wt/wt fish relative to gst:GFP/wt fish (Author response image 4). Hence, we know precisely the background of the antibody.

**Author response image 4. sa2fig4:** Comparison of GFP immunofluorescence (IF) staining between WT and transgenic fish. WT (top) or *Tg(gstp1:GFP)* (bottom) embryos (1.5 dpf, injected with control MOs) were dechorionated, fixed, and immunostained for GFP as described in the original manuscript. Scale bars, 500 μm.

8) The authors should also explain why their studies of KEAP1 function have focused on zebrafish embryos that are at least 24 hours post fertilization (hpf) (and certain T-REX experiments utilized 48-hpf embryos). Zebrafish studies using mRNA-mediated overexpression ae typically limited to the first 24 hours of development since the exogenous mRNA is rapidly degraded. Accordingly, the 2008 J. Biol. Chem. paper by Li et al. examined Keap1a and Keap1b activity in zebrafish gastrulae (8 hpf).

The reviewer unfortunately missed the data: hKeap1 protein is still present at the time points we use (See Figure 4C, and Supporting Figures 1C, 8B, 9, 11); we also have very clear and well-designed controls to show that the expressed protein is necessary for our phenotypes, ruling out effects of the compound alone, light alone, etc. (see the same figures cited above). Indeed, we have used a similar set up/time-scale, to study Akt signaling and Ube2V2 signaling also in zebrafish [Long et al. *Nat Chem Biol* 2017; 13:333338; Zhao et al. *ACS Cent Sci* 2018; 4(2):246-259], where both these proteins were shown to be labeled following Z-REX execution 36 h post fertilization and mRNA-injection.

Aside from what has been done in the past (papers cited above), there are also experimental reasons to study fish at the 36 h stage. Had we followed the 2008 JBC paper the reviewer refers to frequently, where experiments are performed at 8 h post fertilization (*where the tail etc., are not visible*), we would *not* have been able to perform our intended experiments looking at tissue-specific variation in responses… Around 36 h these structures are obvious, rendering our intended experiments actually achievable.

As for the experiments performed 48 h post injection, our intention was to examine *the latency* of AR upregulation following ZREX-assisted Keap1-specific modification, which we recognize is a function of the duration of the expressed protein, and the mRNA, but in reality, is the only fair way to examine longevity of the signal, and to show that we have not in some way seriously impacted zebrafish development. We believe that we are better off not ignoring these potential issues.

We hope that these explanations collectively clarify why the timing we chose was relevant and necessary.

9) The electrophile-labeling results for Ht-PreHNE and Ht-PreNE shown in Figure 7B and Supp. Figure 13 are puzzling. They suggest that ~20% of KEAP1 is HNE(alkyne)-modified and ~12% is NE(alkyne)-modified. Since it is believed that electrophile-mediated activation of NRF2 acts by stoichiometrically suppressing KEAP1 function, the authors should explain how these results relate to the ability of Ht-PreHNE and Ht-PreNE to induce antioxidant responses in embryos expressing HaloTag-functionalized KEAP1 (e.g., Figure 3, Figure 4, Supp. Figure 4, Supp. Figure 7, and Supp. Figure 9).

Our published work on multiple systems and pathways collectively indicate that substoichiometric labeling of a host of electrophile-sensor proteins by HNE and other native reactive lipid-derived metabolites is sufficient to elicit either gain-offunction or dominant-negative signaling; in this aspect, native electrophile signaling behaves similarly to typical signalamplification mechanisms operative for classical enzymatic-PTMs such as phosphorylation:

Liu et al. ACS Central Science 2020 *in press* 10.1021/acscentsci.9b00893

Long et al. RSC Chemical Biology 2020 *in press* 10.1039/D0CB00041H

Poganik et al. Helv Chim Acta 2020 *103*, e2000041

Poganik et al. Front Aging Neurosci 2020 *12*, 1

Long et al. Methods in Enzymology 2020 *633,* 203-230

Poganik et al. FASEB Journal 2019 *33,* 14636-14652

Poganik et al. Trends in Biochem Sci 2019; 44(4):380-381

Long et al. Antioxid Redox Signal 2019.7894

Long et al. Curr Opin Chem Biol 2019; 51:48-56

Long et al. Front Chem 2019; 7:125

Liu et al. Trends Biochem Sci 2019; 44(1):75-89

Parvez et al. Chem Rev 2018; 118(18):8798-8888

Surya et al. ACS Chem Biol 2018; 13(7):1824-1831

Van Hall-Beauvais et al. Curr Protoc Chem Biol 2018;10(3):e43

Zhao et al. ACS Central Science 2018; 4(2):246-259

Poganik et al. Bioessays 2018; 40(5)

Long et al. Biochemistry 2018; 57(2):216-220

Long et al. Cell Chem Biol 2017; 28(8):944-957

Long et al. Cell Chem Biol 2017; 24(7):787-800

Long et al. Nat Chem Biol 2017; 13:333-338

Long et al. ACS Chem Biol 2017; 12(3):586-600

Parvez et al. Nat Protoc 2016; 11:2328-2356

Long et al. Chem Res Toxicol 2016 29(10):1575-1582

Long et al. J Am Chem Soc 2016 138(11):3610-3622

Lin et al. J Am Chem Soc 2015; 137(19):6232-6244

Parvez et al. J Am Chem Soc 2015; 137(1):10-13

Fang et al. J Am Chem Soc 2013; 135(39):14496-9

We are working hard to understand molecular reasons for these effects but based on the published and ongoing work, precise mechanisms are also system-dependent. In the case of Keap1, one simple possibility is that the free Nrf2 is much less than the ~20% Keap1 that is modified, hence making “substoichiometric Keap1-labeling” super-stoichiometric compared to free Nrf2.

However, this simple mechanism is not possible in many of the other proteins we have studied.

Our added data (Figure 8A-B, Supporting Figure 15), show that zKeap1b can release Nrf2 upon electrophile labeling. Clearly going from a degrader, and cytosolic anchor to a releasor can explain how relatively small amount of labeling can raise AR.

10) The authors' studies of Keap1a and Keap1b function in HEK293T cells assumes that both zebrafish proteins maintain their functions in mammalian cells. There is no guarantee that this will be the case, as there will be species-specific structural differences in E3 ligase components and their substrates (not to mention different culture temperatures).

The reviewer unfortunately missed the validation data in our original manuscript where we actually demonstrated that zebrafish Keap1 paralogs are both functional and electrophile responsive when expressed in cultured human cells: they were able to suppress AR in HEK293T (Figures 7C, 7D and Supporting Figures S14 and Author response images 2 and 3) and they were labeled efficiently by T-REX (a feat that is only possible to the most active electrophile-sensor proteins) (Figure 7B, Supporting Figure 13). Keap1a and Keap1b are also expressed similarly at similar plasmid loads (indicative of, although not proof of, similar stability) but certainly meaning that we are comparing the same amount of active protein. Finally, our latest data (Figure 8A-B, Supporting Figure 15), would indicate that release of Nrf2 from Keap1b is responsible for AR upregulation, meaning that Cul3 etc in human is likely not important.

Based on these cell-based assays, the authors claim that Keap1b is less efficient than Keap1a at suppressing antioxidant responses. However, this interpretation is at odds with the previous studies by Li et al. (J. Biol. Chem., 2008) that show Keap1a and Keap1b have comparable effects on Nrf2 stability and antioxidant responses in zebrafish embryos. The authors should comment on this difference.

As explained above, several key validation data sets that we presented in our original manuscript were unfortunately overlooked. Ignoring this, we can break down the reviewer’s question above to two questions: (i) whether the data we presented here are necessarily different from the 2008 JBC paper the reviewer quotes; and (ii), whether the reviewer’s demand for comparison is a relevant question.

As for the first point, as we point out above, the 2008 JBC paper is not really quantitative (see explanations above, Page 8-19), did not compare even equal amounts of mRNA injected between the two constructs, (although this was claimed to give similar protein expression, this trend was not examined for each concentration, and could readily be variable), and we do not know what aspect of Keap1 dominates these assays *when excess GFP-Nrf2 (not full length) is introduced in developing embryos*. As we do know that the data derived from the cell culture experiments with Keap1-isoforms is recapitulated in the fish AND in human mutants, we personally think that the reviewer’s concerns are, with respect, not borne out by the data. Finally, the same authors later showed in a follow up paper that overexpressed Keap1a/Keap1b in zebrafish embryos behave similarly to how we propose in terms of electrophile responsivity as single proteins (the heterozygous state was not investigated) ^8^.

As for the second point, the key relevance here is to investigate the effect in the “mixed” system (although we did first characterize the single expression constructs and ensure that individual plasmid amounts when used alone as well as in combination were such that the dynamic range of the assay was maintained; see Supporting Figure 14A-B; note, Y-axis in 14B is in log-scale; and also see Author response images 2 nd 3). This mixed system shows properties, in both the hKeap1-C273I state as well as the Keap1a/1b state (Figure 7 C-F), that are consistent with our model. Given that all our Keap1 proteins are active in two assays, the human/human-mutant system accounts for the E3-machinery, and the fish data are consistent with the data derived from zKeap1a/1b, we do not see how such variables are relevant or even reasonably fair to raise. *These points further ring true in the light of data showing that it is release of Nrf2 from zKeap1b that is responsible for the AR upregulation properties of zKeap1b upon exposure to electrophiles. As this behavior has also been reported in zebrafish as well, altogether these data fully resolve reviewer’s questions/concerns.*

Finally, we note that the human Keap1-protein is also active when injected into the fish (as is true for most human proteins studied in zebrafish, hence its use as a model system in the field). This observation also indicates the interchangeability of the Keap1/Nrf2 system, as is commonly found for zebrafish/human, as the Guest Editor mentioned regarding human mRNA injection to rescue the deficiency of the corresponding endogenous gene in fish (Page 29).

11) In general this manuscript is very challenging to read,

With respect, having read the reviewer’s comments, we are concerned that they may not have read the manuscript. Respectfully we note that this does not appear to be an issue shared by Reviewers 1 and 3 or the Guest Editor. We also, with respect, note that a lot of the issues that the Reviewer 2 has stemmed, as we explained above and below, from misunderstanding/confusion.

We have thus redoubled our effort to make the manuscript as clear as possible.

and as a result, mechanisms proposed by the authors are difficult to follow. For example, the scheme in Figure 8 suggests that the electrophile-modified form of Keap1b potentiates Nrf2 function. Is this really what the authors mean to convey, as opposed to proposing that electrophile modifications of Keap1b prevent this E3 ligase component from promoting Nrf2 degradation? Or are the authors proposing that modified Keap1b can negatively regulate a suppressor of Nrf2 function?

Please also see our responses to Reviewer 1 Query no. 5 (above). In the electrophile modified state, Nrf2 that is built up on zKeap1b is released. It is this release that promotes AR. zKeap1a (presumably in both modified and unmodified states) suppresses this behavior. We have expanded the figure legends in Figure 8C and indeed throughout the manuscript for improved clarity.

The authors also suggest the Keap1a is "a dominant-negative regulator of the antioxidant response in both basal and electrophile-stimulated states." However, Keap1-mediated Nrf2 degradation under basal conditions is believed to be a catalytic rather than a dominant-negative mechanism. And if electrophile-modified Keap1a acts as a dominant-negative regulator, what is it inhibiting in a stoichiometric manner?

The presence of zKeap1a suppresses the AR upregulation induced upon electrophile modification of Keap1b. This is apparent in both the fish MO experiments (Figure 2A, Supporting Figure 3,4,5) and in the cell-based experiments (when plasmids are mixed) Figure 7D,F, and Author response images 2 and 3. It is now clear that zKeap1a is able to prevent zKeap1b from releasing Nrf2 to allow AR to be upregulated (Figure 8A-B, Supporting Figure 15). This behavior is related to zKeap1a’s inability to release Nrf2 upon electrophile labeling, which has been reported by others. We have now improved the figure/legends in Figure 8C to make things more clear.

12) For the multiple reasons described above, authors' model for Keap1a and Keap1b function (Figure 8) does not have strong experimental support. Their model is also mechanistically counterintuitive. If Keap1a is the primary antagonist of Nrf2/antioxidant responses under basal conditions, then electrophile-induced Nrf2 signaling should be predominantly due to Keap1a modification and suppression.

We are unsure why this should hold true in general, especially seeing as fold-increase in AR upon zKeap1b-modification is higher than what occurs upon Keap1a-modification in cell culture (Figure 7E); (note ability to suppress AR in the basal state and ability to upregulate AR during electrophile treatment are NOT necessarily positively correlated, especially as fold changes induced by electrophiles are small and ability to suppress AR of each paralog is significant). The permissively of zKeap1b for AR upregulation, and the inability of zKeap1a to do likewise has also been previously reported^8^. Thus, in fact, the absolute magnitude of AR stimulated following zKeap1b-modification is significantly higher than that with Keap1a (Figure 7E, and indeed this observation is also very clear in Reviewer’s only figure 4 above), and one may hence naively expect the AR-upregulation to be greater in the “heterozygotic” state, where the AR is lower than in the case where there is only zKeap1b (Figure 7F). But it is not the case since the response is muted in the heterozygotic state. Remarkably, our latest data investigating the respective binding interaction with Nrf2 mirror these observations Figure 8A-B, Supporting Figure 15.

The authors propose instead that electrophile modification of the purportedly less effective antagonist, Keap1b, is the primary driver of increased Nrf2 activity and that modified Keap1a actually suppresses Nrf2 function further. It is difficult to see how electrophiles could mount 3- to 4-fold increases in antioxidant responses under these mechanistic constraints.

The changes in cell culture are more like 2-3 fold (observed when zKeap1b was expressed) (Figure 7E and F). For the most part, this amount of change is sufficient to cover the gamut of AR we measured in the fish. These low dynamic range/magnitude of changes are common in the field and we have ourselves considered and validated extensively; please see our responses to Reviewer 1 (Query 3), and to Guest Editor (Point 2).

To restate this differently, for clarity, as shown in Figure 7E and 7F, zKeap1b can mount a 2-3-fold response in electrophile-induced AR (despite showing a lower suppression effect on basal AR, Figure 7C). However, in the presence of zKeap1a, where if anything, basal AR is *lower* (Figure 7D), i.e., the output in terms of AR-upregulation is muted. Clearly, the electrophile-induced AR upregulation due to zKeap1b [which was ~3-fold (Figure 7E), even though basal AR due to Keap1b was higher, Figure 7C] is thus weakened.

Long et al. Akt3 is a privileged first responder in isozyme-specific electrophile response 2017 Nat Chem Biol, 13, 333-338; Liu et al. Precision Targeting of pten-Null Triple-Negative Breast Tumors Guided by Electrophilic Metabolite Sensing 2020 ACS Central Science, in press, DOI: 10.1021/acscentsci.9b00893Parvez, et al. T-REX on-demand redox targeting in live cells 2016 Nature Protoc. 11, 2328-2356.For instance, see Long et al. 2017 Nat Chem Biol 13, 333-338; Zhao et al. 2018 ACS Cent Sci 4, 246-259; Poganik et al. 2019 FASEB J 33, 14636-14652.Kobayashi, et al. The Antioxidant Defense System Keap1-Nrf2 Comprises a Multiple Sensing Mechanism for Responding to a Wide Range of Chemical Compounds 2009, Mol. Cell Biol. 29, 493-502Parvez et al. 2015 J Am Chem Soc 137, 10-13; Lin et al. 2015 J Am Chem Soc; 137, 6232-6244; Parvez et al. 2016 Nat Protoc 11, 2328-2356; Long et al. 2017 Cell Chem Biol 28, 944-957; Poganik et al. 2019 FASEB J 33, 14636-14652; Poganik et al. 2020 Helv Chim Acta 103, e2000041; Fang et al. 2013 J Am Chem Soc 135 14496-14499; Poganik et al. 2020 in revision in Nat Commun; Long et al. 2020 in revision in Nat Protoc.Long and Urul, et al. Precision Electrophile Tagging in *Caenorhabditis elegans* 2018 Biochemistry 57, 216-220For instance, see: Sant et al. The Role of Nrf1 and Nrf2 in the Regulation of Glutathione and Redox Dynamics in the Developing Zebrafish Embryo 2017 Redox Biol 13, 207-218; Fuse et al. Nrf2 activation attenuates genetic endoplasmic reticulum stress induced by a mutation in the phosphomannomutase 2 gene in zebrafish 2018 PNAS 115, 2758-2763; Goessling et al. S-Nitrosothiol Signaling Regulates Liver Development and Improves Outcome following Toxic Liver Injury 2014 Cell Rep, 6, 56-69; Timme-laragy et al. Nrf2b, Novel Zebrafish Paralog of Oxidant-responsive Transcription Factor NF-E2-related Factor 2 (NRF2) 2012 J. Biol. Chem. 287, 4609-462716 Mills et al. CRISPR-Generated Nrf2a Loss- And Gain-of-Function Mutants Facilitate Mechanistic Analysis of Chemical Oxidative Stress-Mediated Toxicity in Zebrafish 2020 Chem Res Toxicol 33 426-4357 Kobayashi et al. The Antioxidant Defense System Keap1-Nrf2 Comprises a Multiple Sensing Mechanism for Responding to a Wide Range of Chemical Compounds 2009 Mol Cell Biol 29, 493-50218 Timme-Laragy, et al. Nrf2b, Novel Zebrafish Paralog of Oxidant-responsive Transcription Factor NF-E2-related Factor 2 (NRF2) 2012 J Biol. Chem. 287, 4609-4627.

[Editors’ note: what follows is the authors’ response to the second round of review.]

Essential revisions:– The revised document includes a number of 'validation' studies, but they are not consistent with expectations in the field in terms of controls and other validation. That does not mean their interpretation is necessarily wrong, it's that from the controls they do present, we do not know the boundaries where the work is likely limited in interpretation. Morpholino validation includes on-targeting and off-targeting questions. The authors present A LOT of work, but there are issues with both. (A) the authors appear to spend all of their time worried about on-targeting/efficacy measurements.

We are pleased that the reviewer noticed that we focused on on-target validations and made this section robust. We are sorry we did not explain why we eschewed looking at “off-target” effects in this specific manuscript, as we thought that the manifold reasons, as we now explain below, were apparent and because we believe that explaining why any of the panoply of possible experiments were eschewed is typically uncommon in a manuscript. Respectfully, we would also point out that off-target effects can also be validated by studying a simpler and/or an alternative model system; in this case, we use cultured cells.

For the translational blocking reagents, they do appear to be inhibiting their targets (and they now report their dosing per embryo). But their splice-site reagents are showing 30% or more wt RNA still present in all but one circumstance. Will only 70% knockdown be enough to show a phenotype? Not clear.

We are confused by this question.

– Firstly, the reviewer is likely aware that MOs rarely suppress 100% of their target protein expression. Indeed, this trait is shared by most RNAi methods, especially in higher eukaryotes, where we and others have reported typically 90-50% knockdown. Even with 90-50% knockdown, we have observed phenotypically-relevant outputs (. Fu et al., 2018 Nature Chemical Biology *14* 943-954 *Nuclear RNR-α antagonizes cell proliferation by directly inhibiting ZRANB3*; Zhao et al., 2018 ACS Central Science *4* 246-259 *Ube2V2 Is a Rosetta Stone Bridging Redox and Ubiquitin Codes, Coordinating DNA Damage Responses*; Poganik et al., 2019 FASEB J *33* 14636-14652 *Post-transcriptional regulation of Nrf2-mRNA by the mRNA-binding proteins HuR and AUF1*; Liu et al., 2020 ACS Central Science *6* 892-902 *Precision Targeting of pten-Null Triple-Negative Breast Tumors Guided by Electrophilic Metabolite Sensing*; Long et al., 2020 Cell Chemical Biology *27* 122-133 *Clofarabine Commandeers the RNR-α-ZRANB3 Nuclear Signaling Axis*.). Furthermore, knockout/mutagenesis typically creates truncations/modified proteins that could retain partial activity, so this issue is not unique to RNAi, although it is more easily quantifiable in RNAi.

– Secondly, as the reviewer is likely aware, both zKeap1a, a suppressor, and zKeap1b, a promoter, are present in the tissues we are interested in. So, 70% (global) suppression of one isoform likely *underestimates* the perturbation that occurs in the system as a whole. In line with this logic, there is ample evidence that many more genes than previously thought are haplo-insufficient (e.g., Elledge et al. 2013 Cell *155* 948-962; Hurles et al. 2010 PLoS Genetics *6* e1001154). Hence 50% loss has come to be a minimum standard in fields such as targeted protein degradation and RNAi. 70% (which is a pessimistic estimate) suppression healthily exceeds this criterion.

B) Off-targeting questions. In the Stainier guidelines paper, there are clear approaches to experimental design to address this question. Comparing to a mutant is the preferred approach, followed by RNA Rescue etc. We see none of this in this manuscript, or even a reason why they did not do these logical experiments.

Like all *guidelines,* those of Stainier et al. need to be implemented with care and understanding of the apposite caveats, both general and those intrinsic to the specific experiments at hand. *In our particular situation,* we are presented with a system wherein, overexpression of each zKeap1-isoform separately by mRNA injection in fish mirrors the phenotypes we observe upon knockdown; i.e., zKeap1a shows dominant-negative behavior for electrophile-induced AR in fish [work from elsewhere, (Kobayashi et al. 2009 Mol Cell Bio, cited as Ref. 29 in the previous manuscript version) performed with high statistical power/penetrance] and in cells (work in this manuscript), and zKeap1b is permissive for electrophile-induced AR when in excess, relative to zKeap1a in fish (work from elsewhere, i.e., by kobayashi et al. 2009 cited above). Indeed, at several places in the previous version of revised manuscript, these aspects were discussed (with Ref. 29 cited).

We note in passing that these data are already well in agreement with our own, which should have assuaged any concerns from the reviewer. However, Keap1 isoforms are the major upstream regulators of Nrf2, occupying the major node of AR regulation orchestrated by HNE-like electrophile, lying immediately upstream of AR-upregulation. Hence, in the light of the previously published data involving isoform-specific mRNA-overexpression, we will see “rescue” of our phenotypes, irrespective of whether or not such observation stems from MOs functioning on expected targets only: that is, irrespective of whether or not zKeap1a-knockdown has occurred, (over)expression of zKeap1a will suppress electrophile-induced AR in the tail, as zKeap1a levels will be elevated; similarly, irrespective of whether or not zKeap1b knockdown has occurred, (over)expression of zKeap1b will elevate electrophile-induced AR.

The use of human KEAP1, whose function cannot complement loss of zKeap1a anyway, is equally unlikely to be informative (as we show in the T-REX experiments in HEK293T cells). Thus, we have no unequivocal way to rule out that “rescue” is observed simply as a consequence of overexpression, and likely grounds to suspect that any reasonable effort to rescue would be confounding. Taken together, the rescue experiment with the respective targeted zKeap1-isoform is inherently likely to produce desired results. We thus believe that by default, such scenarios should be avoided when planning experiments and data so derived are not valid for MO validations, and thus in our opinion should not be requested by the reviewers.

These concerns are magnified given the poor dynamic range of AR. Such confounding outputs are common, even intrinsic, when dominant-negative effects are at play. We do not believe Stainier mentioned such a system as a case where RNA-rescue may be confounding in their paper, but as they are not uncommon in the biological canon, we should be aware of such caveats. Hence, we rightly obviated RNA-rescue experiments, and in the previous revision, we are sorry that an explanation for this was not included, as our concerns are generic, and that further work in an independent model system was also performed in the manuscript. Nonetheless, upon reviewers’ request, we have now added the reasons in this current revised manuscript.

The reviewer is correct that comparison to, or more correctly, replication of MO experiments in “mutants” may assuage concerns of off-target behavior. However, in the case of zKeap1a/b, such mutants are *not* available. Furthermore, we have found significant issues with implementation/use of Keap1 mutants in general. In electrophile/developmentally-responsive systems, where other regulators can assume important roles upon knockout, such as AR, such mutants are unlikely informative anyway. But more importantly, there are further issues with RNA-based rescue in the specific context of this manuscript. Unlike rescue of RNAi in cell culture, where expression levels of the protein upon rescue can be readily quantified and normalized (even at the single cell level), mRNA injection in fish does not mirror canonical tissue distribution. Hence, one cannot be sure if putative “rescue” or otherwise, be due to severe localized overexpression/system perturbation, or the replenishment of the wild-type state. As we are investigating divergent tissue-specific effects in a complex and delicately balanced binary system oscillating between dominance of a suppressive or permissive paralog, mRNA-based rescue is further likely to be uninformative.

Finally, as we are investigating Keap1/Nrf2 system, which is held as the principal mechanism through which cells defend against electrophile stress (note: overexpression of zKeap1a suppresses all AR stimulation spurred by HNE-like electrophiles), we believe our interpretations and plans of action are logical and with ample precedent. The fact that in embryos, zKeap1a behaves as a dominant-negative regulator of electrophile response underscores this point, just as much as the same data render RNA “rescue” improper.

In the wake of these logical issues, we further submit that performing mRNA-based rescue just for the sake of adhering to Stainier guidelines is likely contrary to the spirit of the guidelines themselves. Indeed Stainier et al. say:

“Of course, there will be exceptions when the full set of guidelines cannot be followed.”

We therefore moved to a more controlled system and investigated the specific effects of the individual zebrafish proteins or a mixture of the twain. We would also point out that the fish and cell-based data are reinforcing (a key point of the paper that may have been missed by the reviewer): our data from cell culture, a well-controlled system, are strongly consistent with our own MO data and those from overexpression by mRNA injection carried out by an independent laboratory (Kobayashi et al. 2009 Mol Cell Biol, cited as Ref. 29). As we see congruence between the trine, we submit that our data are compelling.

We have now added the following to the text:

In the Results section:

“In the light of previous data indicating Keap1a suppresses electrophile-induced AR and Keap1b performs an opposing function, and our own data that showed a complex interplay between the two isoforms in the same tissues, we recognized that rescue of the effects of these MOs by mRNA injection is prone to dominant factors associated with the expression of the specific Keap1 isoforms. Hence, although rescue is a control suggested in the Stainier guidelines for MO usage, where possible, we eschewed further experimentation using MOs, and investigated these effects using complementary methods in fish and cell culture.”

In the Discussion section:

“We point out that such systems are indeed apposite for study by this combination of fish and ectopic expression in human cells. This is because of the control offered by ectopic expression, and because of the overall dominant-negative effects conferred by the zKeap1a isoform. The latter render interpretation of data derived from MO rescue, particularly in a tissue specific manner difficult to interpret.”

In sum, we reiterate that the Stainier paper is actually quite definitive that none of its suggestions are set in stone. In our opinion, advocating such criteria be blindly followed would devalue what is overall a thoughtful, measured, and realistically cautious piece of scientific opinion. We further point out, respectfully, that the Stainier paper focused *entirely* on experiments carried out exclusively in fish, which is manifestly not the case in our manuscript either.

– Statistical assessment – We are concerned about the confidence interval numbers (we think that's what's shown) compared to the data distribution shown in this manuscript. Just one example – Figure 2A: comparing control MO1 with Keap1a+Keap1b MO injections. This should show the largest differential.

We first apologize for not being clear what the numbers above each set of points represent. They represent P values derived from a two-tailed t-test between the points shown (this has now been specified in each figure legend).

Respectfully, the assertion above made by the reviewer is untrue. Taking Figure 2A as an example, we would like to clarify several points. Firstly, the magnitude of *suppression* (e.g., by Nrf2a MO) may be larger than the elevation induced by zKeap1a/b suppression (and the largest differential should always be between the highest and the lowest points, neither of which is the Control-MO1 in Figure 2A); secondly, and more importantly, the difference between the mean of one set and another is not the only factor defining statistical significance. The spread of the datasets being compared (assuming this is what the reviewer meant by ‘data distribution’) also intrinsically affects ability to define if two points differ significantly; hence small differences between sets with low spread may be significantly different, whereas larger differences with higher spread may not be.

Seems to be about 30 individual data points. The authors claim these distributions are 99.86% likely to be different.

We are again sorry that the legend did not make it clear that what these numbers represent. P values derived as such may (although strictly *not* correctly) be used to infer the likelihood of the two data sets being different by chance. Either way, there is reasonable confidence that there is a difference between the twain that is not present for the Control-MO relative to zKeap1a- or Keap1b-alone-MOs.

But the visual data set suggests that the only primary difference is that 1/30 data points are above 1.6x in control MO1 versus 4/30 data points are above 1.6x.

Below we provide the individual data points. However, respectfully, the “visual data” is some oblique reference to the viewer’s eyeballing of how specific data points are distributed. Furthermore, we would respectfully point out that it is rarely informative to use <10% of the data sets as a gauge for the bulk thereof.

**Author response table 1. sa2table1:** 

Control MO1:	Keap1a+ Keap1b MOs:
0.718147	1.207962
0.878549	0.994521
0.948409	1.913047
1.186984	1.378654
0.884733	0.940147
2.546936	1.047857
0.951229	1.642165
0.754858	1.062155
0.851931	1.152548
0.92753	0.849358
0.824372	1.882668
0.846983	2.276351
0.881814	0.776578
0.835752	1.685852
1.077839	1.445002
1.096838	1.091
0.561753	0.992839
0.978788	1.430555
1.246553	1.413931
0.439883	0.520341
0.74971	2.35952
0.914093	1.018022
1.073679	1.400721
1.111666	1.259565
1.065044	1.550881
0.872091	1.638009
1.359449	1.074574
1.50905	1.613914
1.068633	1.139783
0.718333	
0.920063	
1.123783	
0.880229	
1.310766	
1.520599	
0.810581	
1.49313	
0.776006	

Visually, the numerical value just does not align with the graphical representation they present. This same issue is found throughout the manuscript.

In Figure 2A as well as the rest of the data in the manuscript, the mean and s.e.m are shown on the plots. The Control-MO and double-MO sets are clearly different with error bars that are non-overlapping (unlike the zKeap1a- and zKeap1b-MO-only sets, which are, consistent with this observation, not significantly different). It is clear that there is one outlier in the Control-MO set, which is highly elevated in AR, augmenting the spread of the control, and winnowing the chances of observing statistical significance. However, even given this outlier (and we can confirm we did not remove any outlier in any data sets), it is clear that the majority of points in the Control-MO group lie around 1, whereas the points in the double-MO set are more dispersed, with many above 1 (see Table above).

The discussion we made here using Figure 2A as an example extends to all data sets in corollary and should assuage any similar concerns.

Indeed, the data distribution graphs raise a key question of study design – how many times were these experiments independently run? We are not sure combining biological replicates is the best way to go, if that's what they did. It is not clear from the text how they derived the data points they show.

Once again, we sincerely apologize for this confusion.

Briefly, the raw data values for each set were then divided by the mean of the control data set. We are not sure what the reviewer means by “combining biological replicates”. Does this suggestion imply only one fish be shown; or that data sets on different days be ignored; or the variation between different sets on different days not be taken into account (as happens when data sets are not combined)? Neither of these options sounds best scientific practice to us.

We have now expanded the “data quantification and analysis” (on Page 29 of previous manuscript), to include additional sentences:

“Data quantitation and analysis: Imaging data was quantitated using ImageJ (NIH). For assessing AR upregulation in *Tg*(*gstp1*:*GFP)* fish, the area around the head (excluding the eyes) or the tail (median fin fold) were selected using freeform selection tool. Corresponding illustrations are included in each sub-figure for clarity. The mean red fluorescence intensity of the selected region was measured and subtracted from the mean background fluorescence intensity (region with no fish). Any non-transgenic fish larvae were excluded from the quantitation. The mean value for the control group was calculated from the raw, background-subtracted, values within that control group. Then all raw values were divided by the mean for the control. n for imaging experiments represent the number of single cells or fish embryos quantified from at least 7-8 fields of view with controls (empty vector controls for ectopically-overexpressed proteins, shRNA knockdown cell controls for endogenous proteins) shown in the figures. Unless specified, all *t* tests were two-tailed analysis. n for western blot/gels, qRT-PCR, and luciferase assays represents the number of lanes on western blots/gels under identical experimental conditions and each lane is from a separate individual replicate, no. of independent biological replicates as indicated in the figure legends.”

– Why was myc-Nrf2 used for the experiments shown in Figure 7, whereas HA-Nrf2 was used for the experiments shown in Figure 8?

In our experience HA tagging is the most sensitive and indeed it is well established that anti-HA antibody we use is one of the most high-affinity and effective known, hence it is best used for delicate experiments. We find Myc less effective in these situations. *We have now noted these reasons in the revised manuscript text, in the corresponding figure legends.*

– Figure 7. The authors say in the text (beginning on line 454): 'Intriguingly, when zKeap1b was co-transfected with sub-saturating amounts of zKeap1a, no decrease in basal AR was observed relative to zKeap1a alone (Figure 7D), implying that zKeap1a somehow affects the ability of zKeap1b to suppress AR.' Is it possible that the absence of C273 makes zKeap1a a more efficient repressor, because this cysteine when present (as in zKeap1b) senses endogenous electrophile(s)/oxidant(s) at basal state? If repression of AR by zKeap1a is already maximal (as suggested by the data), is co-transfection with zKeap1b expected to have any further effect?

It appears to us that the reviewer may be claiming that zKeap1b is somehow already partly-electrophile-inhibited in the basal state (prior to electrophile exposure/modification) (leading to impaired AR suppression). We would like to clarify that zKeap1b remains responsive to electrophiles, as shown by our data in cells (and also by mRNA-injection data by Kobayashi et al.). Indeed, zKeap1b is significantly more responsive to electrophiles than zKeap1a, which was also shown by Kobayashi et al. 2009.

– Figure 8. The authors say in the text (beginning on line 490): 'We found that zKeap1b accumulated Nrf2 in the basal (i.e., non-electrophile-stimulated) state, whereas relative to zKeap1b, zKeap1a accumulated less Nrf2, and zKeap1a/zKeap1b accumulated an amount of Nrf2 that was significantly more than Keap1a alone and less than zKeap1b alone (Figure 8A-B).'

The suppression by zKeap1a is sub-saturating, as the reviewer also mentioned in the statement directly above. We point out that we get the same effect with the humanized mutant, where basal AR suppression is quite small.

• Could these data be interpreted that zKeap1a degrades Nrf2 better whereas zKeap1b does not?

Given that basal levels of Nrf2 are higher in cells expressing similar levels of zKeap1b than zKeap1a (Supplemental Figure 15; relabeled as ‘Figure 8—figure supplement 1’ in revised manuscript), zKeap1b appears to be less efficient at *promoting* proteasomal degradation of Nrf2. However, whether this is due to an effect on the E3 ligase, how Nrf2 is presented to the ligase, how ubiquitin accumulates on Nrf2, or slow release of Nrf2 from Keap1, among numerous possibilities, is beyond the scope of this paper.

This would be consistent with a scenario where the absence of C273 makes zKeap1a a more efficient repressor, because this cysteine senses endogenous electrophiles/oxidants at basal state, causing partial inactivation of zKeap1b.

Our data show that zKeap1b retains electrophile sensitivity (most models, as well as data from T-REX single-protein-specific electrophile-labeling system, propose a single electrophile modification event), and that zKeap1b upregulates AR better than zKeap1a. We do not believe that this alternative model is at all consistent with zKeap1b being attenuated through electrophile-inactivation in the ground state (non-electrophile-stimulated state).

• Is the presence of less Nrf2 bound to zKeap1b upon treatment with NE necessarily a consequence of release of Nrf2? Was an 18-h treatment necessary to see this effect? It seems a very long time; we would have thought that if NE was causing a release of Nrf2 from zKeap1b, the release should be evident at a much earlier time point. Can the authors be fully confident that release does occur when the released protein has not been observed/accounted for?

As related in our previously-revised manuscript version, the two models that are proposed are release of Nrf2 (prevention of Nrf2’s binding to Keap1), and formation of a permanently-bound state, which could also be referred to as an “abortive ternary complex” (preventing Nrf2 of Keap1-dependent proteasomal degradation). Of the twain, only the former is consistent with our data; indeed, the data are strongly consistent with electrophile-modified zKeap1b having significantly lower affinity for Nrf2 (noting also that, affinity is a function of on- and off-rates). [Nrf2 levels are not greatly affected by electrophile treatment in this system as based on the ‘input’ lanes of the western blot (Supplemental Figure 15; relabeled as ‘Figure 8—figure supplement 1’ in revised manuscript), but amount of Nrf2 pulled down is significantly reduced (Figure 8A-B)]. Given that turnover of Nrf2 on zKeap1b is slow (as otherwise the zKeap1b state would not build up Nrf2 in the first place; Supplemental Figure 15; relabeled as ‘Figure 8—figure supplement 1’ in revised manuscript), release of bound-Nrf2 actually seems likely (unless exclusively on-rate of Nrf2 to zKeap1b were affected by electrophile modification, and off-rate were particularly slow relative to degradation). Although we, accordingly, find it highly likely that release be contributing to AR-upregulation measured, to assuage the reviewer’s concerns, we have now referred to this as “*net release*”.

In our previous revised manuscript, we did certainly discuss that other possible interpretations such as inhibition of binding (which would likely lead to release of Nrf2 anyway, unless degradation were rapid, which it is clearly not) or inhibition of activity (which in isolation would lead to a build-up of NRf2 on zKeap1b, which is not observed) are possible/contributing.

We have expanded on the already existing context below, and qualified the word “release” with the adjective “net” throughout.

“To investigate this matter further, we showed that there are subtle differences in the way zKeap1a and zKeap1b function upon electrophile treatment. Whereas zKeap1a does not undergo net release of Nrf2 upon electrophile treatment, and further does not accumulate a large amount of Nrf2 in the steady-state prior to electrophile treatment, zKeap1b net relinquishes around 40% bound-Nrf2 upon electrophile treatment, and accrues a large amount of bound-Nrf2 in the basal state prior to electrophile treatment. The mixture of zKeap1a/zKeap1b also does not undergo net release of Nrf2 upon electrophile treatment, although it can still accrue substantial bound-Nrf2 in the state prior to electrophile treatment. These data allow rationalization of our results both from zebrafish and human cell culture, and favor a model in which decrease in affinity of electrophile-modified zKeap1b for Nrf2 is a means to upregulate AR in response to electrophilic stress. It is likely that such a mode of action leads to release of bound-Nrf2 from zKeap1b upon electrophile modification, given that turnover of Nrf2 on Keap1b is slow [or otherwise build-up of Nrf2 would not occur upon zKeap1b overexpression (just as it does not occur on zKeap1a)] and generally AR-upregulation is observed even at low-electrophile occupancy on Keap1 (See, for example, (i) Parvez, et al. T-REX on-demand redox targeting in live cells 2017 Nature Protoc. 11, 2328-2356; (ii) Long et al. β-TrCP1 is a vacillatory regulator of Wnt signaling 2017 Cell Chem Biol 24 944-957; (iii) Lin et al. A generalizable platform for interrogating target- and signal-specific consequences of electrophilic modifications in redox-dependent cell signaling 2015 J Am Chem Soc 137 6232-6244; (iv) Parvez et al. Substoichiometric hydroxynonenylation of a single protein recapitulates whole-cell-stimulated antioxidant response 2015 J Am Chem Soc 137 10-13.). Inhibition of rebinding of Nrf2 post dissociation, and inhibition of newly-synthesized Nrf2 binding to zKeap1b may also contribute to AR increase in such circumstances, as binding also contributes to zKeap1b–Nrf2 affinity. The contribution of zKeap1b re(binding) to Nrf2 to AR-upregulation vis-à-vis the contribution of release of bound-Nrf2 is difficult to parse, and indeed beyond the scope of this paper. Of course, other potential/synergistic mechanisms—such as inhibition of zKeap1b-promoted Nrf2 degradation—could occur in tandem. But the comparison of zKeap1a/zKeap1b and zKeap1b systems argues in favor of net release being the key component of AR upregulation.”

• There seems to be less HA-Nrf2 following NE treatment in the input samples (Suppl Figure 15), and it is thus possible that the NE treatment affects the turnover of HA-Nrf2. The normalization for the input addresses this; nonetheless, the semi-quantitative nature of the immunoblotting technique should be kept in mind.

We are unfortunately unsure if the reviewer were suggesting that less HA-Nrf2 indicates a slower turnover, or that increased Nrf2 turnover could lead to an upregulation of AR? We hope that the expanded discussion above alleviates any remaining concerns from the reviewer. We are aware of general issues with western blots including dynamic range, and fully understand the reviewer’s concern. We thus performed multiple independent replicates (6x). We would also like to respectfully note that quantitation of western blots is a valid and trusted way to ascertain changes quantitatively.

• Do the authors know the identity of the ~37 kDa fragment in the anti-FLAG blot?

We thank the reviewer for raising this point. Given the nature of the construct used we cannot be sure what the nature of this band be. However, as it is present almost equally in both zKeap1b and zKeap1b + zkeap1a, its presence cannot be sufficient to explain the differences between these two data sets. We have expanded the corresponding figure legends both in main figure legends and supporting figure legends, to include this statement.

• The results from the zKeap1a/zKeap1b co-transfection experiment are not straightforward to interpret, because Keap1 is a dimer, and thus the simultaneous presence of several dimeric combinations (e.g. zKeap1a/zKeap1b, zKeap1a/zKeap1a, zKeap1b/zKeap1b) is possible.

We see almost complete suppression of the effect of NE on zKeap1b when zKeap1a is present, both in terms of release of Nrf2 from bound-Keap1 (Figure 8A-B), and in terms of AR (Figure 7F). Thus, the vast majority of zKeap1b is affected by the presence of equal amounts of zKeap1a. If this were not the case, we should have seen release/loss of bound Nrf2 from the complex if nothing else (as occurs with zKeap1b alone). As the bound Nrf2 is due to zKeap1b, we would expect release similarly to what we described in zKeap1b, if zKeap1b were behaving independently. To make this more clear, we have now added the below to the revised manuscript:

“There are further *potential* complications in data interpretation due to there being three possible zKeap1 dimeric forms (ignoring higher-order structures) in the zKeap1a/zKeap1b-mixed system. However, an appreciable amount of Nrf2 is built up on zKeap1 in the zKeap1a/Keap1b system (unlike upon expression of zKeap1a alone), and *no* release of Nrf2 was observed upon NE treatment (unlike upon expression of zKeap1b alone). Thus, zKeap1a exerts a significant direct effect on how zKeap1b responds to electrophiles, and hence the heterodimer, or higher-order state(s) containing both proteins, must be a significant component of the zKeap1 present in the assay.”

[Editors’ note: further revisions were suggested prior to acceptance, as described below.]

While two of the original three reviewers are now satisfied, the third reviewer who is an expert on zebrafish has indicated that the zebrafish morpholino experiments are insufficiently rigorous to be acceptable. Below are the latest reviewer comments. We regret having to communicate this disappointing news. We hope that if you ultimately choose to address Reviewer 3's concerns with the requisite additional experimental data, that you will come back to eLife with a manuscript that includes this information when ready.Reviewer #3:None of my technical concerns tied to either the on-target or off-target effects were addressed.

We have used mRNA injection of the targeted gene to rescue the effects of the relevant MOs in the paper: please see Figure 2—figure supplement 1 and 2 in revised manuscript. We are sorry that the reviewer saw fit to ignore cell culture experiments, and judge the zebrafish experiments independently of other experiments carried out in the paper. Although we think that experiments should be independently rigorous, we do not believe that it is particularly fair to ignore other evidence presented in a paper, especially when those experiments were performed precisely to answer some of the questions the reviewer raised. We would like to also point out that we are indeed generally very familiar with RNAi and the reviewer’s suggested experiments (indeed, responsible use of KD/KO approaches as can be seen, for instance, in our recent publications^1^).

1) On-target. The maximal described knockdown was 70%. In our experience, effective morpholinos are readily able to go well beyond 90 or 95% knockdown - one of the key distinguishing features of MOs over siRNA is the normal ability to be well beyond 80% knockdown.

Respectfully, the point above has several issues in reality. First, as the principal goal of an siRNA experiment is to elicit a phenotype, the success of an RNAi experiment should be judged by that metric. In our case, we see phenotypes (that are rescuable by mRNA injection). Second, we hope the reviewer can understand that with two paralogs that are similarly expressed, the absolute magnitude of knockdown is likely underestimated in this specific context under study.

2) Off-target. The authors were given several options to address this concern, and they chose to not offer any new data. The specific suggestion to follow the Stainier guidelines was rebutted, arguing they did not have mutants etc.

We have used mRNA to rescue phenotypes attributed to knockdown of the two paralogs of zebrafish Keap1. In the case of Keap1b, this was successful on 34 hpf-old embryos, identical to those used in other experiments in the paper (Figure 2—figure supplement 2). In the case of Keap1a, mRNA injection was not successful on 34 hpf-old embryos, possibly due to instabilities of Keap1a mRNA, or due to effects incurred due to tissue non-specific expression of Keap1a. We thus examined 8 hpf-old embryos. In this instance, the effects of Keap1a-knockdown were rescuable with Keap1a-mRNA injection (Figure 2—figure supplement 1). We submit that these latest data are sufficient to adhere to the guidelines the reviewer requested that we follow.

I recommend rejection of this manuscript. The zebrafish work does not achieve the level of rigor expected in the field. If the authors were to come back after confirming their results using mutants and/or true rescue experiments with morpholinos that show strong on-target efficacy, the paper could be considered appropriate to publish in eLife.

We would like to point out that in the interim, we have published another paper on zebrafish that used a similar method (Poganik, Huang, et al. 2021 *Nat Commun*, 2021**,**
*12* (1), 5736). This paper was able to answer numerous questions about the mechanism of an approved pleomorphic drug, Tecfidera. We submit that such success underscores that our approach is sound.

References

1. See, for example:

(a) Zhao, Y.; Miranda Herrera, P. A.; Chang, D.; Hamelin, R.; Long, M. J. C.; Aye, Y., Function-guided proximity mapping unveils electrophilic-metabolite sensing by proteins not present in their canonical locales. *Proc Natl Acad Sci U S A* 2022, *119* (5).

(b) Poganik, J. R.; Huang, K. T.; Parvez, S.; Zhao, Y.; Raja, S.; Long, M. J. C.; Aye, Y., Wdr1 and cofilin are necessary mediators of immune-cell-specific apoptosis triggered by Tecfidera. *Nat Commun* 2021, *12* (1), 5736.

(c) Fu, Y.; Long, M. J. C.; Wisitpitthaya, S.; Inayat, H.; Pierpont, T. M.; Elsaid, I. M.; Bloom, J. C.; Ortega, J.; Weiss, R. S.; Aye, Y., Nuclear RNR-alpha antagonizes cell proliferation by directly inhibiting ZRANB3. *Nat Chem Biol* 2018, *14* (10), 943-954.

(d) Liu, X.; Long, M. J. C.; Hopkins, B. D.; Luo, C.; Wang, L.; Aye, Y., Precision Targeting of pten-Null Triple-Negative Breast Tumors Guided by Electrophilic Metabolite Sensing. *ACS Cent Sci* 2020, *6* (6), 892-902.

(e) Poganik, J. R.; Long, M. J. C.; Disare, M. T.; Liu, X.; Chang, S. H.; Hla, T.; Aye, Y., Post-transcriptional regulation of Nrf2-mRNA by the mRNA-binding proteins HuR and AUF1. *FASEB J* 2019, *33* (12), 14636-14652.

(f) Zhao, Y.; Long, M. J. C.; Wang, Y.; Zhang, S.; Aye, Y., Ube2V2 Is a Rosetta Stone Bridging Redox and Ubiquitin Codes, Coordinating DNA Damage Responses. *ACS Cent Sci* 2018, *4* (2), 246-259.